# Proximal Point Imitation Learning

**Luca Viano**
LIONS, EPFL
Lausanne, Switzerland
luca.viano@epfl.ch

**Angeliki Kamoutsi**
ETH Zurich
Zurich, Switzerland
kamoutsa@ethz.ch

**Gergely Neu**
Universitat Pompeu Fabra
Barcelona, Spain
gergely.neu@gmail.com

**Igor Krawczuk**
LIONS, EPFL
Lausanne, Switzerland
igor.krawczuk@epfl.ch

**Volkan Cevher**
LIONS, EPFL
Lausanne, Switzerland
volkan.cevher@epfl.ch

## Abstract

This work develops new algorithms with rigorous efficiency guarantees for infinite horizon imitation learning (IL) with linear function approximation without restrictive coherence assumptions. We begin with the minimax formulation of the problem and then outline how to leverage classical tools from optimization, in particular, the proximal-point method (PPM) and dual smoothing, for online and offline IL, respectively. Thanks to PPM, we avoid nested policy evaluation and cost updates for online IL appearing in the prior literature. In particular, we do away with the conventional alternating updates by the optimization of a single convex and smooth objective over both cost and $Q$-functions. When solved inexactly, we relate the optimization errors to the suboptimality of the recovered policy. As an added bonus, by re-interpreting PPM as dual smoothing with the expert policy as a center point, we also obtain an offline IL algorithm enjoying theoretical guarantees in terms of required expert trajectories. Finally, we achieve convincing empirical performance for both linear and neural network function approximation.

## 1 Introduction

This work is concerned with the prototypical setting of imitation learning (IL) where

1. An expert provides demonstrations of state-action pairs in an environment. The expert could be optimal or suboptimal with respect to an unknown cost/reward function.

2. The learner chooses distance measure between its policy to be learned and the expert empirical distribution estimated from demonstrations.

3. The learner employs an algorithm, which additionally may or may not use interactions with the environment, to minimize the chosen distance.

In IL, the central goal of the learner is to recover a policy competitive with expert with respect to the underlying unknown cost function. IL is important for several real world applications like driving [62], robotics [88], and economics/finance [27] at the expense of following resources: (R1) expert demonstrations, (R2) (optional) interactions with the environment where the expert collected the demonstrations, and (R3) computational resources for solving the problem template.

Interestingly, while there is a vast amount of literature using optimization ideas on the IL problem template, i.e. Lagrangian duality [51, 38, 59, 63, 64], resource guarantees are still widely missing since the optimization literature focuses on the resource (R3) where IL literature mainly focuses on

36th Conference on Neural Information Processing Systems (NeurIPS 2022).

the first two resources (R1) and (R2). Our work leverages deeper connections between optimization tools and IL by showing how classical optimization tools can be applied in a linear programming formulation of IL problem guaranteeing efficiency in all (R1), (R2), (R3).

**Our contributions:** This work aims at designing an algorithm enjoying both theoretical guarantees and convincing empirical performance. Our methodology is rooted in classical optimization tools and the LP approach to MDPs. More precisely, the method uses the recently repopularized overparameterization technique to obtain the Q-function as a Lagrangian multiplier [77, 14] and solves the associated program using a PPM update with appropriately chosen Bregman divergences. This results to an actor-critic algorithm, with the key feature that the policy evaluation step involves optimization of a single concave and smooth objective over both cost and $Q$-functions. In this way, we avoid instability or poor convergence due to adversarial training [51, 122, 70, 105], and can also recover an explicit cost along with Q-function. We further account for potential optimization errors, presenting an error propagation analysis that leads to rigorous guarantees for both online and offline setting. For the context of linear MDPs [14, 121, 55, 22, 116, 7, 84], we provide explicit convergence rates and error bounds for the suboptimality of the learned policy, under mild assumptions, significantly weaker than those found in the literature until now. To our knowledge, such guarantees in this setting are provided for the first time. Finally, we demonstrate that our approach achieves convincing empirical performance for both linear and neural network function approximation.

**Related Literature.** The first algorithm addressing the imitation learning problem is behavioral cloning [93]. Due to the covariate shift problem [98, 99], it has low efficiency in terms of expert trajectories (R1). To address this issue, [100, 87, 4, 95, 111, 85, 123, 5, 68, 69] proposed to cast the problem as inverse reinforcement learning (IRL). IRL improves the efficiency in terms of expert trajectories, at the cost of introducing the need of running reinforcement learning (RL) repetitively, which can be prohibitive in terms of environment samples (R2) and computation (R3). A successive line of work started with [112] highlights that repeated calls to an RL routine can be avoided. This work inspired generative adversarial imitation learning (GAIL) [51] and other follow-up works [38, 59, 63, 64] that leveraged optimization tools like primal-dual algorithms but did not try to deepen the optimization connections to derive efficiency guarantees in terms of all (R1),(R2),(R3). Finally, a recent line of work [40, 57] in IL bypasses the need of optimizing over cost functions and thus avoids instability due to adversarial training. Although these algorithms achieve impressive empirical performance in challenging high dimensional benchmark tasks, they are hampered by limited theoretical understanding. This is the fundamental difference from our work, which enjoys both favorable practical performance and strong theoretical guarantees.

Existing model-free IL theoretical papers with global convergence guarantees assume either a finite horizon episodic MDP setting [70], or tabular MDPs [105], or the infinite horizon case but with restrictive assumptions, such as linear quadratic regulator setting [21], continuous kernelized nonlinear regulator [26, 56], access to a generative model and coherence assumption on the choice of features [58, 14], bounded strong concentrability coefficients [122] or a linear transition law that can be completely specified by a finite-dimensional matrix [70]. On the other hand, we provide convergence guarantees and error bounds for the context of linear MDPs [14, 121, 55, 22, 116, 7, 84] under a mild *feature excitation* condition assumption. Despite being linear, the transition law can still have infinite degrees of freedom. To our knowledge, such guarantees in this setting are provided for the first time.

Our work applies the technique known as regularization in the online learning literature [6, 103] and Bregman proximal-point or smoothing in optimization literature [97, 82] to the LP formulation for MDPs [73, 35, 36, 17, 48, 49, 33, 34, 102, 91, 92, 1, 65, 30, 79, 115, 67, 13, 31, 55, 106]. From this perspective, we can see Deep Inverse Q-Learning [57] and IQ-Learn [40] that consider entropy regularization in the objective as smoothing using uniform distribution as center point. In our case, we instead use as center point the previous iteration of the algorithm (for the online case) or the expert (for the offline case).

From the technical point of view, the most important related works are the analysis of REPS/Q-REPS [90, 14, 89] and O-REPS [124] that first pointed out the connection between REPS and PPM. We build on their techniques with some important differences. In particular, while in the LP formulation of RL, PPM and mirror descent [15, 47] are equivalent, recognizing that they are *not equivalent* in IL is critical for stronger empirical performance. As an independent interest, our techniques can be used to improve upon the best rate for REPS in the tabular setting [89] and to

extend the guarantees to linear MDPs. In order to discuss in more detail our research questions and situate them among prior related theoretical and practical works, we provide in Appendix A an extended literature review.

## 2 Background

### 2.1 Markov Decision Processes

The RL environment and its underlying dynamics are typically abstracted as an MDP given by a tuple $(\mathcal{S}, \mathcal{A}, P, \boldsymbol{\nu}_0, \mathbf{c}, \gamma)$, where $\mathcal{S}$ is the state space, $\mathcal{A}$ is the action space, $P : \mathcal{S} \times \mathcal{A} \to \Delta_{\mathcal{S}}$ is the transition law, $\boldsymbol{\nu}_0 \in \Delta_{\mathcal{S}}$ is the initial state distribution, $\mathbf{c} \in [0,1]^{|\mathcal{S}||\mathcal{A}|}$ is the cost, and $\gamma \in (0,1)$ is the discount factor. For simplicity, we focus on problems where $\mathcal{S}$ and $\mathcal{A}$ are finite but too large to be enumerated. A *stationary Markov policy* $\pi \colon \mathcal{S} \to \Delta_{\mathcal{A}}$ interacts with the environment iteratively, starting with an initial state $s_0 \sim \boldsymbol{\nu}_0$. At round $t$, if the system is at state $s_t$, an action $a_t \sim \pi(\cdot|s_t)$ is sampled and applied to the environment. Then a cost $c(s,a)$ is incurred, and the system transitions to the next state $s_{t+1} \sim P(\cdot|s, a)$. The goal of RL is to solve the optimal control problem $\rho_{\mathbf{c}}^{\star} \triangleq \min_{\pi} \rho_{\mathbf{c}}(\pi)$, where $\rho_{\mathbf{c}}(\pi) \triangleq (1 - \gamma) \langle \boldsymbol{\nu}_0, \mathbf{V}_{\mathbf{c}}^{\pi} \rangle$ is the *normalized total discounted expected cost* of $\pi$.

The *state value function* $\mathbf{V}_{\mathbf{c}}^{\pi} \in \mathbb{R}^{|\mathcal{S}|}$ of $\pi$, given cost $\mathbf{c}$, is defined by $V_{\mathbf{c}}^{\pi}(s) \triangleq \mathbb{E}_s^{\pi} \left[ \sum_{t=0}^{\infty} \gamma^t c(s_t, a_t) \right]$, where $\mathbb{E}_s^{\pi}$ denotes the expectation with respect to the trajectories generated by $\pi$ starting from $s_0 = s$. The *optimal value function* $\mathbf{V}_{\mathbf{c}}^{\star} \in \mathbb{R}^{|\mathcal{S}|}$ is defined by $V_{\mathbf{c}}^{\star}(s) \triangleq \min_{\pi} V_{\mathbf{c}}^{\pi}(s)$. The *optimal state-action value function* $\mathbf{Q}_{\mathbf{c}}^{\star} \in \mathbb{R}^{|\mathcal{S}||\mathcal{A}|}$, given by $Q_{\mathbf{c}}^{\star}(s, a) \triangleq c(s,a) + \gamma \sum_{s'} V_{\mathbf{c}}^{\star}(s') P(s'|s, a)$, is known to characterize optimal behaviors. Indeed $\mathbf{V}_{\mathbf{c}}^{\star}$ is the unique solution to the *Bellman optimality equation* $V_{\mathbf{c}}^{\star}(s) = \min_a Q_{\mathbf{c}}^{\star}(s, a)$. In addition, any deterministic policy $\pi_{\mathbf{c}}^{\star}(s) = \arg\min_a Q_{\mathbf{c}}^{\star}(s, a)$ is known to be optimal.

For every policy $\pi$, we define the *normalized state-action occupancy measure* $\boldsymbol{\mu}_{\pi} \in \Delta_{\mathcal{S} \times \mathcal{A}}$, by $\mu_{\pi}(s, a) \triangleq (1 - \gamma) \sum_{t=0}^{\infty} \gamma^t \mathbb{P}_{\boldsymbol{\nu}_0}^{\pi} [s_t = s, a_t = a]$, where $\mathbb{P}_{\boldsymbol{\nu}_0}^{\pi}[\cdot]$ denotes the probability of an event when following $\pi$ starting from $s_0 \sim \boldsymbol{\nu}_0$. The occupancy measure can be interpreted as the discounted visitation frequency of state-action pairs. This allows us to write $\rho_{\mathbf{c}}(\pi) = \langle \boldsymbol{\mu}_{\pi}, \mathbf{c} \rangle$.

### 2.2 Imitation Learning

Similarly to RL, the IL problem is posed in the MDP formalism, with the critical difference that the true cost $\mathbf{c}_{\text{true}}$ is unknown. Instead, we have access to a finite set of truncated trajectories sampled i.i.d. by executing an expert policy $\pi_{\text{E}}$ in the environment. The goal is to learn a policy that performs better than $\pi_{\text{E}}$ with respect to the unknown $\mathbf{c}_{\text{true}}$. To this end, we adopt the *apprenticeship learning* formalism [4, 112, 50, 51, 105], which carries the assumption that $\mathbf{c}_{\text{true}}$ belongs to a class of cost functions $\mathcal{C}$. We then seek an *apprentice policy* $\pi_{\text{A}}$ that outperforms the expert across $\mathcal{C}$ by solving the following optimization problem

$$\zeta^{\star} \triangleq \min_{\pi} d_{\mathcal{C}}(\pi, \pi_{\text{E}}), \tag{1}$$

where $d_{\mathcal{C}}(\pi, \pi_{\text{E}}) \triangleq \max_{\mathbf{c} \in \mathcal{C}} \left( \rho_{\mathbf{c}}(\pi) - \rho_{\mathbf{c}}(\pi_{\text{E}}) \right)$ defines the $\mathcal{C}$-distance between $\pi$ and $\pi_{\text{E}}$ [51, 28, 122, 70]. Then, $\pi_{\text{A}}$ satisfies the goal of IL, since it holds that $\rho_{\mathbf{c}_{\text{true}}}(\pi_{\text{A}}) - \rho_{\mathbf{c}_{\text{true}}}(\pi_{\text{E}}) \leq \zeta^{\star} \leq 0$. Intuitively, the cost class $\mathcal{C}$ distinguishes the expert from other policies. The maximization in (1) assigns high total cost to non-expert policies and low total cost to $\pi_{\text{E}}$ [51], while the minimization aims to find the policy that matches the expert as close as possible with respect to $d_{\mathcal{C}}$.

By writing $d_{\mathcal{C}}$ in its *dual* form $\bar{d}_{\mathcal{C}}(\boldsymbol{\mu}_{\pi}, \boldsymbol{\mu}_{\pi_{\text{E}}}) \triangleq \max_{\mathbf{c} \in \mathcal{C}} \left( \langle \boldsymbol{\mu}_{\pi}, \mathbf{c} \rangle - \langle \boldsymbol{\mu}_{\pi_{\text{E}}}, \mathbf{c} \rangle \right)$, it can be interpreted as an *integral probability metric* [80, 60] between the occupancy measures $\boldsymbol{\mu}_{\pi}$ and $\boldsymbol{\mu}_{\pi_{\text{E}}}$. Depending on how $\mathcal{C}$ is chosen, $d_{\mathcal{C}}$ turns to a different metric of probability measures like the 1-Wasserstein distance [117, 32] for $\mathcal{C} = \text{Lip}_1(\mathcal{S} \times \mathcal{A})$, the total variation for $\mathcal{C} = \{\mathbf{c} \mid \|\mathbf{c}\|_{\infty} \leq 1\}$, or the maximum mean discrepancy for $\mathcal{C} = \{\mathbf{c} \mid \|\mathbf{c}\|_{\mathcal{H}} \leq 1\}$, where $\text{Lip}_1(\mathcal{S} \times \mathcal{A})$ denotes the space of 1-Lipschitz functions on $\mathcal{S} \times \mathcal{A}$, and $\|\cdot\|_{\mathcal{H}}$ denotes the norm of a reproducing kernel Hilbert space $\mathcal{H}$ [104].

In our theoretical analysis, we focus on linearly parameterized cost classes [111, 112, 51, 70, 105] of the form $\mathcal{C} \triangleq \{\mathbf{c}_{\mathbf{w}} \triangleq \sum_{i=1}^m w_i \boldsymbol{\phi}_i \mid \mathbf{w} \in \mathcal{W}\}$, where $\{\boldsymbol{\phi}_i\}_{i=1}^m \subset \mathbb{R}_+^{|\mathcal{S}||\mathcal{A}|}$ are fixed feature vectors, such that $\|\boldsymbol{\phi}_i\|_1 \leq 1$ for all $i \in [m]$, and $\mathcal{W}$ is a a convex constraint set for the cost weights $\mathbf{w}$. This

assumption is not necessarily restrictive as usually in practice the true cost depends on just a few key properties, but the desirable weighting that specifies how different desiderata should be traded-off is unknown [4]. Moreover, the cost features can be complex nonlinear functions that can be obtained via unsupervised learning from raw state observations [20, 29]. The matrix $\mathbf{\Phi} \triangleq [\phi_1 \quad \dots \quad \phi_m]$ gives rise a *feature expectation vector* (FEV) $\boldsymbol{\rho_\Phi}(\pi) \triangleq (\rho_{\phi_1}(\pi_E), \dots, \rho_{\phi_m}(\pi_E))^\mathsf{T} \in \mathbb{R}^m$ of a policy $\pi$. Then, by choosing $\mathcal{W}$ to be the $\ell_2$ unit ball $B_1^m \triangleq \{\mathbf{w} \in \mathbb{R}^m \mid \|\mathbf{w}\|_2 \leq 1\}$ [4], we get a *feature expectation matching* objective $d_\mathcal{C}(\pi, \pi_{\pi_E}) = \|\boldsymbol{\rho_\Phi}(\pi) - \boldsymbol{\rho_\Phi}(\pi_E)\|_2$, while for $\mathcal{W}$ being the probability simplex $\Delta_{[m]}$ [111, 112] we have a worst-case excess cost objective $d_\mathcal{C}(\pi, \pi_{\pi_E}) = \max_{i \in [m]} \left( \rho_{\phi_i}(\pi) - \rho_{\phi_i}(\pi_E) \right)$. For clarity, we will replace $\mathbf{c}$ by $\mathbf{w}$ in the notation of the quantities defined in Section 2.1.

## 3 A $Q$-Convex-Analytic Viewpoint

Our methodology builds upon the convex-analytic approach to AL, first introduced by [112], with the key difference that we consider a different convex formulation that introduces $Q$-functions as slack variables. This allows to design a practical scalable model-free algorithm with theoretical guarantees.

Let $\mathfrak{F} \triangleq \{\boldsymbol{\mu} \in \mathbb{R}^{|\mathcal{S}||\mathcal{A}|} \mid (\mathbf{B} - \gamma\mathbf{P})^\mathsf{T}\boldsymbol{\mu} = (1 - \gamma)\boldsymbol{\nu}_0, \ \boldsymbol{\mu} \geq \mathbf{0}\}$ be the *state-action polytope*, where $\mathbf{P}$ is the vector form of $P$, i.e., $P_{(s,a),s'} \triangleq P(s'|s,a)$, and $\mathbf{B}$ is a binary matrix defined by $B_{(s,a),s'} \triangleq 1$ if $s = s'$, and $B_{(s,a),s'} \triangleq 0$ otherwise. The linear constraints that define the set $\mathfrak{F}$, also known as *Bellman flow constraints*, precisely characterize the set of state-action occupancy measures.

**Proposition 1** (94). *We have that $\boldsymbol{\mu} \in \mathfrak{F}$ if and only if there exists a unique stationary Markov policy $\pi$ such that $\boldsymbol{\mu} = \boldsymbol{\mu}_\pi$. If $\boldsymbol{\mu} \in \mathcal{F}$ then the policy $\pi_{\boldsymbol{\mu}}(a|x) \triangleq \frac{\mu(x,a)}{\sum_{a' \in \mathcal{A}} \mu(x,a')}$ has occupancy measure $\boldsymbol{\mu}$.*

Using Proposition 1 and the dual form of the $\mathcal{C}$-distance $\bar{d}_\mathcal{C}(\boldsymbol{\mu}, \boldsymbol{\mu}_{\pi_E}) = \max_{\mathbf{w} \in \mathcal{W}} \langle \boldsymbol{\mu} - \boldsymbol{\mu}_{\pi_E}, \mathbf{c_w} \rangle$, it follows that (1) is equivalent to the primal convex program $\zeta^\star = \min_{\boldsymbol{\mu}} \{\bar{d}_\mathcal{C}(\boldsymbol{\mu}, \boldsymbol{\mu}_{\pi_E}) \mid \boldsymbol{\mu} \in \mathfrak{F}\}$. In particular for $\mathcal{W} = \Delta_{[m]}$ and by using an epigraphic transformation, we end up with an LP program [112], while for $\mathcal{W} = B_1^m$ we get a quadratic objective with linear constraints [4].

A slight variation of the above reasoning is to introduce a mirror variable $\mathbf{d}$ and split the Bellman flow constraints in the definition of $\mathfrak{F}$. We then get the primal convex program

$$\zeta^\star = \min_{(\boldsymbol{\mu}, \mathbf{d})} \{\bar{d}_\mathcal{C}(\boldsymbol{\mu}, \boldsymbol{\mu}_{\pi_E}) \mid (\boldsymbol{\mu}, \mathbf{d}) \in \mathfrak{M}\}, \tag{Primal}$$

where the new polytope is given by $\mathfrak{M} \triangleq \{(\boldsymbol{\mu}, \mathbf{d}) \mid \mathbf{B}^\mathsf{T}\mathbf{d} = \gamma\mathbf{P}^\mathsf{T}\boldsymbol{\mu} + (1 - \gamma)\boldsymbol{\nu}_0, \ \boldsymbol{\mu} = \mathbf{d}, \ \mathbf{d} \geq \mathbf{0}\}$. This overparameterization trick has been first introduced by Mehta and Meyn [76] and has been recently revisited by [14, 84, 67, 83, 77, 71]. A salient feature of this equivalent formulation is that it introduces a $Q$-function as Lagrange multiplier to the equality constraint $\mathbf{d} = \boldsymbol{\mu}$, and so lends itself to data-driven algorithms. To motivate further this new formulation, in Appendix C, we shed light to its dual and provide an interpretation of the dual optimizers. In particular, when $\mathcal{W} = B_1^m$, we show that $(\mathbf{V}^\star_{\mathbf{w}_{\text{true}}}, \mathbf{Q}^\star_{\mathbf{w}_{\text{true}}}, \mathbf{w}_{\text{true}})$ is a dual optimizer.

For our theoretical analysis we focus on the linear MDP setting [55], i.e., we assume that the transition law is linear in the feature mapping. We denote by $\phi(s,a)$ the $(s,a)$-th row of $\mathbf{\Phi}$.

**Assumption 1** (Linear MDP). *There exists a collection of $m$ probability measures $\boldsymbol{\omega} = (\omega_1, \dots, \omega_m)$ on $\mathcal{S}$, such that $P(\cdot|s,a) = \langle \boldsymbol{\omega}(\cdot), \phi(s,a) \rangle$, for all $(s,a)$. Moreover $\phi(s,a) \in \Delta_{[m]}$, for all $(s,a)$.*

Assumption 1 essentialy says that the transition matrix $\mathbf{P}$ has rank at most $m$, and $\mathbf{P} = \mathbf{\Phi}\mathbf{M}$ for some matrix $\mathbf{M} \in \mathbb{R}^{m \times |\mathcal{S}|}$. It is worth noting that in the case of continuous MDPs, despite being linear, the transition law $P(\cdot|s,a)$ can still have infinite degrees of freedom. This is a substantial difference from the recent theoretical works on IL [70, 105] which consider either a linear quadratic regulator, or a transition law that can be completely specified by a finite-dimensional matrix such that the degrees of freedom are bounded.

Assumption 1 enables us to consider a relaxation of (Primal). In particular, we aggregate the constraints $\boldsymbol{\mu} = \mathbf{d}$ by imposing $\mathbf{\Phi}^\mathsf{T}\boldsymbol{\mu} = \mathbf{\Phi}^\mathsf{T}\mathbf{d}$ instead, and introduce a variable $\boldsymbol{\lambda} = \mathbf{\Phi}^\mathsf{T}\boldsymbol{\mu}$. It follows that $\boldsymbol{\lambda}$ lies in the $m$-dimensional simplex $\Delta_{[m]}$. Then, we get the following convex program

$$\zeta^\star = \min_{(\boldsymbol{\lambda}, \mathbf{d})} \{\max_{\mathbf{w} \in \mathcal{W}} \langle \boldsymbol{\lambda}, \mathbf{w} \rangle - \langle \boldsymbol{\mu}_{\pi_E}, \mathbf{c_w} \rangle \mid (\boldsymbol{\lambda}, \mathbf{d}) \in \mathfrak{M}_{\mathbf{\Phi}}\}, \tag{Primal'}$$

where $\mathfrak{M}_{\Phi} \triangleq \{(\boldsymbol{\lambda}, \mathbf{d}) \mid \mathbf{B}^{\mathsf{T}}\mathbf{d} = \gamma\mathbf{M}^{\mathsf{T}}\boldsymbol{\lambda} + (1-\gamma)\boldsymbol{\nu}_0, \; \boldsymbol{\lambda} = \boldsymbol{\Phi}^{\mathsf{T}}\mathbf{d}, \; \boldsymbol{\lambda} \in \Delta_{[m]}, \; \mathbf{d} \in \Delta_{\mathcal{S}\times\mathcal{A}}\}$. As shown in [84, 14, 83], for linear MDPs, the set of occupancy measures $\mathfrak{F}$ can be completely characterized by the set $\mathfrak{M}_{\Phi}$ (c.f., Proposition 2). While the number of constraints and variables in (Primal$'$) is intractable for large scale MDPs, in the next paragraph, we show how this problem can be solved using a proximal point scheme.

## 4  Proximal Point Imitation Learning

By using a Lagrangian decomposition, we have that (Primal$'$) is equivalent to the following bilinear saddle-point problem

$$\min_{\mathbf{x}\in\mathcal{X}} \max_{\mathbf{y}\in\mathcal{Y}} \langle \mathbf{y}, \mathbf{A}\mathbf{x} + \mathbf{b} \rangle, \qquad\qquad \text{(SPP)}$$

where $\mathbf{A} \in \mathbb{R}^{(2m+|\mathcal{S}|)\times(m+|\mathcal{S}||\mathcal{A}|)}$, and $\mathbf{b} \in \mathbb{R}^{(m+|\mathcal{S}|+|\mathcal{S}||\mathcal{A}|)}$ are appropriately defined (see Appendix D), $\mathbf{x} \triangleq [\boldsymbol{\lambda}^{\mathsf{T}}, \mathbf{d}^{\mathsf{T}}]^{\mathsf{T}}$, $\mathbf{y} \triangleq [\mathbf{w}^{\mathsf{T}}, \mathbf{V}^{\mathsf{T}}, \boldsymbol{\theta}^{\mathsf{T}}]^{\mathsf{T}}$, $\mathcal{X} \triangleq \Delta_{[m]} \times \Delta_{\mathcal{S}\times\mathcal{A}}$, and $\mathcal{Y} \triangleq \mathcal{W} \times \mathbb{R}^{|\mathcal{S}|} \times \mathbb{R}^m$.

Since in practice we do not have access to the whole policy $\pi_{\mathrm{E}}$, but instead can observe a finite set of i.i.d. sample trajectories $\mathcal{D}_{\mathrm{E}} \triangleq \{(x_0^{(l)}, a_0^{(l)}, x_1^{(l)}, a_1^{(l)}, \dots, x_H^{(l)}, a_H^{(l)})\}_{l=1}^{n_{\mathrm{E}}} \sim \pi_{\mathrm{E}}$, we define the vector $\widehat{\mathbf{b}}$ by replacing $\boldsymbol{\rho}_{\boldsymbol{\Phi}}(\pi_{\mathrm{E}})$ with its empirical counterpart $\boldsymbol{\rho}_{\boldsymbol{\Phi}}(\widehat{\pi_{\mathrm{E}}})$ (by taking sample averages) in the definition of $\mathbf{b}$. We then consider the empirical objective $f(\mathbf{x}) \triangleq \max_{\mathbf{y}\in\mathcal{Y}} \langle \mathbf{y}, \mathbf{A}\mathbf{x} + \widehat{\mathbf{b}} \rangle$ and apply PPM on the decision variable $\mathbf{x}$. For the $\boldsymbol{\lambda}$-variable we use the relative entropy $D(\boldsymbol{\lambda}||\boldsymbol{\lambda}') \triangleq \sum_{i=1}^m \lambda(i)\log\frac{\lambda(i)}{\lambda'(i)}$, while for the occupancy measure $\mathbf{d}$ we use the conditional relative entropy $H(\mathbf{d}||\mathbf{d}') \triangleq \sum_{s,a} d(s,a)\log\frac{\pi_{\mathbf{d}}(a|s)}{\pi_{\mathbf{d}'}(a|s)}$. With this choice we can rewrite the PPM update as

$$(\boldsymbol{\lambda}_{k+1}, \mathbf{d}_{k+1}) = \operatorname*{arg\,min}_{\boldsymbol{\lambda}\in\Delta_{[m]}, \mathbf{d}\in\Delta_{\mathcal{S}\times\mathcal{A}}} \max_{\mathbf{y}\in\mathcal{Y}} \left\langle \mathbf{y}, \mathbf{A}\begin{bmatrix}\boldsymbol{\lambda}\\\mathbf{d}\end{bmatrix} + \widehat{\mathbf{b}} \right\rangle + \frac{1}{\eta}D(\boldsymbol{\lambda}||\boldsymbol{\Phi}^{\mathsf{T}}\mathbf{d}_k) + \frac{1}{\alpha}H(\mathbf{d}||\mathbf{d}_k), \quad (2)$$

where we used primal feasibility to replace $\boldsymbol{\lambda}_k$ with $\boldsymbol{\Phi}^{\mathsf{T}}\mathbf{d}_k$ as the center point of the relative entropy. PPM is implicit, meaning that it requires the evaluation of the gradient at the next iterate $\mathbf{x}_{k+1}$. Such a requirement makes it not implementable in general. However, in the following, we describe a procedure to apply proximal point to our specific $f(\mathbf{x})$. The following Proposition summarizes the result.

**Proposition 2.** *For a parameter $\boldsymbol{\theta} \in \mathbb{R}^m$, we define the logistic state-action value function $\mathbf{Q}_{\boldsymbol{\theta}} \in \mathbb{R}^{|\mathcal{S}||\mathcal{A}|}$ by $\mathbf{Q}_{\boldsymbol{\theta}} \triangleq \boldsymbol{\Phi}\boldsymbol{\theta}$, and the $k$-step logistic state value function $\mathbf{V}_{\boldsymbol{\theta}}^k \in \mathbb{R}^{|\mathcal{S}|}$ by*

$$V_{\boldsymbol{\theta}}^k(s) \triangleq -\frac{1}{\alpha}\log\left(\sum_a \pi_{\mathbf{d}_{k-1}}(a|s)e^{-\alpha Q_{\boldsymbol{\theta}}(s,a)}\right).$$

*Moreover, we define the $k$-step reduced Bellman error function $\boldsymbol{\delta}_{\mathbf{w},\boldsymbol{\theta}}^k \in \mathbb{R}^m$ by $\boldsymbol{\delta}_{\mathbf{w},\boldsymbol{\theta}}^k \triangleq \mathbf{w} + \gamma\mathbf{M}\mathbf{V}_{\boldsymbol{\theta}}^k - \boldsymbol{\theta}$. Then, the PPM update $(\boldsymbol{\lambda}_k^\star, \mathbf{d}_k^\star)$ in 2 is given by*

$$\lambda_k^\star(i) \propto (\boldsymbol{\Phi}^{\mathsf{T}}\mathbf{d}_{k-1})(i)\,e^{-\eta\delta_{\mathbf{w}_k^\star,\boldsymbol{\theta}_k^\star}^k(i)}, \qquad\qquad (3)$$

$$\pi_{\mathbf{d}_k^\star}(a|s) \propto \pi_{\mathbf{d}_{k-1}}(a|s)\,e^{-\alpha Q_{\boldsymbol{\theta}_k^\star}(s,a)}, \qquad\qquad (4)$$

*where $(\mathbf{w}_k^\star, \boldsymbol{\theta}_k^\star)$ is the maximizer over $\mathcal{W}\times\mathbb{R}^m$ of the $k$-step logistic policy evaluation objective*

$$\mathcal{G}_k(\mathbf{w}, \boldsymbol{\theta}) \triangleq -\frac{1}{\eta}\log\sum_{i=1}^m (\boldsymbol{\Phi}^{\mathsf{T}}\mathbf{d}_{k-1})(i)e^{-\eta\delta_{\mathbf{w},\boldsymbol{\theta}}^k(i)} + (1-\gamma)\langle\boldsymbol{\nu}_0, \mathbf{V}_{\boldsymbol{\theta}}^k\rangle - \langle\boldsymbol{\rho}_{\boldsymbol{\Phi}}(\widehat{\pi_{\mathrm{E}}}), \mathbf{w}\rangle. \quad (5)$$

*Moreover, it holds that $\mathcal{G}_k(\mathbf{w}_k^\star, \boldsymbol{\theta}_k^\star) = \langle\boldsymbol{\lambda}_k^\star, \mathbf{w}_k^\star\rangle - \langle\boldsymbol{\rho}_{\boldsymbol{\Phi}}(\widehat{\pi_{\mathrm{E}}}), \mathbf{w}_k^\star\rangle + \frac{1}{\eta}D(\boldsymbol{\lambda}_k^\star||\boldsymbol{\Phi}^{\mathsf{T}}\boldsymbol{\lambda}_{k-1}) + \frac{1}{\alpha}H(\mathbf{d}_k^\star||\mathbf{d}_{k-1})$. If in addition Assumption 1 holds, then $\mathbf{d}_k^\star$ is a valid occupancy measure, i.e., $\mathbf{d}_k^\star \in \mathfrak{F}$ and so $\mathbf{d}_k^\star = \boldsymbol{\mu}_{\pi_{\mathbf{d}_k^\star}}$.*

The proof of Proposition 2 is broken down into a sequence of lemmas and is presented in Appendix E. It employs an `analytical-oracle` $\mathbf{g}: \mathcal{Y} \to \mathcal{X}$ given by

$$\mathbf{g}(\mathbf{y}; \mathbf{x}_k) \triangleq \operatorname*{arg\,min}_{\boldsymbol{\lambda}\in\Delta_{[m]}, \mathbf{d}\in\Delta_{\mathcal{S}\times\mathcal{A}}} \left\langle \mathbf{y}, \mathbf{A}\begin{bmatrix}\boldsymbol{\lambda}\\\mathbf{d}\end{bmatrix} + \widehat{\mathbf{b}} \right\rangle + \frac{1}{\eta}D(\boldsymbol{\lambda}||\boldsymbol{\Phi}^{\mathsf{T}}\mathbf{d}_k) + \frac{1}{\alpha}H(\mathbf{d}||\mathbf{d}_k),$$

and a `max-oracle` $\mathbf{h} : \mathcal{X} \to \mathcal{Y}$ given by $\mathbf{h}(\mathbf{x}) \triangleq \arg\max_{\mathbf{y} \in \mathcal{Y}} \langle \mathbf{y}, \mathbf{Ag}(\mathbf{y}; \mathbf{x}) \rangle + \frac{1}{\tau} D_\Omega(\mathbf{g}(\mathbf{y}; \mathbf{x}) || \mathbf{x})$, where we used $D_\Omega$ to compact the two divergences. By noting that the PPM update Equation (2) can be rewritten as $\mathbf{x}_{k+1} = \mathbf{g}(\mathbf{h}(\mathbf{x}_k); \mathbf{x}_k)$, its analytical computation is reduced to the characterization of the two aforementioned oracles. In particular, the updates (3)–(4) come from the `analytical-oracle` while (5) is the objective of the `max-oracle`.

The choice of conditional entropy as Bregman divergence for the $\boldsymbol{\lambda}$ variable living in the probability simplex is standard in the optimization literature and is known to mitigate the effect of dimension. In particular, as noted in [85], the classic REPS algorithm [90] can be seen as mirror descent with relative entropy regularization. On the other hand, the choice of conditional entropy as Bregman divergence for the $\mathbf{d}$ variable is less standard and has been popularized by Q-REPS [14]. Such particular divergence leads to an actor-critic algorithm that comes with several merits. By Proposition 2, it is apparent that we get analytical softmin updates for the policy $\pi_{\mathbf{d}}$ rather than the occupancy measure $\mathbf{d}$. Moreover, these softmin updates are expressed in terms of the logistic $Q$-function and do not involve the unknown transition matrix $\mathbf{P}$. Consequently, we avoid the problematic occupancy measure approximation and the restrictive coherence assumption on the choice of features needed in [13, 58], as well as the biased policy updates appearing in REPS [90, 89]. In addition, the newly introduced logistic policy evaluation objective $\mathcal{G}_k(\mathbf{w}, \boldsymbol{\theta})$ has several desired properties. It is concave and smooth in $(\mathbf{w}, \boldsymbol{\theta})$ and has bounded gradients. Therefore, it does not suffer from the pathologies of the squared Bellman error [78] and does not require heuristic gradient clipping techniques. Moreover, unlike [58] it allows a model-free implementation without the need for a generative model (see Section 4.1)

We stress the fact that the `max-oracle` of our proximal point scheme performs the cost update and policy evaluation phases jointly. This is a rather novel feature of our algorithm that differs from the separate cost update and policy evaluation step used in recent theoretical imitation learning works [122, 105, 70]. Our joint optimization over cost and $Q$-functions avoids instability due to adversarial training and can also recover an explicit cost along with the $Q$-function without requiring knowledge or additional interaction with the environment (see Section 5). It is worth noting that application of primal-dual mirror descent to (SPP) does not have this favorable property. While in the standard MDP setting, proximal point and mirror descent coincide because of the linear objective, in imitation learning proximal point optimization makes a difference. In Appendix K, we include a more detailed discussion and numerical comparison between PPM and mirror descent updates.

## 4.1 Practical Implementation

Exact optimization of the logistic policy evaluation objective is infeasible in practical scenarios, due to unknown dynamics and limited computation power. In this section, we design a practical algorithm that uses only sample transitions by obtaining stochastic (albeit biased) gradient estimators.

Proposition 2 gives rise to Proximal Point Imitation Learning ($\mathsf{P}^2\mathsf{IL}$), a model-free actor-critic IRL algorithm described in Algorithm 1. The key feature of $\mathsf{P}^2\mathsf{IL}$ is that the policy evaluation step involves optimization of a single smooth and concave objective over both cost and state-action value function parameters. In this way, we avoid instability or poor convergence in optimization due to nested policy evaluation and cost updates, as well as the undesirable properties of the widely used squared Bellman error. In particular, the $k$th iteration of $\mathsf{P}^2\mathsf{IL}$ consists of the following two steps : (i) (**Critic Step**) Computation of an approximate maximizer $(\mathbf{w}_k, \boldsymbol{\theta}_k) \approx \arg\max_{\mathbf{w},\boldsymbol{\theta}} \mathcal{G}_k(\mathbf{w}, \boldsymbol{\theta})$ of the concave logistic policy evaluation objective, by using a biased stochastic gradient ascent subroutine; (ii) (**Actor Step**) Soft-min policy update $\pi_k(a|s) \propto \pi_{k-1}(a|s) e^{-\alpha Q_{\boldsymbol{\theta}_k}(s,a)}$ expressed in terms of the logistic $Q$-function.

The domain $\Theta$ in Algorithm 1 is the $\ell_\infty$-ball with appropriately chosen radius $D$ to be specified later (see Proposition 3). Moreover, $\Pi_\Theta(\mathbf{x}) \triangleq \arg\min_{\mathbf{y} \in \Theta} \|\mathbf{x} - \mathbf{y}\|_2$ (resp. $\Pi_{\mathcal{W}}(\mathbf{w})$) denotes the Euclidean projection of $\mathbf{x}$ (resp. $\mathbf{w}$) onto $\Theta$ (resp. $\mathcal{W}$).

In order to estimate the gradients $\nabla_{\boldsymbol{\theta}} \mathcal{G}_k(\mathbf{w}, \boldsymbol{\theta})$ and $\nabla_{\mathbf{w}} \mathcal{G}_k(\mathbf{w}, \boldsymbol{\theta})$ we invoke the Biased Stochastic Gradient Estimator subroutine (BSGE) (Algorithm 2) given in Appendix H. By using the linear MDP Assumption 1 and leveraging ridge regression and plug-in estimators, the proposed stochastic gradients can be computed via simple linear algebra with computational complexity $\text{poly}(m, n(t))$, independent of the size of the state space.

---

**Algorithm 1** Proximal Point Imitation Learning: $\text{P}^2\text{IL}(\mathbf{\Phi}, \mathcal{D}_\text{E}, K, \eta, \alpha)$

---

**Input:** Feature matrix $\mathbf{\Phi}$, expert demonstrations $\mathcal{D}_\text{E}$, number of iterations $K$, step sizes $\eta$ and $\alpha$, number of SGD iterations T, SGD learning rates $\boldsymbol{\beta} = \{\beta_t\}_{t=0}^{T-1}$, number-of-samples function $n : \mathbb{N} \to \mathbb{N}$

Initialize $\pi_0$ as uniform distribution over $\mathcal{A}$

Compute the empirical FEV $\boldsymbol{\rho_\Phi}(\widehat{\pi}_\text{E})$ using expert demonstrations $\mathcal{D}_\text{E}$

**for** $k = 1, \ldots K$ **do**
  // Critic-step (policy evaluation)
  Initialize $\boldsymbol{\theta}_{k,0} = \mathbf{0}$ and $\mathbf{w}_{k,0} = \mathbf{0}$

  Run $\pi_{k-1}$ and collect i.i.d. samples $\mathcal{B}_k = \{(s_{k-1}^{(n)}, a_{k-1}^{(n)}, s_{k-1}'^{(n)})\}_{n=1}^{n(T)}$ such that
  $(s_{k-1}^{(n)}, a_{k-1}^{(n)}) \sim \boldsymbol{\mu}_{\pi_{k-1}}$ and $s_{k-1}'^{(n)} \sim \mathsf{P}(\cdot|s_{k-1}^{(n)}, a_{k-1}^{(n)})$
  **for** $t = 0, \ldots T - 1$ **do**
    Compute biased stochastic gradient estimators

$$\left(\widehat{\nabla}_\mathbf{w}\mathcal{G}_k(\mathbf{w}_{k,t}, \boldsymbol{\theta}_{k,t}), \widehat{\nabla}_{\boldsymbol{\theta}}\mathcal{G}_k(\mathbf{w}_{k,t}, \boldsymbol{\theta}_{k,t})\right) = \text{BSGE}\left(k, \mathbf{w}_{k,t}, \boldsymbol{\theta}_{k,t}, n(t)\right)$$

$$\mathbf{w}_{k,t+1} = \Pi_\mathcal{W}\left(\mathbf{w}_{k,t} + \beta_t \widehat{\nabla}_\mathbf{w}\mathcal{G}_k(\mathbf{w}_{k,t}, \boldsymbol{\theta}_{k,t})\right)$$

$$\boldsymbol{\theta}_{k,t+1} = \Pi_\Theta\left(\boldsymbol{\theta}_{k,t} + \beta_t \widehat{\nabla}_{\boldsymbol{\theta}}\mathcal{G}_k(\mathbf{w}_{k,t}, \boldsymbol{\theta}_{k,t})\right)$$

  **end for**
  $(\mathbf{w}_k, \boldsymbol{\theta}_k) = (\frac{1}{T}\sum_{t=1}^{T} \mathbf{w}_{k,t}, \frac{1}{T}\sum_{t=1}^{T} \boldsymbol{\theta}_{k,t})$
  // Actor-step (policy update)
  Policy update: $\pi_k(a|s) \propto \pi_{k-1}(a|s)\, e^{-\alpha Q_{\boldsymbol{\theta}_k}(s,a)}$
**end for**
**Output:** Mixed policy $\widehat{\pi}_K$ of $\{\pi_k\}_{k\in[K]}$

---

### 4.2 Theoretical Analysis

The first step in our theoretical analysis is to study the propagation of optimization errors made by the algorithm on the true policy evaluation objective. In particular at each iteration step $k$, the ideal policy evaluation update $(\mathbf{w}_k^\star, \boldsymbol{\theta}_k^\star)$ and the ideal policy update $\pi_k^\star$ are given by $(\mathbf{w}_k^\star, \boldsymbol{\theta}_k^\star) = \arg\max_{\mathbf{w},\boldsymbol{\theta}} \mathcal{G}_k(\mathbf{w}, \boldsymbol{\theta})$, and $\pi_k^\star(a|s) = \pi_{k-1}(a|s)e^{-\alpha(Q_{\boldsymbol{\theta}_k^\star}(s,a) - V_{\boldsymbol{\theta}_k^\star}^k(s))}$. On the other hand, consider the realised policy evaluation update $(\mathbf{w}_k, \boldsymbol{\theta}_k)$ such that $\mathcal{G}_k(\mathbf{w}_k^\star, \boldsymbol{\theta}_k^\star) - \mathcal{G}_k(\mathbf{w}_k, \boldsymbol{\theta}_k) = \epsilon_k$, the corresponding policy $\pi_k$ given by $\pi_k = \pi_{k-1}(a|s)e^{-\alpha(Q_{\boldsymbol{\theta}_k}(s,a) - V_{\boldsymbol{\theta}_k}^k(s))}$, and let $\mathbf{d}_k \triangleq \boldsymbol{\mu}_{\pi_k}$. We denote by $\widehat{\pi}_K$ the extracted mixed policy of $\{\pi_k\}_{k=1}^K$. We are interested in upper-bounding the suboptimality gap $d_\mathcal{C}(\widehat{\pi}_K, \pi_\text{E})$ of Algorithm 1 as a function of $\varepsilon_k$. To this end, we need the following assumption.

**Assumption 2.** *It holds that $\lambda_{\min}(\mathbb{E}_{(s,a)\sim\mathbf{d}_k} \phi(s,a)\phi(s,a)^\mathsf{T}) \geq \beta$, for all $k \in [K]$.*

Assumption 2 states that every occupancy measure $\mathbf{d}_k$ induces a positive definite feature covariance matrix, and so every policy $\pi_k$ explores uniformly well in the feature space. This assumption is common in the RL theory literature [2, 46, 37, 66, 3, 7]. It is also related to the condition of persistent excitation from the control literature [81].

The following proposition ensures that $\max_{\mathbf{w},\boldsymbol{\theta}\in\mathcal{W}\times\mathbb{R}^m} \mathcal{G}_k(\mathbf{w}, \boldsymbol{\theta}) = \max_{\mathbf{w},\boldsymbol{\theta}\in\mathcal{W}\times\Theta} \mathcal{G}_k(\mathbf{w}, \boldsymbol{\theta})$. Therefore, this constraint does not change the problem optimality, but will considerably accelerate the convergence of the algorithm by considering smaller domains.

**Proposition 3.** *There exists a maximizer $\boldsymbol{\theta}_k^\star$ such that $\|\boldsymbol{\theta}_k^\star\|_\infty \leq \frac{1+|\log\beta|}{1-\gamma} \triangleq D$.*

We can now state our error propagation theorem.

**Theorem 1.** *Let $\widehat{\pi}_K$ be the output of running Algorithm 1 for $K$ iterations, with $n_\text{E} \geq \frac{2\log(\frac{2m}{\delta})}{\varepsilon^2}$ expert trajectories of length $H \geq \frac{1}{1-\gamma}\log(\frac{1}{\varepsilon})$. Let $C \triangleq \frac{1}{\beta\eta}\left(\sqrt{\frac{2\alpha}{1-\gamma}} + \sqrt{8\eta}\right) + \sqrt{\frac{18\alpha}{1-\gamma}}$. Then, with probability at least $1 - \delta$, it holds that $d_\mathcal{C}(\widehat{\pi}_K, \pi_\text{E}) \leq \frac{1}{K}\left(\frac{D(\boldsymbol{\lambda}^*\|\mathbf{\Phi}^\mathsf{T}\mathbf{d}_0)}{\eta} + \frac{H(\mathbf{d}^*\|\mathbf{d}_0)}{\alpha} + C\sum_k \sqrt{\epsilon_k} + \sum_k \epsilon_k\right) + \varepsilon.$*

By Theorem 1, whenever the policy evaluation errors $\varepsilon_k$, as well as the estimation error $\varepsilon$ can be kept small, Algorithm 1 ouputs a policy $\widehat{\pi}_K$ with small suboptimality gap $\rho_{\mathbf{c}_\text{true}}(\widehat{\pi}_K) - \rho_{\mathbf{c}_\text{true}}(\pi_\text{E})$.

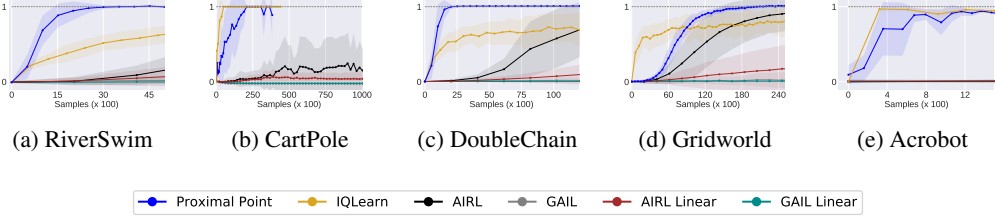

Figure 1: **Online IL Experiments**. We show the total returns vs the number of env steps.

Notably, there is no direct dependence on the size of the state space or the dimension of the feature space. In the ideal case, where $\varepsilon_k = 0$ for all $k$, the convergence rate is $\mathcal{O}(1/K)$. The provided error propagation analysis still holds with general function approximation, i.e., in the context of deep RL. Indeed, by choosing $\boldsymbol{\Phi} = \mathbf{I}$, Assumption 1 is trivially satisfied and the $\boldsymbol{\theta}$ variable in the objective $\mathcal{G}_k$ is replaced by a $Q$-function. In practice, the estimation error $\varepsilon$ can be made arbitrary small, by increasing the number of expert demonstrations $n_E$. Moreover, the next theorem ensures that under Assumptions 1 and 2 the biased stochastic gradient ascent (BSGA) subroutine has sublinear convergence rate.

**Theorem 2.** *Let $(\mathbf{w}_k, \boldsymbol{\theta}_k)$ be the output of the BSGA subroutine in Algorithm 1 for $T$ iterations, with $n(t) \geq \max\left( \mathcal{O}\left( \frac{\gamma^2 m D t}{(\eta + \alpha)^2 \beta} \log \frac{Tm}{\delta} \right), \mathcal{O}\left( \frac{mt}{(\eta + \alpha)^2 \beta} \log \frac{Tm}{\delta} \right) \right)$ sample transitions, and learning rates $\beta_t = \mathcal{O}(\frac{1}{\sqrt{t}})$. Then, $\epsilon_k = \mathcal{G}_k(\mathbf{w}_k^\star, \boldsymbol{\theta}_k^\star) - \mathcal{G}_k(\mathbf{w}_k, \boldsymbol{\theta}_k) \leq \mathcal{O}(\frac{\max\{\eta, 1\} m D}{\beta \sqrt{T}})$, with probability $1 - \delta$.*

**Corollary 1** (Resource guarantees). *Choose $\eta = \alpha = 1$ and let $K = \Omega\left(\epsilon^{-1}\right)$, $T = \Omega\left(\epsilon^{-4}\right)$. Then for $\Omega\left(KT\right) = \Omega\left(\epsilon^{-5}\right)$ sample transitions, $\Omega\left(\varepsilon^{-2}\right)$ expert trajectories and approximately solving $\Omega\left(\epsilon^{-1}\right)$ concave maximization problems, we can ensure $d_{\mathcal{C}}(\widehat{\pi}, \pi_E) \leq \mathcal{O}(\epsilon + \varepsilon)$, with high probability.*

**Offline Setting.** Finally, we notice that using $\boldsymbol{\Phi}^\intercal \boldsymbol{\mu}_{\pi_E}$ as the reference distribution for the relative entropy we can obtain an offline algorithm that does not require environment interactions. By reinterpreting smoothing [82] as one step of proximal point, and using similar arguments as in the proof of Theorem 1, we can provide similar theoretical guarantees for the offline setting. The formal statement of the theoretical result as well as the optimization of the empirical policy evaluation objective are presented in Appendix J (see Theorems 4 and 6).

## 5 Experiments

In this section, we demonstrate that our approach achieves convincing empirical performance in both online and offline IL settings on several environments.[1] The precise setting is detailed in Appendix L.

**Online Setting.** We first present results in various tabular environments where we can implement our algorithm without any practical relaxation outperforming GAIL [51], AIRL [38] and IQ-Learn [40]. Results are given in Figure 1. Good performance but inferior to IQ-Learn is observed also for continuous states environments (CartPole and Acrobot) where we used neural networks function approximation.

**Offline Setting.** Figures 2a to 2c shows that our method is competitive with the state-of-the-art offline IL methods IQLearn [40] and AVRIL [25] that recently showed performances superior to other methods like [54][64]. We also tried our algorithm in the complex image-based `Pong` task from the Atari suite. Figure 2d shows that the algorithm reaches the expert level after observing $2e5$ expert samples. We did not find AVRIL competitive in this setting, and skip it for brevity. In these settings, we verified that the algorithmic performance is convincing even for costs parameterized by neural networks.

**Continuous control experiments.** We attain the expert performance also in 2 MuJoCo environments: `Ant`, `HalfCheetah`, `Hopper`, and `Walker` (see Figures 2e to 2h). The additional difficulty in implementing the algorithm in continuous control experiments is that the analytical form of the policy

---

[1]The code is available at the following link `https://github.com/lviano/P2IL`.

improvement step is no longer computationally tractable because this would require to compute an integral over the continuous action space. Therefore, we approximated this update using the Soft Actor Critic (SAC) [44] algorithm. SAC requires environment samples making the algorithm online. The good empirical result opens the question of analyzing policy improvement errors as in [41].

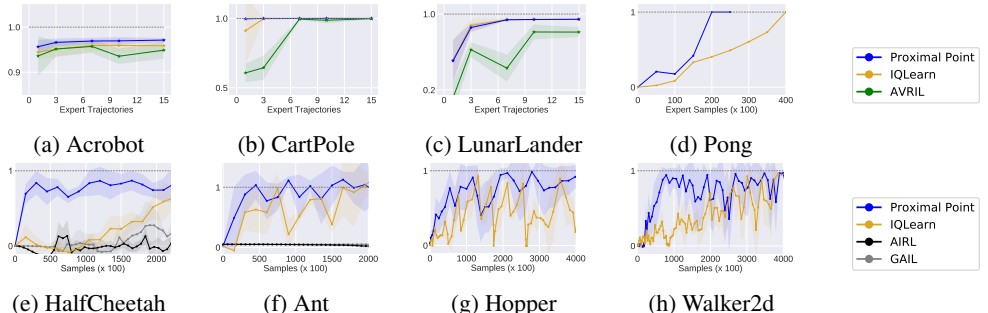

(a) Acrobot    (b) CartPole    (c) LunarLander    (d) Pong

(e) HalfCheetah    (f) Ant    (g) Hopper    (h) Walker2d

Figure 2: **Neural function approximation experiments.** Figures 2a to 2c show the total returns vs the number of expert trajectories. Figures 2e to 2h show the total returns vs the number of env steps. Figure 2d shows the total return vs the number of expert state-action pairs.

**Recovered Costs.** A unique algorithmic feature of the proposed methodology is that we can explicitly recover a cost along with the Q-function without requiring adversarial training. In Figure 3, we visualize our recovered costs in a simple 5x5 `Gridworld`. Most importantly, we verify that the recovered costs induce nearly optimal policies w.r.t. the unknown true cost function. Compared to IQ-Learn [40], we do not require knowledge or further interaction with the environment. Therefore, the recovered cost functions show promising transfer capability to new dynamics.

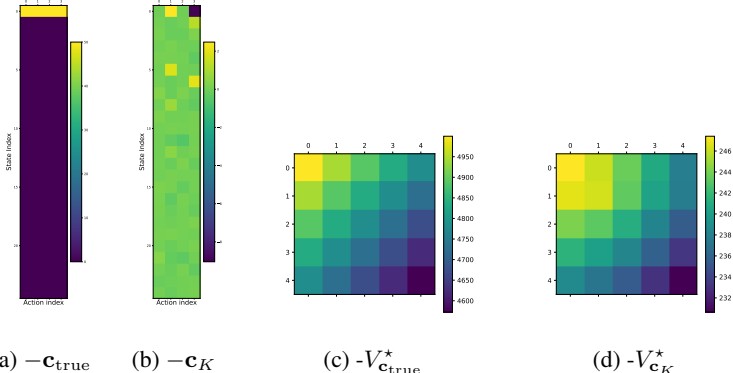

(a) $-\mathbf{c}_{\mathrm{true}}$    (b) $-\mathbf{c}_K$    (c) $-V^{\star}_{\mathbf{c}_{\mathrm{true}}}$    (d) $-V^{\star}_{\mathbf{c}_K}$

Figure 3: **Recovered Costs in `Gridworld`.** Comparison between the true cost $\mathbf{c}_{\mathrm{true}}$ and the cost $\mathbf{c}_K$ recovered by P$^2$IL. We notice that the optimal value functions $V^{\star}_{\mathbf{c}_{\mathrm{true}}}$ and $V^{\star}_{\mathbf{c}_K}$ present the same pattern. Hence, the optimal policy with respect to $\mathbf{c}_K$ is nearly optimal with respect to $\mathbf{c}_{\mathrm{true}}$.

**Cost Transfer Setting.** We experimented with a transfer cost setting on a `Gridworld` (Figure 4). We consider two different Gridworld MDP environments, say $M$ and $\widetilde{M}$, with opposite action effects. This means that action `Down` in $\widetilde{M}$ corresponds to action `Left` in $M$ and vice versa. Similarly, the effects of `Up` and `Right` are swapped between $\widetilde{M}$ and $M$. We denote by $\mathbf{V}^{\pi}_{\widetilde{M},\mathbf{c}_{\mathrm{true}}}$ (resp. $\mathbf{V}^{\star}_{\widetilde{M},\mathbf{c}_{\mathrm{true}}}$) the value function of policy $\pi$ (resp. optimal value function) in the MDP environment $\widetilde{M}$ with cost function $\mathbf{c}_{\mathrm{true}}$. Moreover, we denote by $\pi^{\star}_{M,\mathbf{c}}$ the optimal policy in the MDP environment $M$ under cost function $\mathbf{c}$. Figure (a) gives the corresponding optimal value function. Figure (b) presents the value function of the expert policy $\pi_{\mathrm{E}} = \pi^{\star}_{M,\mathbf{c}_{\mathrm{true}}}$ used as target by P$^2$IL. Figure (d) shows the value function of the learned imitating policy $\pi_K$ from P$^2$IL. Finally, Figure (b) depicts the value function of the optimal policy $\pi^{\star}_{\widetilde{M},\mathbf{c}_K}$ for the environment $\widetilde{M}$ endowed with the recovered cost function $\mathbf{c}_K$ by

P²IL (with access to samples from $M$). We conclude that the policy $\pi^\star_{\widetilde{M},\mathbf{c}_K}$ is optimal in $\widetilde{M}$ with cost $\mathbf{c}_{\text{true}}$. By contrast, the expert policy $\pi_{\text{E}} = \pi^\star_{M,\mathbf{c}_{\text{true}}}$ used as target by P²IL performs poorly and as a consequence also the imitating policy $\pi_K$ does so. All in all, we notice that the recovered cost induces an optimal policy for the new dynamics while the imitating policy fails. Albeit, cost transfer is successful in this experiment we do not expect this fact to be true in general because we do not tackle the issue of cost shaping [87].

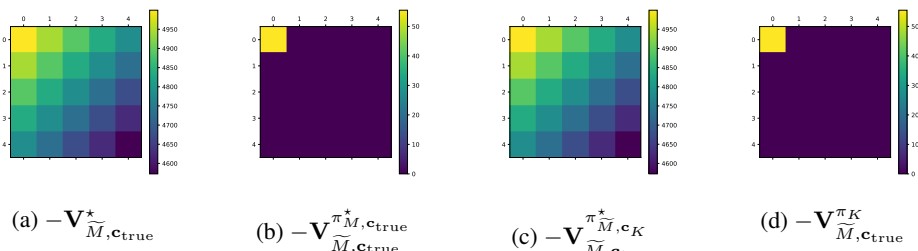

$$(a) -\mathbf{V}^\star_{\widetilde{M},\mathbf{c}_{\text{true}}} \qquad (b) -\mathbf{V}^{\pi^\star_{M,\mathbf{c}_{\text{true}}}}_{\widetilde{M},\mathbf{c}_{\text{true}}} \qquad (c) -\mathbf{V}^{\pi^\star_{\widetilde{M},\mathbf{c}_K}}_{\widetilde{M},\mathbf{c}_{\text{true}}} \qquad (d) -\mathbf{V}^{\pi_K}_{\widetilde{M},\mathbf{c}_{\text{true}}}$$

Figure 4: **Cost Transfer Experiment in** `Gridworld`. We compare the performance of several policies in the new MDP environment $\widetilde{M}$ with cost function $\mathbf{c}_{\text{true}}$. We notice that the recovered cost induces an optimal policy for the new dynamics while the imitating policy fails.

## 6 Discussion and Outlook

In this work, we studied a Proximal Point Imitation Learning (P²IL) algorithm with both theoretical guarantees and convincing empirical performance. Our methodology is rooted in classical optimization tools and the LP approach to MDPs. The most significant merits of P²IL are the following: (i) It optimizes a convex and smooth logistic Bellman evaluation objective over both cost and Q-functions. In particular, it avoids instability due to adversarial training and can also recover an explicit cost along with Q function; (ii) In the context of linear MDPs, it comes with efficient resource guarantees and error bounds for the suboptimality of the learned policy (Theorem 2 and Corollary 1). In particular, given $\text{poly}(1/\varepsilon, \log(1/\delta), m)$ many samples, it recovers an $\varepsilon$-optimal policy, with probability $1 - \delta$. Notably, the bound is independent of the size of the state-action space; (iii) Beyond the linear MDP setting, it can be implemented in a model-free manner, for both online and offline setups, with general function approximation without losing its theoretical specifications. This is justified by providing an error propagation analysis (Theorems 1 and 4), guaranteeing that small optimization errors lead to high-quality output policy; (iv) It enjoys not only strong theoretical guarantees but also favorable empirical performance. At the same time, our newly introduced methods bring challenges and open questions. One interesting question is whether one can accelerate the PPM updates and improve the convergence rate. Another direction for future work is to provide rigorous arguments for the near-optimality of the recovered cost function. On the practical side, we plan to conduct experiments in more challenging environments than MuJoCo and Atari. We hope our new techniques will be useful to future algorithm designers and lay the foundations for overcoming current limitations and challenges. In Appendix B, we point out in detail a few interesting future directions.

## Acknowledgements

The authors would like to thank the anonymous reviewer for their suggestions to improve the presentation and for motivating us to inspect the recovered cost function. This work has received funding from the Enterprise for Society Center (E4S), the European Research Council (ERC) under the European Union's Horizon 2020 research and innovation programme grant agreement OCAL, No. 787845, the European Research Council (ERC) under the European Union's Horizon 2020 research and innovation programme (grant agreement n° 725594 - time-data), the Swiss National Science Foundation (SNSF) under grant number 200021_205011. Gergely Neu was supported by the European Research Council (ERC) under the European Union's Horizon 2020 research and innovation programme (Grant agreement No. 950180). Luca Viano acknowledges travel support from ELISE (GA no 951847).

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
