**Theoretical Imitation Learning.** Our work is related to recent actor-critic IL schemes with theoretical guarantees for different MDP models, and different policy evaluation objectives (e.g., minimizing the squared Bellman error) [21, 122, 26, 70, 105]. Contrary to these actor-critic schemes, in our proximal-point imitation learning algorithm, the policy evaluation step involves optimization of a single objective over both cost and $Q$-functions. In this way, we avoid instability or poor convergence due to nested policy evaluation and cost update steps [40] as well as the undesirable properties of the widely used squared Bellman error [78]. Moreover, for the context of linear MDPs [14, 121, 55, 22, 115, 7, 84], we provide guarantees and convergence rates for the suboptimality of the learned policy, under mild assumptions, significantly weaker than those found in the literature until now. To our knowledge, such guarantees in this setting are provided for the first time. It is worth noting that in the case of continuous MDPs, despite being linear, the transition law can still have infinite degrees of freedom. This is a substantial difference from the recent theoretical works on IL [21, 122, 26, 70, 105] which consider either tabular MDPs [105], or a linear quadratic regulator [21], or a linear transition law that can be completely specified by a finite-dimensional matrix [70]. In the last case, the degrees of freedom are bounded, and thus mitigate the challenges in estimating the transition model. Indeed, the linear MDP setting studied in [70] reduces the unknown dynamics problem to estimating an unknown finite-dimensional matrix, which differs from our nonparametric approach. We also note that [122, 118] require the restrictive assumption of bounded concentrability coefficients, while this is not the case for the analysis in this paper. The convergence and generalization of actor-critic IL schemes for general MDPs has been studied in [28]. However, the authors in [28] only provide local optimality convergence guarantees, i.e., convergence to a stationary point. On the contrary, our algorithm provides global convergence guarantees for the linear MDP setting. Moreover, we account for potential policy evaluation errors , presenting an error propagation analysis that leads to rigorous guarantees for both online and offline setting, beyond the linear MDP assumption. Indeed, it is worth noting that the provided error propagation analysis justifies using our derived actor-critic scheme with general function approximation. A scalable deep reinforcement learning implementation is possible, without losing the theoretical guarantees of Theorem 1. The work [26] studies offline IL for the continuous kernelized nonlinear regulator and Gaussian process setting [56]. We notice that this setting is different from the linear MDP model studied in this paper, and each one does not imply the other. Finally, a recent theoretical IL work that is rooted in the LP approach to MDPs is [58]. The authors consider a Lagrangian reformulation of the problem and design a stochastic primal-dual algorithm with explicit performance bounds on the quality of the extracted policy. The most important limitations of the primal-dual algorithm [58] are (i) the need of a generative oracle, (ii) restricted coherence assumptions on the choice of features, as well as (iii) the problematic occupancy measure approximation. These limitations lead to poor practical performance for challenging high-dimensional and model-free IL setups. On the other hand, our algorithm overcomes these difficulties by applying a proximal point update to an alternative $Q$-LP formulation [77, 83]. This results to a model-free actor-critic scheme with explicit tractable softmax policy updates. Compared with the setting in [58], where access to a generative-model oracle is assumed, we only have the ability to execute learned policies in the underlying MDP to generate trajectories. This assumption is considerably weaker that having a simulator-based MDP, however it is stronger than having "irreversible experience", where the learner must follow a single trajectory without having access to a *reset action*, that obtains a new trajectory from the initial state distribution. Most importantly our algorithm enjoys not only strong theoretical guarantees, but also favorable practical performance.

**Approximate Linear Programming.** There is an emerging body of literature [33, 1, 30, 109, 65, 79, 115, 67, 10, 16, 74, 31, 55, 106, 14] that studies ALP for the forward RL. While this approach dates back to 1960s [73], it has recently witnessed an interesting renaissance for its potential to provide a solid formal framework for newly derived methods, as well as a deeper understanding of existing empirically successful algorithms. In this paper, we present scalable imitation learning algorithms with theoretical guarantees rooted in the LP approach, highlighting how historical key limitations have been eliminated. Prior approximate linear programming (ALP) approaches developed algorithms for solving large-scale and/or continuous MDPs on a low-dimensional subspace by reducing the number of constraints (e.g., by constraint sampling) [33, 34]. However, these prior works either scale badly with the size of the state-action spaces or require access to samples from a distribution

that depends on the optimal policy. Moreover, they focus mainly on the approximation of the optimal value but not so much on extracting a near optimal policy. On the other hand, a recent line of works [30, 67, 115, 55, 106] solve the problem for large-scale MDPs by employing stochastic primal-dual methods, in light of Lagrangian duality. Although this approach achieves state-of-the-art sample complexity guarantees, it shows poor performance in practice. First, current primal-dual algorithms need access to a simulator, mitigating implicitly the problem of exploration, Second, when dealing with linear relaxations of MDPs [14, 58] one needs to impose a restrictive coherence assumptions to ensure that small duality gap for the linearly relaxed LP implies small suboptimality gap for the extracted policy. Finally, while their is enough intuition behind the use of linear function approximation for value functions, this is not the case for occupancy measure approximation. A new breed of algorithms that seem to overcome these difficulties is based on an alternative $Q$-LP formulation of RL. This approach has been first introduced by Mehta and Meyn [76] and has been recently revisited by [14, 84, 67, 83, 77, 71]. A salient feature of this equivalent formulation is that it introduces a $Q$-function as slack variables, and so lends itself to data-driven algorithms. Our work is inspired by these line of works. The most related works are the analysis of REPS/Q-REPS [90, 14, 89] and O-REPS [124] that first pointed out the connection between REPS and PPM. We build on their techniques with some important differences. In particular, while in the LP formulation of RL, PPM and mirror descent [15, 47] are equivalent, recognizing that they are *not equivalent* in IL is critical for stronger empirical performance. Moreover, our techniques can be used to improve upon the best rate for REPS in the tabular setting [89] and to extend their guarantees to Linear MDPs.

**State-of-the-art Imitation Learning.** Generative adversarial imitation learning (GAIL) [51] and other follow-up works [38, 59, 63, 64] formulate the IL as a minimax adversarial problem similar to a GAN [43] and leverage primal-dual optimization tools. In particular, GAIL solves IL with alternating updates of both policy and cost functions. On the other hand, a recent line of work [40, 11, 96] bypasses the need of optimizing over cost functions and thus avoids instability due to adversarial training. Although these algorithms achieve impressive empirical performance in challenging high dimensional benchmark tasks, they are hampered by limited theoretical understanding. This is the fundamental difference from our work, which enjoys both favorable practical performance and strong theoretical guarantees. Moreover, a unique algorithmic feature of our proposed methodology is a convex and smooth logistic policy evaluation objective that optimizes jointly cost and $Q$-functions. As a result, our algorithm has the additional practical benefit that can also recover an explicit cost along with the Q-function without requiring knowledge or further interaction with the environment (as in [40, 11, 96]). Therefore, the recovered cost functions show promising transfer capability to new dynamics. In addition, unlike IQ-Learn [40], in our online IL algorithm, instead of regularizing the IL objective, the key idea is to penalize the divergence between the current policy and the policy obtained at the previous iteration. We do so by employing a Bregman proximal point update. Most importantly, as we have already highlighted, the convergence properties of [40, 11, 96]) remain largely elusive in the function approximation and model-free regime. It is unclear whether the sampling-based variants of their algorithms converge to a global optimum or if they converge at all, even for the simple tabular setting.

# B  Future directions

In this work, we studied a proximal point imitation learning algorithm with both theoretical guarantees and convincing empirical performance in challenging benchmark tasks. Our methodology is rooted in classical stochastic optimization tools and in the LP approach to MDPs. We hope that our new techniques will be useful for future algorithm designers and lay foundations for overcoming current limitations and challenges. We point out a few interesting directions.

**Accelerated proximal point.** An appealing possibility is to study an accelerated proximal point scheme with inexact updates and achieve faster convergence rates. While there has been an effort in this direction [119, 120], the acceleration relies on the triangle/quadrangle scaling property assumption [45] that does not hold for KL divergence over the simplex. Understanding if it is possible to accelerate PPM without such an assumption is an open question, whose solution has direct application to the LP formulation of RL and imitation learning.

**Primal-dual methods with conditional relative entropy.** Recent primal-dual RL algorithms rooted in the LP approach to MDPs achieve state-of-the-art sample complexity guarantees. See, for example, [13] for exact gradients, [23, 55] for stochastic gradients, and [58] for the imitation

learning problem. The most important disadvantages of primal-dual RL algorithms are (i) the need of a generative oracle, (ii) restricted coherence assumptions on the choice of features, as well as (iii) the problematic occupancy measure approximation. Unfortunately, these limitations lead to poor practical performance for challenging high-dimensional and model-free RL and IL setups. On the other hand, our algorithm overcomes these difficulties but requires to approximately solve a small-dimensional convex program repetitively. It is also challenging to account for the biased gradient estimates beyond the linear MDP setting. It is promising to investigate if by combining the alternative $Q$-LP formulation and the conditional relative entropy as Bregman divergence in a primal-dual mirror descent scheme, one can avoid the current practical limitations of primal-dual RL methods. It is also interesting that in this case, the action-value parameters will be updated by taking one gradient step each time, instead of solving a small-dimensional convex program.

**Inexact policy improvement update.** The error propagation presented in this work accommodates for errors only in the policy evaluation phase, while it assumes that the policy improvement step can be implemented exactly. This happens in other related works like [14, 114]. In contrast, the error propagation analysis in [41] takes into account an error in the policy improvement step but unfortunately it does not provide a way to ensure that such an error is small. Future research effort will aim to include in our error propagation analysis a term given by inexact policy improvement steps, ensure that such errors are small and characterizing the deterioration in the sample complexity under policy improvement errors. This kind of analysis would be important for continuous actions environment where the softmax policy update can not be computed in closed-form.

## C  Dual Program Interpretation

To motivate further the Primal formulation, we shed light to its dual and provide an interpretation of the dual optimizers. For brevity, we focus on the case $\mathcal{W} = B_1^m$. The proof can be found in Appendix C.1 and is based on strong duality between the two convex programs.

**Proposition 4.** *The dual convex program is given by*

$$\zeta^\star = \max_{(\mathbf{w}, \mathbf{V}, \mathbf{Q})} \left\{ (1 - \gamma) \langle \boldsymbol{\nu}_0, \mathbf{V} \rangle - \langle \boldsymbol{\mu}_{\pi_{\mathrm{E}}}, \mathbf{c}_{\mathbf{w}} \rangle \mid \mathbf{Q} \geq \mathbf{B}\mathbf{V}, \ \mathbf{Q} = \mathbf{c}_{\mathbf{w}} + \gamma \mathbf{P}\mathbf{V}, \right.$$
$$\left. \mathbf{V} \in \mathbb{R}^{|\mathcal{S}|}, \ \mathbf{Q} \in \mathbb{R}^{|\mathcal{S}||\mathcal{A}|}, \ \mathbf{w} \in \mathcal{W} \right\}. \quad \text{(Dual)}$$

*Moreover, for $\mathcal{W} = B_1^m$, a triple $(\mathbf{V}_{\mathrm{A}}, \mathbf{Q}_{\mathrm{A}}, \mathbf{w}_{\mathrm{A}})$ is dual optimal if and only if (i) $\pi_{\mathrm{E}}$ is optimal for the RL problem with cost $\mathbf{c} = \mathbf{c}_{\mathbf{w}_{\mathrm{A}}}$, (ii) $\mathbf{V}_{\mathrm{A}} = \mathbf{V}_{\mathbf{w}_{\mathrm{A}}}^\star$, (iii) $\mathbf{Q}_{\mathrm{A}} = \mathbf{Q}_{\mathbf{w}_{\mathrm{A}}}^\star$, and (iv) $\mathbf{w}_{\mathrm{A}} \in \mathcal{W}$. In particular, $(\mathbf{V}_{\mathbf{w}_{\mathrm{true}}}^\star, \mathbf{Q}_{\mathbf{w}_{\mathrm{true}}}^\star, \mathbf{w}_{\mathrm{true}})$ is a dual optimizer.*

Proposition 4 states that the set of dual optimal costs $\mathbf{c}_{\mathbf{w}_{\mathrm{A}}}$ is the set of costs in $\mathcal{C}$ for which the expert is optimal. In this case, the optimal $\mathbf{V}_{\mathrm{A}}$ coincides with the corresponding optimal value function[2], while the optimal $\mathbf{Q}_{\mathrm{A}}$ coincides with the corresponding optimal state-action value function. In particular, the true weights $\mathbf{w}_{\mathrm{true}}$, the true optimal value function $\mathbf{V}_{\mathbf{w}_{\mathrm{true}}}^\star$ and the true optimal state-action value function $\mathbf{Q}_{\mathbf{w}_{\mathrm{true}}}^\star$ are dual optimizers. Therefore, the presented $Q$-convex approach allows to recover an optimal solution to the original problem (1) from both the (Primal) and (Dual) formulations: it can be obtained either as the induced policy of a primal optimal occupancy measure or as a greedy policy associated to a dual optimal $Q$-function. In Section 4, we generalize the later observation to implement PPM using softmin updates in terms of $Q$-functions.

### C.1  Proof of Proposition 4

We recall the alternative $Q$-LP approach to MDPs [76, 77, 84, 83, 14]. Let $\mathbf{c} \in \mathbb{R}^{|\mathcal{S}||\mathcal{A}|}$ be a cost function. The forward RL problem is equivalent to the following linear programs[3]

$$\rho_{\mathbf{c}}^\star = \min_{(\boldsymbol{\mu}, \mathbf{d}) \in \mathbb{R}^{2|\mathcal{S}||\mathcal{A}|}} \left\{ \langle \boldsymbol{\mu}, \mathbf{c} \rangle \mid \mathbf{B}^{\mathsf{T}} \mathbf{d} = \gamma \mathbf{P}^{\mathsf{T}} \boldsymbol{\mu} + (1 - \gamma) \boldsymbol{\nu}_0, \ \mathbf{d} = \boldsymbol{\mu}, \ \mathbf{d} \geq \mathbf{0} \right\} \quad \text{(Primal $Q$-LP)}$$

$$= \max_{(\mathbf{V}, \mathbf{Q}) \in \mathbb{R}^{|\mathcal{S}| + |\mathcal{S}||\mathcal{A}|}} \left\{ (1 - \gamma) \langle \boldsymbol{\nu}_0, \mathbf{u} \rangle \mid \mathbf{Q} \geq \mathbf{B}\mathbf{V}, \ \mathbf{Q} = \mathbf{c} + \gamma \mathbf{P}\mathbf{V}, \ \mathbf{V} \in \mathbb{R}^{|\mathcal{S}|}, \right\}, \quad \text{(Dual $Q$-LP)}$$

---

[2]To be precise, this is the case if $\boldsymbol{\nu}_0 \in \mathbb{R}_{++}^{|\mathcal{S}|}$, otherwise they coincide $\boldsymbol{\nu}_0$-almost surely.

[3]Note that usually in the literature the primal LP is (Dual $Q$-LP).

We have that if $\pi^\star$ is an optimal policy for the forward RL problem with cost $\mathbf{c}$, then $(\boldsymbol{\mu}_{\pi^\star}, \boldsymbol{\mu}_{\pi^\star})$ is optimal for (Primal $Q$-LP) and conversely if $(\boldsymbol{\mu}^\star, \mathbf{d}^\star)$ is optimal for (Primal $Q$-LP), then $\pi_{\boldsymbol{\mu}^\star}$ is an optimal policy for the forward RL problem with cost $\mathbf{c}$. Moreover, $(\mathbf{V}_\mathbf{c}^\star, \mathbf{Q}_\mathbf{c}^\star)$ is an optimal solution to (Dual $Q$-LP) and it is the unique optimizer when $\boldsymbol{\nu}_0 \in \mathbb{R}_{++}^{|\mathcal{S}|}$. For the following results, we will assume without loss of generality that $\boldsymbol{\nu}_0 \in \mathbb{R}_{++}^{|\mathcal{S}|}$.

*Proof of Proposition 4.* We first derive the dual convex program. We have,

$$
\begin{aligned}
\zeta^\star &= \min_{(\boldsymbol{\mu},\mathbf{d})\in\mathfrak{M}} \max_{\mathbf{w}\in\mathcal{W}} \left\langle \boldsymbol{\mu} - \boldsymbol{\mu}_{\pi_\mathrm{E}}, \mathbf{c}_\mathbf{w} \right\rangle \\
&= \max_{\mathbf{w}\in\mathcal{W}} \min_{(\boldsymbol{\mu},\mathbf{d})\in\mathfrak{M}} \left\langle \boldsymbol{\mu} - \boldsymbol{\mu}_{\pi_\mathrm{E}}, \mathbf{c}_\mathbf{w} \right\rangle \\
&= \max_{\mathbf{w}\in\mathcal{W}} \min_{\boldsymbol{\mu},\mathbf{d}\geq 0} \max_{\mathbf{V},\mathbf{Q}} \{ \left\langle \boldsymbol{\mu} - \boldsymbol{\mu}_{\pi_\mathrm{E}}, \mathbf{c}_\mathbf{w} \right\rangle + \left\langle \gamma\mathbf{P}^\mathsf{T}\boldsymbol{\mu} + (1-\gamma)\boldsymbol{\nu}_0 - \mathbf{B}^\mathsf{T}\mathbf{d}, \mathbf{V} \right\rangle + \left\langle \mathbf{d} - \boldsymbol{\mu}, \mathbf{Q} \right\rangle \} \\
&= \max_{\mathbf{w}\in\mathcal{W}} \min_{\boldsymbol{\mu},\mathbf{d}\geq 0} \max_{\mathbf{V},\mathbf{Q}} \{ (1-\gamma) \left\langle \boldsymbol{\nu}_0, \mathbf{V} \right\rangle - \left\langle \boldsymbol{\mu}_{\pi_\mathrm{E}}, \mathbf{c}_\mathbf{w} \right\rangle + \left\langle \boldsymbol{\mu}, \mathbf{c}_\mathbf{w} + \gamma\mathbf{PV} - \mathbf{Q} \right\rangle + \left\langle \mathbf{d}, \mathbf{Q} - \mathbf{BV} \right\rangle \} \\
&= \max_{\mathbf{w}\in\mathcal{W}} \max_{\mathbf{V},\mathbf{Q}} \min_{\boldsymbol{\mu},\mathbf{d}\geq 0} \{ (1-\gamma) \left\langle \boldsymbol{\nu}_0, \mathbf{V} \right\rangle - \left\langle \boldsymbol{\mu}_{\pi_\mathrm{E}}, \mathbf{c}_\mathbf{w} \right\rangle + \left\langle \boldsymbol{\mu}, \mathbf{c}_\mathbf{w} + \gamma\mathbf{PV} - \mathbf{Q} \right\rangle + \left\langle \mathbf{d}, \mathbf{Q} - \mathbf{BV} \right\rangle \} \\
&= \max_{(\mathbf{w},\mathbf{V},\mathbf{Q})} \Big\{ (1-\gamma) \left\langle \boldsymbol{\nu}_0, \mathbf{V} \right\rangle - \left\langle \boldsymbol{\mu}_{\pi_\mathrm{E}}, \mathbf{c}_\mathbf{w} \right\rangle \mid \mathbf{Q} \geq \mathbf{BV}, \ \mathbf{Q} = \mathbf{c}_\mathbf{w} + \gamma\mathbf{PV},
\end{aligned}
$$

$$
\mathbf{V} \in \mathbb{R}^{|\mathcal{S}|}, \ \mathbf{Q} \in \mathbb{R}^{|\mathcal{S}||\mathcal{A}|}, \ \mathbf{w} \in \mathcal{W} \Big\}, \tag{Dual}
$$

where the second equality follows by Sion's minimax theorem [107], since $\mathcal{M}$ is convex and compact, $\mathcal{W}$ is convex and the objective is bilinear, the third equality follows by introducing Lagrange multipliers $\mathbf{V}$ and $\mathbf{Q}$, and the fifth equality follows by linear duality. Note that the derivations hold for any convex set $\mathcal{W}$.

From now on we consider the case $\mathcal{W} = B_1^m = \{ \mathbf{w} \in \mathbb{R}^m \mid \|\mathbf{w}\|_2 \leq 1 \}$. Then, the (Primal) program can be written in the form

$$
\begin{aligned}
\zeta^\star &= \min_{(\boldsymbol{\mu},\mathbf{d})} \{ \bar{d}_\mathcal{C}(\boldsymbol{\mu}, \boldsymbol{\mu}_{\pi_\mathrm{E}}) \mid (\boldsymbol{\mu},\mathbf{d}) \in \mathfrak{M} \} \\
&= \min_{(\boldsymbol{\mu},\mathbf{d})} \{ \max_{\mathbf{w}\in\mathcal{W}} \left\langle \boldsymbol{\mu} - \boldsymbol{\mu}_{\pi_\mathrm{E}}, \mathbf{c}_\mathbf{w} \right\rangle \mid (\boldsymbol{\mu},\mathbf{d}) \in \mathfrak{M} \} \\
&= \min_{(\boldsymbol{\mu},\mathbf{d})} \{ \max_{\mathbf{w}\in\mathcal{W}} \left\langle \boldsymbol{\Phi}^\mathsf{T}\boldsymbol{\mu} - \boldsymbol{\Phi}^\mathsf{T}\boldsymbol{\mu}_{\pi_\mathrm{E}}, \mathbf{w} \right\rangle \mid (\boldsymbol{\mu},\mathbf{d}) \in \mathfrak{M} \} \\
&= \min_{(\boldsymbol{\mu},\mathbf{d})} \{ \left\| \boldsymbol{\Phi}^\mathsf{T}\boldsymbol{\mu} - \boldsymbol{\Phi}^\mathsf{T}\boldsymbol{\mu}_{\pi_\mathrm{E}} \right\|_2 \mid (\boldsymbol{\mu},\mathbf{d}) \in \mathfrak{M} \}, \tag{Primal}
\end{aligned}
$$

where in the last equality we used that the $\ell_2$-norm is self-dual, that is, the dual norm of the $\ell_2$-norm is still the $\ell_2$-norm. Therefore, when $\mathcal{W} = B_1^m$, we get a quadratic objective with linear constraints [4].

Assume first that $(\mathbf{V}_\mathrm{A}, \mathbf{Q}_\mathrm{A}, \mathbf{w}_\mathrm{A})$ is optimal for (Dual). Then,

$$
\mathbf{Q}_\mathrm{A} \geq \mathbf{BV}_\mathrm{A}, \quad \mathbf{Q}_\mathrm{A} = \mathbf{c}_{\mathbf{w}_\mathrm{A}} + \gamma\mathbf{PV}_\mathrm{A}, \quad \mathbf{w}_\mathrm{A} \in \mathcal{W}, \tag{6}
$$

$$
(1-\gamma) \left\langle \boldsymbol{\nu}_0, \mathbf{V}_\mathrm{A} \right\rangle - \left\langle \boldsymbol{\mu}_{\pi_\mathrm{E}}, \mathbf{c}_{\mathbf{w}_\mathrm{A}} \right\rangle = \zeta^\star = 0, \tag{7}
$$

where (6) holds because $(\mathbf{V}_\mathrm{A}, \mathbf{Q}_\mathrm{A}, \mathbf{w}_\mathrm{A})$ is feasible to (Dual), and (7) holds by optimality. Therefore, $(\mathbf{V}_\mathrm{A}, \mathbf{Q}_\mathrm{A})$ is feasible for (Dual $Q$-LP) with cost $\mathbf{c} = \mathbf{c}_{\mathbf{w}_\mathrm{A}}$. Moreover, $(\boldsymbol{\mu}_{\pi_\mathrm{E}}, \boldsymbol{\mu}_{\pi_\mathrm{E}})$ is feasible for (Primal $Q$-LP) with cost $\mathbf{c} = \mathbf{c}_{\mathbf{w}_\mathrm{A}}$. Therefore,

$$
(1-\gamma) \left\langle \boldsymbol{\nu}_0, \mathbf{V}_\mathrm{A} \right\rangle \leq \rho_{\mathbf{w}_\mathrm{A}}^\star \leq \left\langle \boldsymbol{\mu}_{\pi_\mathrm{E}}, \mathbf{c}_{\mathbf{w}_\mathrm{A}} \right\rangle. \tag{8}
$$

However, by (7) we get that $(1-\gamma) \left\langle \boldsymbol{\nu}_0, \mathbf{V}_\mathrm{A} \right\rangle = \left\langle \boldsymbol{\mu}_{\pi_\mathrm{E}}, \mathbf{c}_{\mathbf{w}_\mathrm{A}} \right\rangle$. Thus, $(\boldsymbol{\mu}_{\pi_\mathrm{E}}, \boldsymbol{\mu}_{\pi_\mathrm{E}})$ is optimal for (Primal $Q$-LP) with cost $\mathbf{c} = \mathbf{c}_{\mathbf{w}_\mathrm{A}}$ and $(\mathbf{V}_\mathrm{A}, \mathbf{Q}_\mathrm{A})$ is optimal for (Dual $Q$-LP) with cost $\mathbf{c} = \mathbf{c}_{\mathbf{w}_\mathrm{A}}$. Thus $\pi_\mathrm{E}$ is optimal for the forward RL problem with cost $\mathbf{c}_{\mathbf{w}_\mathrm{A}}$, $\mathbf{V}_\mathrm{A} = \mathbf{V}_{\mathbf{c}_{\mathbf{w}_\mathrm{A}}}^\star$, and $\mathbf{Q}_\mathrm{A} = \mathbf{Q}_{\mathbf{c}_{\mathbf{w}_\mathrm{A}}}^\star$

Conversely, assume that $\mathbf{w}_\mathrm{A} \in \mathcal{W}$, $\pi_\mathrm{E}$ is optimal for $\mathbf{c}_{\mathbf{w}_\mathrm{A}}$, $\mathbf{V}_\mathrm{A} = \mathbf{V}_{\mathbf{w}_\mathrm{A}}^\star$, and $\mathbf{Q}_\mathrm{A} = \mathbf{Q}_{\mathbf{w}_\mathrm{A}}^\star$. Then, we have that $(\boldsymbol{\mu}_{\pi_\mathrm{E}}, \boldsymbol{\mu}_{\pi_\mathrm{E}})$ is optimal for (Primal $Q$-LP) with cost $\mathbf{c}_{\mathbf{w}_\mathrm{A}}$, and $(\mathbf{V}_\mathrm{A}, \mathbf{Q}_\mathrm{A})$ is optimal for (Dual $Q$-LP) with cost $\mathbf{c}_{\mathbf{w}_\mathrm{A}}$. By dual feasibility, we get

$$
\mathbf{Q}_\mathrm{A} \geq \mathbf{BV}_\mathrm{A}, \quad \mathbf{Q}_\mathrm{A} = \mathbf{c}_{\mathbf{w}_\mathrm{A}} + \gamma\mathbf{PV}_\mathrm{A}. \tag{9}
$$

Moreover, by primal-dual optimality, we have

$$(1 - \gamma) \langle \boldsymbol{\nu}_0, \mathbf{V}_A \rangle = \langle \boldsymbol{\mu}_{\pi_E}, \mathbf{c}_{\mathbf{w}_A} \rangle. \tag{10}$$

From (9), we get that $(\mathbf{V}_A, \mathbf{Q}_A, \mathbf{w}_A)$ is feasible to (Dual). Since $\zeta^\star = 0$, by (10), we conclude that $(\mathbf{V}_A, \mathbf{Q}_A, \mathbf{w}_A)$ is optimal for (Dual). $\qquad\square$

## D  Saddle-Point Formulation

By using a compact notation, we have that Primal$'$ is equivalent to the following bilinear saddle-point problem

$$\min_{\mathbf{x} \in \mathcal{X}} \max_{\mathbf{y} \in \mathcal{Y}} \langle \mathbf{y}, \mathbf{A}\mathbf{x} + \mathbf{b} \rangle, \tag{SPP}$$

where

$$\mathbf{A} \triangleq \begin{bmatrix} \mathbf{I}_m & 0 \\ -\gamma \mathbf{M}^\mathsf{T} & \mathbf{B}^\mathsf{T} \\ \mathbf{I}_m & -\boldsymbol{\Phi}^\mathsf{T} \end{bmatrix}, \qquad\qquad \mathbf{b} \triangleq \begin{bmatrix} -\boldsymbol{\rho}_{\boldsymbol{\Phi}}(\pi_E) \\ (1-\gamma)\boldsymbol{\nu}_0 \\ 0 \end{bmatrix},$$

$\mathbf{x} \triangleq [\boldsymbol{\lambda}^\mathsf{T}, \mathbf{d}^\mathsf{T}]^\mathsf{T}, \mathbf{y} \triangleq [\mathbf{w}^\mathsf{T}, \mathbf{V}^\mathsf{T}, \boldsymbol{\theta}^\mathsf{T}]^\mathsf{T}, \mathcal{X} \triangleq \Delta_{[m]} \times \Delta_{\mathcal{S} \times \mathcal{A}}$, and $\mathcal{Y} \triangleq \mathcal{W} \times \mathbb{R}^{|\mathcal{S}|} \times \mathbb{R}^m$.

## E  Proof of Proposition 2

*Proof of Proposition 2.* We break the proof in three parts. In the first two parts, we introduce and compute the explicit forms of the oracles, while in the third part we derive the proximal point updates.

**Analytical oracle.** We characterize the `analytical-oracle` by employing the first-order optimality conditions for $\boldsymbol{\lambda}$ and $\mathbf{d}$. In particular, at each iteration step $k$, for any $[\mathbf{w}^\mathsf{T}, \mathbf{V}^\mathsf{T}, \boldsymbol{\theta}^\mathsf{T}]^\mathsf{T}$, we have that the Lagrangian of the optimization problem in the definition of the `analytical-oracle` has the form

$$\langle \boldsymbol{\lambda}, \mathbf{w} \rangle - \langle \boldsymbol{\rho}_{\boldsymbol{\Phi}}(\widehat{\pi}_E), \mathbf{w} \rangle + \langle \mathbf{V}, \gamma \mathbf{M}^\mathsf{T} \boldsymbol{\lambda} + (1-\gamma)\boldsymbol{\nu}_0 - \mathbf{B}^\mathsf{T} \mathbf{d} \rangle$$
$$+ \left\langle \boldsymbol{\theta}, \boldsymbol{\Phi}^\mathsf{T} \mathbf{d} - \boldsymbol{\lambda} \right\rangle + \frac{1}{\eta} D(\boldsymbol{\lambda} || \boldsymbol{\lambda}_k) + \frac{1}{\alpha} H(\mathbf{d} || \mathbf{d}_k) + \langle \boldsymbol{\lambda}, \tau \mathbf{1} \rangle - \tau,$$

where we considered a Lagrangian multiplier $\tau$ for the simplex constraint $\sum_i \boldsymbol{\lambda}(i) = 1$. Now taking the derivatives with respect to to $\boldsymbol{\lambda}$ and $\mathbf{d}$, we obtain the following first order optimality conditions:

$$\left( \mathbf{w} + \gamma \mathbf{M}\mathbf{V} - \boldsymbol{\theta} \right)(i) + \tau + \frac{1}{\eta} \log \frac{\boldsymbol{\lambda}(i)}{\boldsymbol{\lambda}_k(i)} + \frac{1}{\eta} = 0, \text{ for all } i \in [m],$$

$$\left( \mathbf{B}\mathbf{V} + \boldsymbol{\Phi}\boldsymbol{\theta} \right)(s,a) + \frac{1}{\alpha} \log \frac{\pi_{\mathbf{d}}(a|s)}{\pi_{\mathbf{d}_k}(a|s)} = 0, \text{ for all} (s,a) \in \mathcal{S} \times \mathcal{A}.$$

Therefore, we obtain

$$\boldsymbol{\lambda}(i) = \boldsymbol{\lambda}_k(i) \, e^{-\eta \boldsymbol{\delta}^k_{\mathbf{w},\boldsymbol{\theta}}(i) + 1 - \eta\tau}, \tag{11}$$

where $\boldsymbol{\delta}^k_{\mathbf{w},\boldsymbol{\theta}} \triangleq \mathbf{w} + \gamma \mathbf{M}\mathbf{V}^k_{\boldsymbol{\theta}} - \boldsymbol{\theta}$. In addition, the simplex constraint $\sum_i \boldsymbol{\lambda}(i) = 1$ is satisfied by choosing $\tau = \tau^k_{\mathbf{w},\boldsymbol{\theta}}$, where

$$\tau^k_{\mathbf{w},\boldsymbol{\theta}} \triangleq \frac{1}{\eta} \log \left( \sum_{i=1}^m (\boldsymbol{\Phi}^\mathsf{T} \mathbf{d}_k)(i) e^{-\eta \boldsymbol{\delta}^k_{\mathbf{w},\boldsymbol{\theta}}(i)} \right). \tag{12}$$

Moreover, by setting $\mathbf{Q}_{\boldsymbol{\theta}} = \boldsymbol{\Phi}\boldsymbol{\theta}$, we get

$$\pi_{\mathbf{d}}(a|s) = \pi_{\mathbf{d}_k}(a|s) \, e^{-\alpha(Q_{\boldsymbol{\theta}}(s,a) - V(s))}. \tag{13}$$

Equation (4) follows by noting that the constraint $\sum_a \pi_{\mathbf{d}}(a|x)$ implies that $\mathbf{V}$ equals the logistic value function $\mathbf{V}^k_{\boldsymbol{\theta}}$ given in Proposition 2. Finally, since $(\boldsymbol{\lambda}_k, \mathbf{d}_k)$ are ideal updates, they are primal feasible. Hence, we can use the constraint $\boldsymbol{\lambda}_k = \boldsymbol{\Phi}^\mathsf{T} \mathbf{d}_k$ in Equation (11) to obtain Equation (3).

All in all, for any $\mathbf{y} = [\mathbf{w}^\mathsf{T}, \mathbf{V}^\mathsf{T}, \boldsymbol{\theta}^\mathsf{T}]^\mathsf{T}$ the `analytical-oracle` outputs $\mathbf{g}(\mathbf{y}; \mathbf{x}_k) = [\boldsymbol{\lambda}^\mathsf{T}, \mathbf{d}^\mathsf{T}]^\mathsf{T}$ with

$$\boldsymbol{\lambda}(i) \propto (\boldsymbol{\Phi}^\mathsf{T} \mathbf{d}_k)(i) \, e^{-\eta \boldsymbol{\delta}^k_{\mathbf{w},\boldsymbol{\theta}}(i)}, \tag{14}$$

$$\pi_{\mathbf{d}}(a|s) = \pi_{\mathbf{d}_k}(a|s) \, e^{-\alpha \left(Q_{\boldsymbol{\theta}}(s,a) - V^k_{\boldsymbol{\theta}}(s)\right)}. \tag{15}$$

Note that the derivatives with respect to $\boldsymbol{\lambda}$ and $\mathbf{d}$ differ from the ones in Logistic Q-Learning [14]. In our case, $\boldsymbol{\delta}^k_{\mathbf{w},\boldsymbol{\theta}}$ depends on both cost weights $\mathbf{w}$ and logistic action-value parameters $\boldsymbol{\theta}$. In addition, $\boldsymbol{\delta}^k_{\mathbf{w},\boldsymbol{\theta}}$ is the reduced Bellman error in the feature space rather than in the high dimensional state-action space.

**Max oracle.** Since the objective in (2) is convex in $\mathbf{x}$ and linear in $\mathbf{y}$, $\mathcal{X}$ is convex and compact, and $\mathcal{Y}$ is convex, by virtue of Sion's minimax theorem [107], we can exchange the $\min$ and $\max$ in Equation (2). We then have

$$\min_{\mathbf{x}\in\mathcal{X}}\max_{\mathbf{y}\in\mathcal{Y}}\left\langle\mathbf{y},\mathbf{Ax}+\widehat{\mathbf{b}}\right\rangle+\frac{1}{\tau}D_\Omega(\mathbf{x}||\mathbf{x}_k)=\max_{\mathbf{y}\in\mathcal{Y}}\min_{\mathbf{x}\in\mathcal{X}}\left\langle\mathbf{y},\mathbf{Ax}+\widehat{\mathbf{b}}\right\rangle+\frac{1}{\tau}D_\Omega(\mathbf{x}||\mathbf{x}_k).$$

Therefore, we get

$$\mathbf{y}^\star=\arg\max_{\mathbf{y}\in\mathcal{Y}}\min_{\mathbf{x}\in\mathcal{X}}\left\langle\mathbf{y},\mathbf{Ax}+\widehat{\mathbf{b}}\right\rangle+\frac{1}{\tau}D_\Omega(\mathbf{x}||\mathbf{x}_k)$$

$$=\arg\max_{\mathbf{y}\in\mathcal{Y}}\left\langle\mathbf{y},\mathbf{Ag}(\mathbf{y};\mathbf{x}_k)+\widehat{\mathbf{b}}\right\rangle+\frac{1}{\tau}D_\Omega(\mathbf{g}(\mathbf{y};\mathbf{x}_k)||\mathbf{x}_k)$$

$$=\mathbf{h}(\mathbf{x}_k).$$

**Proximal point updates via max and analytical oracles.** It remains to prove the closed-form expressions for $\pi_{\mathbf{d}^\star}$ and $\boldsymbol{\lambda}^\star$ given in Equation (3) and Equation (4), respectively. We start rewriting the objective of the `max-oracle` as a function of $\boldsymbol{\lambda}$ and $\mathbf{d}$. In particular, we have

$$\left\langle\mathbf{y},\mathbf{Ag}(\mathbf{y};\mathbf{x}_k)+\widehat{\mathbf{b}}\right\rangle+\frac{1}{\tau}D_\Omega(\mathbf{g}(\mathbf{y};\mathbf{x}_k)||\mathbf{x}_k)$$

$$=\min_{\mathbf{d}\in\Delta_{\mathcal{S}\times\mathcal{A}},\boldsymbol{\lambda}\in\Delta_{[m]}}\left\langle\mathbf{y},\mathbf{A}\left[\begin{array}{c}\boldsymbol{\lambda}\\\mathbf{d}\end{array}\right]+\widehat{\mathbf{b}}\right\rangle+\frac{1}{\alpha}H(\mathbf{d}||\mathbf{d}_k)+\frac{1}{\eta}D(\boldsymbol{\lambda}||\boldsymbol{\lambda}_k).$$

The minimizers of the previous expression are characterized via the `analytical-oracle`. In particular, plugging in the analytical forms for $\boldsymbol{\lambda},\mathbf{d}$ and $\mathbf{V}$, we obtain

$$\min_{\mathbf{d}\in\Delta_{\mathcal{S}\times\mathcal{A}},\boldsymbol{\lambda}\in\Delta_{[m]}}\left\langle\mathbf{y},\mathbf{A}\left[\begin{array}{c}\boldsymbol{\lambda}\\\mathbf{d}\end{array}\right]+\widehat{\mathbf{b}}\right\rangle+\frac{1}{\alpha}H(\mathbf{d}||\mathbf{d}_k)+\frac{1}{\eta}D(\boldsymbol{\lambda}||\boldsymbol{\lambda}_k)$$

$$=\langle\boldsymbol{\lambda},\mathbf{w}\rangle-\langle\boldsymbol{\rho}_\Phi(\widehat{\pi}_{\mathrm{E}}),\mathbf{w}\rangle+\frac{1}{\eta}\left\langle\boldsymbol{\lambda},-\eta\boldsymbol{\delta}^k_{\mathbf{w},\boldsymbol{\theta}}-\eta\tau^k_{\mathbf{w},\boldsymbol{\theta}}\right\rangle$$

$$+\frac{1}{\alpha}\left\langle\mathbf{d},-\alpha(\boldsymbol{\Phi\theta}-\mathbf{BV}^k_{\boldsymbol{\theta}})\right\rangle+\left\langle\boldsymbol{\lambda},\gamma\mathbf{M}^\intercal\mathbf{V}^k_{\boldsymbol{\theta}}\right\rangle$$

$$-\left\langle\mathbf{d},\mathbf{BV}^k_{\boldsymbol{\theta}}\right\rangle+(1-\gamma)\left\langle\boldsymbol{\nu}_0,\mathbf{V}^k_{\boldsymbol{\theta}}\right\rangle+\langle\mathbf{d},\boldsymbol{\Phi\theta}\rangle-\langle\boldsymbol{\lambda},\boldsymbol{\theta}\rangle$$

$$=-\langle\boldsymbol{\rho}_\Phi(\widehat{\pi}_{\mathrm{E}}),\mathbf{w}\rangle+(1-\gamma)\left\langle\boldsymbol{\nu}_0,\mathbf{V}^k_{\boldsymbol{\theta}}\right\rangle-\tau^k_{\mathbf{w},\boldsymbol{\theta}}$$

$$=-\langle\boldsymbol{\rho}_\Phi(\widehat{\pi}_{\mathrm{E}}),\mathbf{w}\rangle+(1-\gamma)\left\langle\boldsymbol{\nu}_0,\mathbf{V}^k_{\boldsymbol{\theta}}\right\rangle-\frac{1}{\eta}\log\left(\sum_{i=1}^m(\boldsymbol{\Phi}^\intercal\mathbf{d}_k)(i)e^{-\eta\boldsymbol{\delta}^k_{\mathbf{w},\boldsymbol{\theta}}(i)}\right)$$

$$\triangleq\mathcal{G}_k(\mathbf{w},\boldsymbol{\theta}).$$

This is the objective of the `max-oracle` in Proposition 2. Given that the `max-oracle` returns $(\mathbf{w}^\star_k,\boldsymbol{\theta}^\star_k)$, the corresponding primal variables $(\mathbf{d}^\star_k,\boldsymbol{\lambda}^\star_k)$ satisfy $(\mathbf{d}^\star_k,\boldsymbol{\lambda}^\star_k)=\mathbf{g}([\mathbf{w}^\star_k,\mathbf{V}_{\boldsymbol{\theta}^\star_k},\boldsymbol{\theta}^\star_k];\mathbf{x}_{k-1})$. This completes the proof of the first part of Proposition 2.

It remains to show the dual form of the max-oracle objective $\mathcal{G}_k(\mathbf{w},\boldsymbol{\theta})$. In particular, we will show that

$$\max_{\mathbf{w},\boldsymbol{\theta}}\mathcal{G}_k(\mathbf{w},\boldsymbol{\theta})=\max_{\mathbf{w}}\langle\boldsymbol{\lambda}_{k+1},\mathbf{w}\rangle-\langle\boldsymbol{\rho}_\Phi(\pi_{\mathrm{E}}),\mathbf{w}\rangle+\frac{1}{\eta}D(\boldsymbol{\lambda}_{k+1}||\boldsymbol{\lambda}_k)+\frac{1}{\alpha}H(\mathbf{d}_{k+1}||\mathbf{d}_k).\tag{16}$$

We first recall that

$$\mathcal{G}_k(\mathbf{w},\boldsymbol{\theta})=\min_{\mathbf{d}\in\Delta_{\mathcal{S}\times\mathcal{A}},\boldsymbol{\lambda}\in\Delta_{[m]}}\left\langle\mathbf{y},\mathbf{A}\left[\begin{array}{c}\boldsymbol{\lambda}\\\mathbf{d}\end{array}\right]+\widehat{\mathbf{b}}\right\rangle+\frac{1}{\alpha}H(\mathbf{d}||\mathbf{d}_k)+\frac{1}{\eta}D(\boldsymbol{\lambda}||\boldsymbol{\lambda}_k).$$

Then, by taking the maximum over $\mathbf{y} = [\mathbf{w}, \mathbf{V}, \boldsymbol{\theta}]$ on both sides and using Sion's minimax theorem, we get

$$\max_{\mathbf{w}, \boldsymbol{\theta}} \mathcal{G}_k(\mathbf{w}, \boldsymbol{\theta}) = \max_{\mathbf{y} \in \mathcal{Y}} \min_{\mathbf{d} \in \Delta_{\mathcal{S} \times \mathcal{A}}, \boldsymbol{\lambda} \in \Delta_{[m]}} \left\langle \mathbf{y}, \mathbf{A} \begin{bmatrix} \boldsymbol{\lambda} \\ \mathbf{d} \end{bmatrix} + \widehat{\mathbf{b}} \right\rangle + \frac{1}{\alpha} H(\mathbf{d}\|\mathbf{d}_k) + \frac{1}{\eta} D(\boldsymbol{\lambda}\|\boldsymbol{\lambda}_k)$$

$$= \min_{\mathbf{d} \in \Delta_{\mathcal{S} \times \mathcal{A}}, \boldsymbol{\lambda} \in \Delta_{[m]}} \max_{\mathbf{y} \in \mathcal{Y}} \left\langle \mathbf{y}, \mathbf{A} \begin{bmatrix} \boldsymbol{\lambda} \\ \mathbf{d} \end{bmatrix} + \widehat{\mathbf{b}} \right\rangle + \frac{1}{\alpha} H(\mathbf{d}\|\mathbf{d}_k) + \frac{1}{\eta} D(\boldsymbol{\lambda}\|\boldsymbol{\lambda}_k)$$

$$= \max_{\mathbf{y} \in \mathcal{Y}} \left\langle \mathbf{y}, \mathbf{A} \begin{bmatrix} \boldsymbol{\lambda}_{k+1} \\ \mathbf{d}_{k+1} \end{bmatrix} + \widehat{\mathbf{b}} \right\rangle + \frac{1}{\alpha} H(\mathbf{d}_{k+1}\|\mathbf{d}_k) + \frac{1}{\eta} D(\boldsymbol{\lambda}_{k+1}\|\boldsymbol{\lambda}_k),$$

where in the last equality we used the definition of proximal point update in Equation (2). Finally, by LP strong duality, we have that $\max_{\mathbf{w}} \langle \boldsymbol{\lambda}_{k+1}, \mathbf{w} \rangle - \langle \boldsymbol{\rho}_{\boldsymbol{\Phi}}(\widehat{\pi}_{\mathrm{E}}), \mathbf{w} \rangle = \max_{\mathbf{y} \in \mathcal{Y}} \left\langle \mathbf{y}, \mathbf{A} \left[ \boldsymbol{\lambda}_{k+1}^{\mathsf{T}}, \mathbf{d}_{k+1}^{\mathsf{T}} \right]^{\mathsf{T}} + \widehat{\mathbf{b}} \right\rangle$. Hence, we conclude that (16) holds.

$\square$

# F   Proof of Proposition 3

*Proof of Proposition 3.* From first order optimality conditions for $\boldsymbol{\lambda}_k^\star$, we get

$$\left( \mathbf{w}_k^\star + \gamma \mathbf{M} \mathbf{V}_{\boldsymbol{\theta}_k^\star}^k - \boldsymbol{\theta}_k^\star \right)(i) + \tau_{\mathbf{w}_k^\star, \boldsymbol{\theta}_k^\star}^k + \frac{1}{\eta} \log \frac{\boldsymbol{\lambda}_k^\star(i)}{\boldsymbol{\lambda}_{k-1}(i)} - \frac{1}{\eta} = 0, \text{ for all } i \in [m]. \qquad (17)$$

We define the regularized cost weights by $\widetilde{\mathbf{w}}_k^\star \triangleq \mathbf{w}_k^\star + \frac{1}{\eta} \log \frac{\boldsymbol{\lambda}_k^\star(i)}{\boldsymbol{\lambda}_{k-1}(i)}$, and the costant (wrt the vector index $i$) $c \triangleq -\tau_{\mathbf{w}_k^\star, \boldsymbol{\theta}_k^\star}^k + \frac{1}{\eta}$. This gives for all $i \in [m]$

$$\left( \widetilde{\mathbf{w}}_k^\star + \gamma \mathbf{M} \mathbf{V}_{\boldsymbol{\theta}_k^\star}^k - \boldsymbol{\theta}_k^\star \right)(i) = c.$$

We define the *span norm* as $\|\mathbf{x}\|_{\mathrm{sp}} = \inf_{c \in \mathbb{R}} \|\mathbf{x} - c\mathbf{1}\|_\infty$. Then multiplying by $\boldsymbol{\Phi}$ from the left, we have that $\boldsymbol{\Phi}\widetilde{\mathbf{w}}_k^\star + \gamma \mathbf{P} \mathbf{V}_{\boldsymbol{\theta}_k^\star}^k - \boldsymbol{\Phi}\boldsymbol{\theta}_k^\star = c\mathbf{1}$. Moreover, we can write

$$\mathbf{V}_{\boldsymbol{\theta}_k^\star}^k(s) = -\frac{1}{\alpha} \log \left( \sum_a \pi_{\mathbf{d}_{k-1}}(a|s) e^{-\alpha(\boldsymbol{\theta}_k^\star)^{\mathsf{T}} \phi(s,a)} \right)$$

$$= -\frac{1}{\alpha} \log \left( \sum_a \pi_{\mathbf{d}_{k-1}}(a|s) e^{-\alpha(\boldsymbol{\Phi}\widetilde{\mathbf{w}}_k^\star + \gamma \mathbf{P} \mathbf{V}_{\boldsymbol{\theta}_k^\star}^k)(s,a) + \alpha c} \right)$$

$$= -\frac{1}{\alpha} \log \left( \sum_a \pi_{\mathbf{d}_{k-1}}(a|s) e^{-\alpha(\boldsymbol{\Phi}\widetilde{\mathbf{w}}_k^\star + \gamma \mathbf{P} \mathbf{V}_{\boldsymbol{\theta}_k^\star}^k)(s,a)} \right) + c.$$

We set $(\mathcal{T} \mathbf{V}_{\boldsymbol{\theta}_k^\star}^k)(s) \triangleq -\frac{1}{\alpha} \log \left( \sum_a \pi_{\mathbf{d}_{k-1}}(a|s) e^{-\alpha(\boldsymbol{\Phi}\widetilde{\mathbf{w}}_k^\star + \gamma \mathbf{P} \mathbf{V}_{\boldsymbol{\theta}_k^\star}^k)(s,a)} \right)$. Note that $\mathcal{T}$ is the soft-Bellman operator [86, 41] that is a $\gamma$-contraction with respect to $\|\cdot\|_\infty$-norm. It follows that

$$\left\| \mathbf{V}_{\boldsymbol{\theta}_k^\star}^k \right\|_{\mathrm{sp}} = \left\| \mathcal{T} \mathbf{V}_{\boldsymbol{\theta}_k^\star}^k + c \right\|_{\mathrm{sp}} = \left\| \mathcal{T} \mathbf{V}_{\boldsymbol{\theta}_k^\star}^k \right\|_{\mathrm{sp}} \le \left\| \mathcal{T} \mathbf{V}_{\boldsymbol{\theta}_k^\star}^k - \mathcal{T}\mathbf{0} \right\|_{\mathrm{sp}} + \|\mathcal{T}\mathbf{0}\|_{\mathrm{sp}} \le \gamma \left\| \mathbf{V}_{\boldsymbol{\theta}_k^\star}^k \right\|_{\mathrm{sp}} + \|\boldsymbol{\Phi}\widetilde{\mathbf{w}}_k^\star\|_{\mathrm{sp}}.$$

Therefore, $\left\|\mathbf{V}_{\boldsymbol{\theta}_k^\star}^k\right\|_{\mathrm{sp}} \le \frac{\|\boldsymbol{\Phi}\widetilde{\mathbf{w}}_k^\star\|_{\mathrm{sp}}}{1-\gamma} \le \frac{1+\log\frac{1}{\beta}}{1-\gamma}$. Moreover, using the relation $\widetilde{\mathbf{w}}_k^\star + \gamma\mathbf{M}\mathbf{V}_{\boldsymbol{\theta}_k^\star}^k - \boldsymbol{\theta}_k^\star = c\mathbf{1}$, we have that

$$\|\boldsymbol{\theta}_k^\star\|_{\mathrm{sp}} \le \left\|\widetilde{\mathbf{w}}_k^\star + \gamma\mathbf{M}\mathbf{V}_{\boldsymbol{\theta}_k^\star}^k - c\mathbf{1}\right\|_{\mathrm{sp}}$$

$$= \left\|\widetilde{\mathbf{w}}_k^\star + \gamma\mathbf{M}\mathbf{V}_{\boldsymbol{\theta}_k^\star}^k\right\|_{\mathrm{sp}}$$

$$\le \|\widetilde{\mathbf{w}}_k^\star\|_{\mathrm{sp}} + \gamma\left\|\mathbf{M}\mathbf{V}_{\boldsymbol{\theta}_k^\star}^k\right\|_{\mathrm{sp}}$$

$$\le 1 + \log\left(\frac{1}{\beta}\right) + \gamma\frac{1+\log\left(\frac{1}{\beta}\right)}{1-\gamma}$$

$$= \frac{1+\log\left(\frac{1}{\beta}\right)}{1-\gamma}.$$

This proves that for every maximizer the span norm is bounded. Finally, for showing that there exists a maximizer with bounded infinity norm, we want to prove that the negative logistic Bellman error is shift invariant in $\boldsymbol{\theta}$. That is, $\mathcal{G}_k(\mathbf{w}, \boldsymbol{\theta} + c\mathbf{1}) = \mathcal{G}_k(\mathbf{w}, \boldsymbol{\theta})$. Towards this goal, we start proving that $\mathbf{V}_{\boldsymbol{\theta}+c\mathbf{1}}^k = \mathbf{V}_{\boldsymbol{\theta}}^k + c$ for any constant $c \in \mathbb{R}$. Indeed,

$$\mathbf{V}_{\boldsymbol{\theta}+c\mathbf{1}}^k(s) = -\frac{1}{\alpha}\log\left(\sum_a \pi_{\mathbf{d}_{k-1}}(a|s)e^{-\alpha\boldsymbol{\theta}^\mathsf{T}\boldsymbol{\phi}(s,a)+-\alpha c\mathbf{1}^\mathsf{T}\boldsymbol{\phi}(s,a)}\right)$$

$$= -\frac{1}{\alpha}\log\left(\sum_a \pi_{\mathbf{d}_{k-1}}(a|s)e^{-\alpha\boldsymbol{\theta}^\mathsf{T}\boldsymbol{\phi}(s,a)-\alpha c}\right)$$

$$= -\frac{1}{\alpha}\log\left(\sum_a \pi_{\mathbf{d}_{k-1}}(a|s)e^{-\alpha\boldsymbol{\theta}^\mathsf{T}\boldsymbol{\phi}(s,a)}\right) - \frac{1}{\alpha}\log\left(e^{-\alpha c}\right)$$

$$= \mathbf{V}_{\boldsymbol{\theta}}^k(s) + c$$

At this point, we can show the shift invariance of $\mathcal{G}_k$.

$$\mathcal{G}_k(\mathbf{w}, \boldsymbol{\theta} + c\mathbf{1}) = -\frac{1}{\eta}\log\sum_{i=1}^m (\boldsymbol{\Phi}^\mathsf{T}\mathbf{d}_{k-1})(i)e^{-\eta\left(\mathbf{w}(i)+\gamma(\mathbf{M}\mathbf{V}_{\boldsymbol{\theta}+c\mathbf{1}}^k)(i)-\boldsymbol{\theta}(i)-c\right)}$$

$$+ (1-\gamma)\left\langle\boldsymbol{\nu}_0, \mathbf{V}_{\boldsymbol{\theta}+c\mathbf{1}}^k\right\rangle - \left\langle\boldsymbol{\rho}_{\boldsymbol{\Phi}}(\widehat{\pi}_\mathrm{E}), \mathbf{w}\right\rangle$$

$$= -\frac{1}{\eta}\log\sum_{i=1}^m (\boldsymbol{\Phi}^\mathsf{T}\mathbf{d}_{k-1})(i)e^{-\eta\left(\mathbf{w}(i)+\gamma(\mathbf{M}\mathbf{V}_{\boldsymbol{\theta}}^k)(i)+\gamma c-\boldsymbol{\theta}(i)-c\right)}$$

$$+ (1-\gamma)\left\langle\boldsymbol{\nu}_0, \mathbf{V}_{\boldsymbol{\theta}}^k\right\rangle + (1-\gamma)c - \left\langle\boldsymbol{\rho}_{\boldsymbol{\Phi}}(\widehat{\pi}_\mathrm{E}), \mathbf{w}\right\rangle$$

$$= -\frac{1}{\eta}\log\underbrace{\sum_{i=1}^m (\boldsymbol{\Phi}^\mathsf{T}\mathbf{d}_{k-1})(i)\,e^{-\eta\gamma c+\eta c}}_{=1} + (1-\gamma)c + \mathcal{G}_k(\mathbf{w}, \boldsymbol{\theta})$$

$$= -(1-\gamma)c + (1-\gamma)c + \mathcal{G}_k(\mathbf{w}, \boldsymbol{\theta})$$

$$= \mathcal{G}_k(\mathbf{w}, \boldsymbol{\theta})$$

It follows that there exists a maximizer $\boldsymbol{\theta}_k^\star$ for which $\|\boldsymbol{\theta}_k^\star\|_{\mathrm{sp}} = \|\boldsymbol{\theta}_k^\star\|_\infty$. To see this, we show that we can find a value of $c$ for which the span seminorm equals the $\ell_\infty$-norm, that is $\|\boldsymbol{\theta}_k^\star + c\mathbf{1}\|_\infty = \|\boldsymbol{\theta}_k^\star\|_{\mathrm{sp}}$. By definition of the span norm (and assuming that the infimum is attained), the equality is attained for $c = \arg\min_c \|\boldsymbol{\theta}_k^\star + c\mathbf{1}\|_\infty = \frac{\max_{i\in[m]}\boldsymbol{\theta}_k^\star(i)+\min_{i\in[m]}\boldsymbol{\theta}_k^\star(i)}{2}$. Then, choosing the shift for which $\max_{i\in[m]}\boldsymbol{\theta}_k^\star(i) = -\min_{i\in[m]}\boldsymbol{\theta}_k^\star(i)$, gives the maximizer for which $\|\boldsymbol{\theta}_k^\star\|_{\mathrm{sp}} = \|\boldsymbol{\theta}_k^\star\|_\infty$. This concludes the proof for the bound on the $\ell_\infty$-norm. $\qquad\square$

# G   Proof of Theorem 1

We will analyze the proximal point method applied to SPP. We use a similar error propagation analysis as in [14].

By Proposition 2, the ideal updates $(\boldsymbol{\theta}_k^\star, \mathbf{w}_k^\star, \pi_k^\star, \boldsymbol{\lambda}_k^\star, \mathbf{d}_k^\star)$ are given by

$$(\mathbf{w}_k^\star, \boldsymbol{\theta}_k^\star) = \arg\max_{\mathbf{w},\boldsymbol{\theta}} \mathcal{G}_k(\mathbf{w}, \boldsymbol{\theta}), \qquad \boldsymbol{\lambda}_k^\star(i) = (\boldsymbol{\Phi}^\mathsf{T} \mathbf{d}_{k-1}^\star)(i)\, e^{-\eta(\boldsymbol{\delta}_{\boldsymbol{\theta}_k^\star, \mathbf{w}_k^\star}^k(i) + \tau_{\boldsymbol{\theta}_k^\star, \mathbf{w}_k^\star}^k)},$$

$$\mathbf{d}_k^\star = \boldsymbol{\mu}_{\pi_k^\star}, \qquad \pi_k^\star(a|s) = \pi_{\mathbf{d}_{k-1}^\star}(a|s)\, e^{-\alpha(Q_{\boldsymbol{\theta}_k^\star}(s,a) - V_{\boldsymbol{\theta}_k^\star}^k(s))},$$

where $\tau_{\boldsymbol{\theta}_k^\star, \mathbf{w}_k^\star}^k$ is a normalization constant. By feasibility of the ideal updates we also have $\boldsymbol{\lambda}_k^\star = \boldsymbol{\Phi}^\mathsf{T} \mathbf{d}_k^\star$
On the other hand, the realized updates $(\boldsymbol{\theta}_k, \mathbf{w}_k, \pi_k, \boldsymbol{\lambda}_k, \mathbf{d}_k)$ are given by

$$(\mathbf{w}_k, \boldsymbol{\theta}_k) = \arg\max_{\mathbf{w},\boldsymbol{\theta}} \mathcal{G}_k^{\epsilon_k}(\mathbf{w}, \boldsymbol{\theta}), \qquad \boldsymbol{\lambda}_k(i) = (\boldsymbol{\Phi}^\mathsf{T} \mathbf{d}_{k-1})(i)\, e^{-\eta(\boldsymbol{\delta}_{\boldsymbol{\theta}_k, \mathbf{w}_k}^k(i) + \tau_{\boldsymbol{\theta}_k, \mathbf{w}_k}^k)},$$

$$\mathbf{d}_k = \boldsymbol{\mu}_{\pi_k}, \qquad \pi_k(a|s) = \pi_{\mathbf{d}_{k-1}}(a|s)\, e^{-\alpha(Q_{\boldsymbol{\theta}_k}(s,a) - V_{\boldsymbol{\theta}_k}^k(s))},$$

where $\tau_{\boldsymbol{\theta}_k, \mathbf{w}_k}^k$ is a normalization constant, and the notation $(\mathbf{w}_k, \boldsymbol{\theta}_k) = \arg\max_{\mathbf{w},\boldsymbol{\theta}} \mathcal{G}_k^{\epsilon_k}(\mathbf{w}, \boldsymbol{\theta})$ means that $\mathcal{G}_k(\mathbf{w}_k^\star, \boldsymbol{\theta}_k^\star) - \mathcal{G}_k(\mathbf{w}_k, \boldsymbol{\theta}_k) = \epsilon_k$. We start by introducing some auxiliary results

**Lemma 1.** *For any occupancy measures* $\mathbf{d}_1, \mathbf{d}_2 \in \mathfrak{F}$, *and for any cost vectors* $\mathbf{c}, \mathbf{c}' \in \mathcal{C}$, *we have:*

$$\langle \boldsymbol{\mu}_{\pi_E} - \mathbf{d}_1, \mathbf{c} \rangle - \min_{\mathbf{c}' \in \mathcal{C}} \langle \boldsymbol{\mu}_{\pi_E} - \mathbf{d}_2, \mathbf{c}' \rangle \geq d_\mathcal{C}(\pi_E, \pi_{\mathbf{d}_2}) - d_\mathcal{C}(\pi_E, \pi_{\mathbf{d}_1}).$$

*Proof.* We have that

$$\langle \boldsymbol{\mu}_{\pi_E} - \mathbf{d}_1, \mathbf{c} \rangle - \min_{\mathbf{c}' \in \mathcal{C}} \langle \boldsymbol{\mu}_{\pi_E} - \mathbf{d}_2, \mathbf{c}' \rangle \geq \min_{\mathbf{c} \in \mathcal{C}} \langle \boldsymbol{\mu}_{\pi_E} - \mathbf{d}_1, \mathbf{c} \rangle - \min_{\mathbf{c}' \in \mathcal{C}} \langle \boldsymbol{\mu}_{\pi_E} - \mathbf{d}_2, \mathbf{c}' \rangle$$

$$= \max_{\mathbf{c}'} \langle \mathbf{d}_2 - \boldsymbol{\mu}_{\pi_E}, \mathbf{c}' \rangle - \max_{\mathbf{c}} \langle \mathbf{d}_1 - \boldsymbol{\mu}_{\pi_E}, \mathbf{c} \rangle$$

$$= d_\mathcal{C}(\pi_E, \pi_{\mathbf{d}_2}) - d_\mathcal{C}(\pi_E, \pi_{\mathbf{d}_1}).$$

$\square$

**Corollary 2.** *Let* $\mathbf{d}^\star = \operatorname{argmin}_{\mathbf{d} \in \mathfrak{F}} \max_{\mathbf{c} \in \mathcal{C}} \langle \mathbf{d}, \mathbf{c} \rangle - \langle \boldsymbol{\mu}_{\pi_E}, \mathbf{c} \rangle$. *Setting* $\mathbf{c} = \boldsymbol{\Phi}\mathbf{w}_k$, $\mathbf{d}_1 = \mathbf{d}^\star$, $\mathbf{d}_2 = \mathbf{d}_k$, *we ge that* $\langle \boldsymbol{\mu}_{\pi_E} - \mathbf{d}^\star, \boldsymbol{\Phi}\mathbf{w}_k \rangle - \min_{\mathbf{c} \in \mathcal{C}} \langle \boldsymbol{\mu}_{\pi_E} - \mathbf{d}_k, \mathbf{c} \rangle \geq d_\mathcal{C}(\pi_E, \pi_{\mathbf{d}_k}) - d_\mathcal{C}(\pi_E, \pi_{\mathbf{d}^\star})$.

**Lemma 2.** *It holds that* $\frac{D(\boldsymbol{\lambda}_k^\star \| \boldsymbol{\lambda}_k)}{\eta} + \frac{H(\mathbf{d}_k^\star \| \mathbf{d}_k)}{\alpha} = \langle \boldsymbol{\rho}_{\boldsymbol{\Phi}}(\widehat{\pi_E}) - \boldsymbol{\lambda}_k^\star, (\mathbf{w}_k^\star - \mathbf{w}_k) \rangle + \epsilon_k$.

*Proof.* The proof is analogous to Lemma 1 in [14]. $\square$

**Lemma 3** (First order optimality conditions for $\mathcal{G}_k$)**.** *For all* $k \in [K]$, *it holds that*

$$\langle \boldsymbol{\rho}_{\boldsymbol{\Phi}}(\widehat{\pi_E}) - \boldsymbol{\lambda}_k^\star, \mathbf{w}_k^\star - \mathbf{w}_k \rangle \leq 0.$$

*Proof.* We start by taking the gradient of $\mathcal{G}_k(\mathbf{w}_k, \boldsymbol{\theta}_k)$ with respect to $\mathbf{w}$. In particular, the partial derivative with respect to the $i^{th}$ component is given by

$$\frac{\partial \mathcal{G}_k(\mathbf{w}_k^\star, \boldsymbol{\theta}_k^\star)}{\partial \mathbf{w}(i)} = -(\boldsymbol{\rho}_{\boldsymbol{\Phi}}(\widehat{\pi_E}))(i) + \frac{(\boldsymbol{\Phi}^\mathsf{T} \mathbf{d}_{k-1})(i) e^{-\eta \boldsymbol{\delta}_{\boldsymbol{\theta}_k^\star, \mathbf{w}_k^\star}^k(i)}}{\sum_{i=1}^m (\boldsymbol{\Phi}^\mathsf{T} \mathbf{d}_{k-1})(i) e^{-\eta \boldsymbol{\delta}_{\boldsymbol{\theta}_k^\star, \mathbf{w}_k^\star}^k(i)}}$$

$$= -(\boldsymbol{\rho}_{\boldsymbol{\Phi}}(\widehat{\pi_E}))(i) + \boldsymbol{\lambda}_k^\star(i).$$

Therefore,

$$\nabla_{\mathbf{w}} \mathcal{G}_k(\mathbf{w}_k^\star, \boldsymbol{\theta}_k^\star) = -\boldsymbol{\rho}_{\boldsymbol{\Phi}}(\widehat{\pi_E}) + \boldsymbol{\lambda}_k^\star.$$

Then, by using the first-order optimality conditions for a concave function, we have

$$\langle \nabla_{\mathbf{w}} \mathcal{G}_k(\mathbf{w}_k^\star, \boldsymbol{\theta}_k^\star), \mathbf{w}_k - \mathbf{w}_k^\star \rangle \leq 0, \quad \forall k.$$

By replacing the expression for $\nabla_{\mathbf{w}} \mathcal{G}_k(\mathbf{w}_k^\star, \boldsymbol{\theta}_k^\star)$, we obtain

$$\langle -\boldsymbol{\rho}_{\boldsymbol{\Phi}}(\widehat{\pi_E}) + \boldsymbol{\lambda}_k^\star, \mathbf{w}_k - \mathbf{w}_k^\star \rangle \leq 0 \quad \forall k \iff \langle \boldsymbol{\rho}_{\boldsymbol{\Phi}}(\widehat{\pi_E}) - \boldsymbol{\lambda}_k^\star, \mathbf{w}_k^\star - \mathbf{w}_k \rangle, \leq 0 \quad \forall k. \tag{18}$$

$\square$

We also need the following auxiliary result.

**Lemma 4.** *For all $k \in [K]$, it holds that*

$$\langle \boldsymbol{\rho}_{\boldsymbol{\Phi}}(\widehat{\pi}_{\mathrm{E}}) - \boldsymbol{\Phi}^{\mathsf{T}}\mathbf{d}_k, \mathbf{w}_k^{\star}\rangle \leq \min_{\mathbf{w}\in\mathcal{W}}\langle \boldsymbol{\rho}_{\boldsymbol{\Phi}}(\widehat{\pi}_{\mathrm{E}}) - \boldsymbol{\Phi}^{\mathsf{T}}\mathbf{d}_k, \mathbf{w}\rangle + 2\,\|\mathbf{d}_k - \mathbf{d}_k^{\star}\|_1. \tag{19}$$

*Proof.* By introducing $\bar{\mathbf{w}}_k^{\star} = \arg\min_{\mathbf{w}\in\mathcal{W}}\langle\boldsymbol{\rho}_{\boldsymbol{\Phi}}(\widehat{\pi}_{\mathrm{E}}) - \boldsymbol{\Phi}^{\mathsf{T}}\mathbf{d}_k, \mathbf{w}\rangle$, and applying triangular inequality, we obtain

$$\langle \boldsymbol{\rho}_{\boldsymbol{\Phi}}(\widehat{\pi}_{\mathrm{E}}) - \boldsymbol{\Phi}^{\mathsf{T}}\mathbf{d}_k, \mathbf{w}_k^{\star}\rangle = \langle \boldsymbol{\rho}_{\boldsymbol{\Phi}}(\widehat{\pi}_{\mathrm{E}}) - \boldsymbol{\Phi}^{\mathsf{T}}\mathbf{d}_k, \bar{\mathbf{w}}_k^{\star}\rangle + \langle \boldsymbol{\rho}_{\boldsymbol{\Phi}}(\widehat{\pi}_{\mathrm{E}}) - \boldsymbol{\Phi}^{\mathsf{T}}\mathbf{d}_k, \mathbf{w}_k^{\star} - \bar{\mathbf{w}}_k^{\star}\rangle$$
$$= \min_{\mathbf{w}\in\mathcal{W}}\langle \boldsymbol{\rho}_{\boldsymbol{\Phi}}(\widehat{\pi}_{\mathrm{E}}) - \boldsymbol{\Phi}^{\mathsf{T}}\mathbf{d}_k, \mathbf{w}\rangle + \langle \boldsymbol{\rho}_{\boldsymbol{\Phi}}(\widehat{\pi}_{\mathrm{E}}) - \boldsymbol{\Phi}^{\mathsf{T}}\mathbf{d}_k, \mathbf{w}_k^{\star} - \bar{\mathbf{w}}_k^{\star}\rangle.$$

Moreover, we have

$$\langle \boldsymbol{\rho}_{\boldsymbol{\Phi}}(\widehat{\pi}_{\mathrm{E}}) - \boldsymbol{\Phi}^{\mathsf{T}}\mathbf{d}_k, \mathbf{w}_k^{\star} - \bar{\mathbf{w}}_k^{\star}\rangle = \max_{\mathbf{w}\in\mathcal{W}}\langle \boldsymbol{\rho}_{\boldsymbol{\Phi}}(\widehat{\pi}_{\mathrm{E}}) - \boldsymbol{\Phi}^{\mathsf{T}}\mathbf{d}_k, \mathbf{w}_k^{\star} - \mathbf{w}\rangle$$
$$= \max_{\mathbf{w}\in\mathcal{W}}\langle \boldsymbol{\rho}_{\boldsymbol{\Phi}}(\widehat{\pi}_{\mathrm{E}}) + \boldsymbol{\Phi}^{\mathsf{T}}\mathbf{d}_k^{\star} - \boldsymbol{\Phi}^{\mathsf{T}}\mathbf{d}_k^{\star} - \boldsymbol{\Phi}^{\mathsf{T}}\mathbf{d}_k, \mathbf{w}_k^{\star} - \mathbf{w}\rangle$$
$$= \max_{\mathbf{w}\in\mathcal{W}}\langle \boldsymbol{\rho}_{\boldsymbol{\Phi}}(\widehat{\pi}_{\mathrm{E}}) - \boldsymbol{\Phi}^{\mathsf{T}}\mathbf{d}_k^{\star}, \mathbf{w}_k^{\star} - \mathbf{w}\rangle + \langle \mathbf{d}_k^{\star} - \mathbf{d}_k, \boldsymbol{\Phi}(\mathbf{w}_k^{\star} - \bar{\mathbf{w}}_k^{\star})\rangle$$
$$\leq \underbrace{\max_{\mathbf{w}\in\mathcal{W}}\langle \boldsymbol{\rho}_{\boldsymbol{\Phi}}(\widehat{\pi}_{\mathrm{E}}) - \boldsymbol{\lambda}_k^{\star}, \mathbf{w}_k^{\star} - \mathbf{w}\rangle}_{:=(A)} + \|\mathbf{d}_k^{\star} - \mathbf{d}_k\|_1 \|\boldsymbol{\Phi}(\mathbf{w}_k^{\star} - \bar{\mathbf{w}}_k^{\star})\|_{\infty}$$
$$\leq 2\,\|\mathbf{d}_k^{\star} - \mathbf{d}_k\|_1.$$

The first equality holds because the term in $\mathbf{w}_k^{\star}$ is a constant wrt $\mathbf{w}$, the variable of the max. In the last inequality follows from (A) being zero as we show next:

$$\max_{\mathbf{w}\in\mathcal{W}}\langle \boldsymbol{\rho}_{\boldsymbol{\Phi}}(\widehat{\pi}_{\mathrm{E}}) - \boldsymbol{\lambda}_k^{\star}, \mathbf{w}_k^{\star} - \mathbf{w}\rangle = \max_{\mathbf{w}\in\mathcal{W}}\langle \boldsymbol{\rho}_{\boldsymbol{\Phi}}(\widehat{\pi}_{\mathrm{E}}) - \boldsymbol{\lambda}_k^{\star}, \mathbf{w}_k^{\star} - \mathbf{w}\rangle + \frac{1}{\eta}D(\boldsymbol{\lambda}_k^{\star}\|\boldsymbol{\Phi}^{\mathsf{T}}\mathbf{d}_{k-1})$$
$$- \frac{1}{\eta}D(\boldsymbol{\lambda}_k^{\star}\|\boldsymbol{\Phi}^{\mathsf{T}}\mathbf{d}_{k-1}) + \frac{1}{\alpha}H(\mathbf{d}_k^{\star}\|\mathbf{d}_{k-1}) - \frac{1}{\alpha}H(\mathbf{d}_k^{\star}\|\mathbf{d}_{k-1})$$
$$= \max_{\mathbf{w}\in\mathcal{W}}\left(\langle \boldsymbol{\lambda}_k^{\star} - \boldsymbol{\rho}_{\boldsymbol{\Phi}}(\widehat{\pi}_{\mathrm{E}}), \mathbf{w}\rangle + \frac{1}{\eta}D(\boldsymbol{\lambda}_k^{\star}\|\boldsymbol{\Phi}^{\mathsf{T}}\mathbf{d}_{k-1})\right.$$
$$\left. + \frac{1}{\alpha}H(\mathbf{d}_k^{\star}\|\mathbf{d}_{k-1})\right) - \langle \boldsymbol{\lambda}_k^{\star} - \boldsymbol{\rho}_{\boldsymbol{\Phi}}(\widehat{\pi}_{\mathrm{E}}), \mathbf{w}_k^{\star}\rangle$$
$$- \frac{1}{\eta}D(\boldsymbol{\lambda}_k^{\star}\|\boldsymbol{\Phi}^{\mathsf{T}}\mathbf{d}_{k-1}) - \frac{1}{\alpha}H(\mathbf{d}_k^{\star}\|\mathbf{d}_{k-1})$$
$$= \max_{\mathbf{w}\in\mathcal{W}}\left(\langle \boldsymbol{\lambda}_k^{\star} - \boldsymbol{\rho}_{\boldsymbol{\Phi}}(\widehat{\pi}_{\mathrm{E}}), \mathbf{w}\rangle + \frac{1}{\eta}D(\boldsymbol{\lambda}_k^{\star}\|\boldsymbol{\Phi}^{\mathsf{T}}\mathbf{d}_{k-1})\right.$$
$$\left. + \frac{1}{\alpha}H(\mathbf{d}_k^{\star}\|\mathbf{d}_{k-1})\right) - \max_{\mathbf{w}\in\mathcal{W}}\min_{\boldsymbol{\lambda},\mathbf{d}\in\mathfrak{M}_{\boldsymbol{\Phi}}}\left(\langle \boldsymbol{\lambda} - \boldsymbol{\rho}_{\boldsymbol{\Phi}}(\widehat{\pi}_{\mathrm{E}}), \mathbf{w}\rangle\right.$$
$$\left. + \frac{1}{\eta}D(\boldsymbol{\lambda}\|\boldsymbol{\Phi}^{\mathsf{T}}\mathbf{d}_{k-1}) + \frac{1}{\alpha}H(\mathbf{d}\|\mathbf{d}_{k-1})\right)$$
$$= 0.$$
$\square$

**Lemma 5** (Lower Bound on feature expectation vectors). *Let Assumption 2 hold. We then have $(\boldsymbol{\Phi}^{\mathsf{T}}\mathbf{d}_k)(j) \geq \beta$ for all $j \in [m]$.*

*Proof.* Let $\mathbf{e}_j \in \mathbb{R}^m$ the vector with zeros everywhere but in position $j$ where it takes the value of 1. Then, we observe that

$$\beta \leq \lambda_{\min}\big(\mathbb{E}_{s,a\sim\mathbf{d}_k}[\boldsymbol{\phi}(s,a)\boldsymbol{\phi}(s,a)^{\mathsf{T}}]\big)$$
$$= \min_{\{\mathbf{x}\in\mathbb{R}^m:\|\mathbf{x}\|_2=1\}}\mathbf{x}^{\mathsf{T}}\mathbb{E}_{s,a\sim\mathbf{d}_k}[\boldsymbol{\phi}(s,a)\boldsymbol{\phi}(s,a)^{\mathsf{T}}]\mathbf{x}$$
$$\leq \mathbf{e}_j^{\mathsf{T}}\mathbb{E}_{s,a\sim\mathbf{d}_k}[\boldsymbol{\phi}(s,a)\boldsymbol{\phi}(s,a)^{\mathsf{T}}]\mathbf{e}_j$$
$$= \mathbb{E}_{s,a\sim\mathbf{d}_k}[\phi_j^2(s,a)] \leq \mathbb{E}_{s,a\sim\mathbf{d}_k}[\phi_j(s,a)] = (\boldsymbol{\Phi}^{\mathsf{T}}\mathbf{d}_k)(j).$$

$\square$

**Theorem 3** (Error propagation with empirical expert feature expectation vector). *Let* $\mathbf{d}^\star = \arg\min_{\mathbf{d}\in\mathfrak{F}} \max_{\mathbf{c}\in\mathcal{C}} \langle \mathbf{d}, \mathbf{c}\rangle - \langle \boldsymbol{\mu}_{\pi_E}, \mathbf{c}\rangle$, *and let* $\boldsymbol{\lambda}^\star$ *be any state-action occupancy measure such that* $(\boldsymbol{\lambda}^\star, \mathbf{d}^\star) \in \mathfrak{M}_{\boldsymbol{\Phi}}$. *Moreover, let* $C \triangleq \frac{1}{\beta\eta}\left(\sqrt{\frac{2\alpha}{1-\gamma}} + \sqrt{8\eta}\right) + \sqrt{\frac{18\alpha}{1-\gamma}}$. *Then, we have that*

$$\frac{1}{K}\sum_k \langle \boldsymbol{\rho}_{\boldsymbol{\Phi}}(\widehat{\pi}_{\mathrm{E}}) - \boldsymbol{\Phi}^\intercal \mathbf{d}^\star, \mathbf{w}_k\rangle - \min_{\mathbf{w}\in\mathcal{W}} \langle \boldsymbol{\rho}_{\boldsymbol{\Phi}}(\widehat{\pi}_{\mathrm{E}}) - \boldsymbol{\Phi}^\intercal \mathbf{d}_k, \mathbf{w}\rangle$$

$$\leq \frac{D(\boldsymbol{\lambda}^\star\|\boldsymbol{\Phi}^\intercal\mathbf{d}_0)}{K\eta} + \frac{H(\mathbf{d}^\star\|\mathbf{d}_0)}{K\alpha} + \frac{C}{K}\sum_k \sqrt{\epsilon_k} + \frac{\sum_k \epsilon_k}{K}.$$

*Proof.* We have that

$$\begin{aligned}
D(\boldsymbol{\lambda}^\star\|\boldsymbol{\lambda}_k) &= D(\boldsymbol{\lambda}^\star\|\boldsymbol{\Phi}^\intercal\mathbf{d}_{k-1}) + \eta\langle\boldsymbol{\lambda}^\star, \mathbf{w}_k + \gamma\mathbf{M}\mathbf{V}_{\boldsymbol{\theta}_k}^k - \boldsymbol{\theta}_k\rangle + \eta\tau_{\boldsymbol{\theta}_k,\mathbf{w}_k}^k \\
&= D(\boldsymbol{\lambda}^\star\|\boldsymbol{\Phi}^\intercal\mathbf{d}_{k-1}) + \eta\langle\boldsymbol{\lambda}^\star, \mathbf{w}_k - \boldsymbol{\theta}_k\rangle + \eta\langle\gamma\mathbf{M}^T\boldsymbol{\lambda}^\star, \mathbf{V}_{\boldsymbol{\theta}_k}^k\rangle + \eta\tau_{\boldsymbol{\theta}_k,\mathbf{w}_k}^k \\
&= D(\boldsymbol{\lambda}^\star\|\boldsymbol{\Phi}^\intercal\mathbf{d}_{k-1}) + \eta\langle\boldsymbol{\lambda}^\star, \mathbf{w}_k - \boldsymbol{\theta}_k\rangle + \eta\langle\mathbf{B}^\intercal\mathbf{d}^\star, \mathbf{V}_{\boldsymbol{\theta}_k}^k\rangle - \eta(1-\gamma)\langle\boldsymbol{\nu}_0, \mathbf{V}_{\boldsymbol{\theta}_k}^k\rangle \\
&\quad + \eta\tau_{\boldsymbol{\theta}_k,\mathbf{w}_k}^k \\
&= D(\boldsymbol{\lambda}^\star\|\boldsymbol{\Phi}^\intercal\mathbf{d}_{k-1}) + \eta\langle\boldsymbol{\lambda}^\star, \mathbf{w}_k - \boldsymbol{\theta}_k\rangle + \eta\langle\mathbf{B}^\intercal\mathbf{d}^\star, \mathbf{V}_{\boldsymbol{\theta}_k}^k\rangle - \eta(1-\gamma)\langle\boldsymbol{\nu}_0, \mathbf{V}_{\boldsymbol{\theta}_k}^k\rangle \\
&\quad + \eta\tau_{\boldsymbol{\theta}_k,\mathbf{w}_k}^k \\
&= D(\boldsymbol{\lambda}^\star\|\boldsymbol{\Phi}^\intercal\mathbf{d}_{k-1}) + \eta\langle\boldsymbol{\lambda}^\star, \mathbf{w}_k - \boldsymbol{\theta}_k\rangle + \eta\langle\mathbf{B}^\intercal\mathbf{d}^\star, \mathbf{V}_{\boldsymbol{\theta}_k}^k\rangle - \eta\mathcal{G}_k(\boldsymbol{\theta}_k, \mathbf{w}_k) \\
&\quad - \eta\langle\boldsymbol{\rho}_{\boldsymbol{\Phi}}(\widehat{\pi}_{\mathrm{E}}), \mathbf{w}_k\rangle \\
&\leq D(\boldsymbol{\lambda}^\star\|\boldsymbol{\Phi}^\intercal\mathbf{d}_{k-1}) + \eta\langle\boldsymbol{\lambda}^\star, \mathbf{w}_k - \boldsymbol{\theta}_k\rangle + \eta\langle\mathbf{B}^\intercal\mathbf{d}^\star, \mathbf{V}_{\boldsymbol{\theta}_k}^k\rangle - \eta\mathcal{G}_k(\boldsymbol{\theta}_k^\star, \mathbf{w}_k^\star) \\
&\quad + \eta\epsilon_k - \eta\langle\boldsymbol{\rho}_{\boldsymbol{\Phi}}(\widehat{\pi}_{\mathrm{E}}), \mathbf{w}_k\rangle \\
&\leq D(\boldsymbol{\lambda}^\star\|\boldsymbol{\Phi}^\intercal\mathbf{d}_{k-1}) + \eta\langle\boldsymbol{\lambda}^\star, \mathbf{w}_k - \boldsymbol{\theta}_k\rangle + \eta\langle\mathbf{B}^\intercal\mathbf{d}^\star, \mathbf{V}_{\boldsymbol{\theta}_k}^k\rangle + \eta\langle\boldsymbol{\rho}_{\boldsymbol{\Phi}}(\widehat{\pi}_{\mathrm{E}}) - \boldsymbol{\lambda}_k^\star, \mathbf{w}_k^\star\rangle \\
&\quad - D(\boldsymbol{\lambda}_k^\star\|\boldsymbol{\Phi}^\intercal\mathbf{d}_{k-1}) - \eta\frac{H(\mathbf{d}_k^\star\|\mathbf{d}_{k-1})}{\alpha} + \eta\epsilon_k - \eta\langle\boldsymbol{\rho}_{\boldsymbol{\Phi}}(\widehat{\pi}_{\mathrm{E}}), \mathbf{w}_k\rangle \\
&\leq D(\boldsymbol{\lambda}^\star\|\boldsymbol{\Phi}^\intercal\mathbf{d}_{k-1}) + \eta\langle\boldsymbol{\lambda}^\star, \mathbf{w}_k\rangle + \eta\langle\mathbf{d}^\star, \mathbf{B}\mathbf{V}_{\boldsymbol{\theta}_k}^k - \boldsymbol{\Phi}\boldsymbol{\theta}_k\rangle + \eta\langle\boldsymbol{\rho}_{\boldsymbol{\Phi}}(\widehat{\pi}_{\mathrm{E}}) - \boldsymbol{\lambda}_k^\star, \mathbf{w}_k^\star\rangle \\
&\quad + \eta\epsilon_k - \eta\langle\boldsymbol{\rho}_{\boldsymbol{\Phi}}(\widehat{\pi}_{\mathrm{E}}), \mathbf{w}_k\rangle \\
&\leq D(\boldsymbol{\lambda}^\star\|\boldsymbol{\Phi}^\intercal\mathbf{d}_{k-1}) + \eta\langle\mathbf{d}^\star, \boldsymbol{\Phi}\mathbf{w}_k\rangle + \eta\langle\mathbf{d}^\star, \mathbf{B}\mathbf{V}_{\boldsymbol{\theta}_k}^k - \boldsymbol{\Phi}\boldsymbol{\theta}_k\rangle + \eta\langle\boldsymbol{\rho}_{\boldsymbol{\Phi}}(\widehat{\pi}_{\mathrm{E}}) - \boldsymbol{\Phi}^\intercal\mathbf{d}_k, \mathbf{w}_k^\star\rangle \\
&\quad + \eta\langle\mathbf{d}_k - \mathbf{d}_k^\star, \boldsymbol{\Phi}\mathbf{w}_k^\star\rangle + \eta\epsilon_k - \eta\langle\boldsymbol{\rho}_{\boldsymbol{\Phi}}(\widehat{\pi}_{\mathrm{E}}), \mathbf{w}_k\rangle \\
&\leq D(\boldsymbol{\lambda}^\star\|\boldsymbol{\Phi}^\intercal\mathbf{d}_{k-1}) + \eta\langle\mathbf{d}^\star, \boldsymbol{\Phi}\mathbf{w}_k\rangle + \eta\langle\mathbf{d}^\star, \mathbf{B}\mathbf{V}_{\boldsymbol{\theta}_k}^k - \boldsymbol{\Phi}\boldsymbol{\theta}_k\rangle + \eta\langle\boldsymbol{\rho}_{\boldsymbol{\Phi}}(\widehat{\pi}_{\mathrm{E}}) - \boldsymbol{\Phi}^\intercal\mathbf{d}_k, \mathbf{w}_k^\star\rangle \\
&\quad + \eta\|\mathbf{d}_k - \mathbf{d}_k^\star\|_1 + \eta\epsilon_k - \eta\langle\boldsymbol{\rho}_{\boldsymbol{\Phi}}(\widehat{\pi}_{\mathrm{E}}), \mathbf{w}_k\rangle \quad \text{Using Lemma 4} \\
&\leq D(\boldsymbol{\lambda}^\star\|\boldsymbol{\Phi}^\intercal\mathbf{d}_{k-1}) + \eta\langle\mathbf{d}^\star, \boldsymbol{\Phi}\mathbf{w}_k\rangle + \eta\langle\mathbf{d}^\star, \mathbf{B}\mathbf{V}_{\boldsymbol{\theta}_k}^k - \boldsymbol{\Phi}\boldsymbol{\theta}_k\rangle \\
&\quad + \min_{\mathbf{w}\in\mathcal{W}} \eta\langle\boldsymbol{\rho}_{\boldsymbol{\Phi}}(\widehat{\pi}_{\mathrm{E}}) - \boldsymbol{\Phi}^\intercal\mathbf{d}_k, \mathbf{w}\rangle + 3\eta\|\mathbf{d}_k - \mathbf{d}_k^\star\|_1 + \eta\epsilon_k - \eta\langle\boldsymbol{\rho}_{\boldsymbol{\Phi}}(\widehat{\pi}_{\mathrm{E}}), \mathbf{w}_k\rangle.
\end{aligned}$$

Therefore, it follows that

$$\begin{aligned}
\langle\boldsymbol{\rho}_{\boldsymbol{\Phi}}(\widehat{\pi}_{\mathrm{E}}) - \boldsymbol{\Phi}^\intercal\mathbf{d}^\star, \mathbf{w}_k\rangle - \min_{\mathbf{w}\in\mathcal{W}}\langle\boldsymbol{\rho}_{\boldsymbol{\Phi}}(\widehat{\pi}_{\mathrm{E}}) - \boldsymbol{\Phi}^\intercal\mathbf{d}_k, \mathbf{w}\rangle &\leq \frac{D(\boldsymbol{\lambda}^\star\|\boldsymbol{\Phi}^\intercal\mathbf{d}_{k-1}) - D(\boldsymbol{\lambda}^\star\|\boldsymbol{\lambda}_k)}{\eta} \\
&\quad + \langle\mathbf{d}^\star, \mathbf{B}\mathbf{V}_{\boldsymbol{\theta}_k}^k - \boldsymbol{\Phi}\boldsymbol{\theta}_k\rangle + 3\|\mathbf{d}_k - \mathbf{d}_k^\star\|_1 + \epsilon_k. \quad (20)
\end{aligned}$$

Then, by using $H(\mathbf{d}^\star\|\mathbf{d}_k) = H(\mathbf{d}^\star\|\mathbf{d}_{k-1}) - \alpha\langle\mathbf{d}^\star, \boldsymbol{\Phi}\boldsymbol{\theta}_k - \mathbf{B}\mathbf{V}_{\boldsymbol{\theta}_k}^k\rangle$, we obtain

$$\begin{aligned}
\langle\boldsymbol{\rho}_{\boldsymbol{\Phi}}(\widehat{\pi}_{\mathrm{E}}) - \boldsymbol{\Phi}^\intercal\mathbf{d}^\star, \mathbf{w}_k\rangle - \min_{\mathbf{w}\in\mathcal{W}}\langle\boldsymbol{\rho}_{\boldsymbol{\Phi}}(\widehat{\pi}_{\mathrm{E}}) - \boldsymbol{\Phi}^\intercal\mathbf{d}_k, \mathbf{w}\rangle &\leq \frac{D(\boldsymbol{\lambda}^\star\|\boldsymbol{\Phi}^\intercal\mathbf{d}_{k-1}) - D(\boldsymbol{\lambda}^\star\|\boldsymbol{\lambda}_k)}{\eta} \\
&\quad + \frac{H(\mathbf{d}^\star\|\mathbf{d}_{k-1}) - H(\mathbf{d}^\star\|\mathbf{d}_k)}{\alpha} \\
&\quad + 3\|\mathbf{d}_k - \mathbf{d}_k^\star\|_1 + \epsilon_k.
\end{aligned}$$

Summing over iteration indices $k$ and dividing by the total number of iterations $K$, we obtain

$$\frac{1}{K}\sum_k \langle \boldsymbol{\rho_\Phi}(\widehat{\pi}_E) - \boldsymbol{\Phi^\mathsf{T}} \mathbf{d}^\star, \mathbf{w}_k \rangle - \min_{\mathbf{w}\in\mathcal{W}} \langle \boldsymbol{\rho_\Phi}(\widehat{\pi}_E) - \boldsymbol{\Phi^\mathsf{T}} \mathbf{d}_k, \mathbf{w} \rangle \leq \frac{1}{K}\sum_k \left( \frac{D(\boldsymbol{\lambda}^\star || \boldsymbol{\Phi^\mathsf{T}} \mathbf{d}_{k-1})}{\eta} \right.$$
$$\left. - \frac{D(\boldsymbol{\lambda}^\star || \boldsymbol{\lambda}_k)}{\eta} + \frac{H(\mathbf{d}^\star || \mathbf{d}_{k-1}) - H(\mathbf{d}^\star || \mathbf{d}_k)}{\alpha} + 3 \left\| \mathbf{d}_k - \mathbf{d}_k^\star \right\|_1 \right) + \frac{\sum_k \epsilon_k}{K}. \quad (21)$$

Moreover, by a telescoping sum, we get

$$\sum_k \left( \frac{D(\boldsymbol{\lambda}^\star || \boldsymbol{\Phi^\mathsf{T}} \mathbf{d}_{k-1}) - D(\boldsymbol{\lambda}^\star || \boldsymbol{\lambda}_k)}{\eta} + \frac{H(\mathbf{d}^\star || \mathbf{d}_{k-1}) - H(\mathbf{d}^\star || \mathbf{d}_k)}{\alpha} \right)$$
$$= \sum_k \left( \frac{D(\boldsymbol{\lambda}^\star || \boldsymbol{\Phi^\mathsf{T}} \mathbf{d}_{k-1}) - D(\boldsymbol{\lambda}^\star || \boldsymbol{\Phi^\mathsf{T}} \mathbf{d}_k)}{\eta} + \frac{D(\boldsymbol{\lambda}^\star || \boldsymbol{\Phi^\mathsf{T}} \mathbf{d}_k) - D(\boldsymbol{\lambda}^\star || \boldsymbol{\lambda}_k)}{\eta} \right.$$
$$\left. + \frac{H(\mathbf{d}^\star || \mathbf{d}_{k-1}) - H(\mathbf{d}^\star || \mathbf{d}_k)}{\alpha} \right)$$
$$= \frac{D(\boldsymbol{\lambda}^\star || \boldsymbol{\Phi^\mathsf{T}} \mathbf{d}_0) - D(\boldsymbol{\lambda}^\star || \boldsymbol{\Phi^\mathsf{T}} \mathbf{d}_K)}{\eta} + \frac{H(\mathbf{d}^\star || \mathbf{d}_0) - H(\mathbf{d}^\star || \mathbf{d}_K)}{\alpha}$$
$$+ \sum_k \frac{D(\boldsymbol{\lambda}^\star || \boldsymbol{\Phi^\mathsf{T}} \mathbf{d}_k) - D(\boldsymbol{\lambda}^\star || \boldsymbol{\lambda}_k)}{\eta}$$
$$\leq \frac{D(\boldsymbol{\lambda}^\star || \boldsymbol{\Phi^\mathsf{T}} \mathbf{d}_0)}{\eta} + \frac{H(\mathbf{d}^\star || \mathbf{d}_0)}{\alpha} + \sum_k \frac{D(\boldsymbol{\lambda}^\star || \boldsymbol{\Phi^\mathsf{T}} \mathbf{d}_k) - D(\boldsymbol{\lambda}^\star || \boldsymbol{\lambda}_k)}{\eta}$$

Combining this derivation with (21), we get

$$\frac{1}{K}\sum_k \langle \boldsymbol{\rho_\Phi}(\widehat{\pi}_E) - \boldsymbol{\Phi^\mathsf{T}} \mathbf{d}^\star, \mathbf{w}_k \rangle - \min_{\mathbf{w}\in\mathcal{W}} \langle \boldsymbol{\rho_\Phi}(\widehat{\pi}_E) - \boldsymbol{\Phi^\mathsf{T}} \mathbf{d}_k, \mathbf{w} \rangle \leq \frac{D(\boldsymbol{\lambda}^\star || \boldsymbol{\lambda}_0)}{K\eta} + \frac{H(\mathbf{d}^\star || \mathbf{d}_0)}{K\alpha}$$
$$+ \frac{1}{K}\sum_k \left( \frac{D(\boldsymbol{\lambda}^\star || \boldsymbol{\Phi^\mathsf{T}} \mathbf{d}_k) - D(\boldsymbol{\lambda}^\star || \boldsymbol{\lambda}_k)}{\eta} + 3 \left\| \mathbf{d}_k - \mathbf{d}_k^\star \right\|_1 \right) + \frac{\sum_k \epsilon_k}{K}. \quad (22)$$

In order to bound the term $D(\boldsymbol{\lambda}^\star || \boldsymbol{\Phi^\mathsf{T}} \mathbf{d}_k) - D(\boldsymbol{\lambda}^\star || \boldsymbol{\lambda}_k)$, we introduce the Bregman projection to the space of feature expectation vectors induced by valid occupancy measures $\tilde{\boldsymbol{\lambda}}_k = \arg\min_{\{\lambda = \Phi \mathbf{d} | \mathbf{d} \in \mathcal{F}\}} D(\boldsymbol{\lambda} || \boldsymbol{\lambda}_k)$. We then have

$$D(\boldsymbol{\lambda}^\star || \boldsymbol{\Phi^\mathsf{T}} \mathbf{d}_k) - D(\boldsymbol{\lambda}^\star || \boldsymbol{\lambda}_k) = D(\boldsymbol{\lambda}^\star || \boldsymbol{\Phi^\mathsf{T}} \mathbf{d}_k) - D(\boldsymbol{\lambda}^\star || \boldsymbol{\lambda}_k) + D(\boldsymbol{\lambda}^\star || \tilde{\boldsymbol{\lambda}}_k) - D(\boldsymbol{\lambda}^\star || \tilde{\boldsymbol{\lambda}}_k)$$
$$\leq D(\boldsymbol{\lambda}^\star || \boldsymbol{\Phi^\mathsf{T}} \mathbf{d}_k) - D(\boldsymbol{\lambda}^\star || \tilde{\boldsymbol{\lambda}}_k) - D(\tilde{\boldsymbol{\lambda}}_k || \boldsymbol{\lambda}_k)$$
$$\leq D(\boldsymbol{\lambda}^\star || \boldsymbol{\Phi^\mathsf{T}} \mathbf{d}_k) - D(\boldsymbol{\lambda}^\star || \tilde{\boldsymbol{\lambda}}_k),$$

where in the second inequality, we used Lemma 11.3 in [24]. Furthermore,

$$
\begin{aligned}
D(\boldsymbol{\lambda}^\star || \boldsymbol{\Phi}^\mathsf{T} \mathbf{d}_k) - D(\boldsymbol{\lambda}^\star || \tilde{\boldsymbol{\lambda}}_k) &= \sum_{i=1}^{m} \boldsymbol{\lambda}^\star(i) \log \frac{\tilde{\boldsymbol{\lambda}}_k(i)}{\boldsymbol{\Phi}^\mathsf{T} \mathbf{d}_k(i)} \\
&\leq \sum_{i=1}^{m} \boldsymbol{\lambda}^\star(i) \left( \frac{\tilde{\boldsymbol{\lambda}}_k(i)}{\boldsymbol{\Phi}^\mathsf{T} \mathbf{d}_k(i)} - 1 \right) \\
&\leq \sum_{i=1}^{m} \frac{\boldsymbol{\lambda}^\star(i)}{\boldsymbol{\Phi}^\mathsf{T} \mathbf{d}_k(i)} \left| \tilde{\boldsymbol{\lambda}}_k(i) - \boldsymbol{\Phi}^\mathsf{T} \mathbf{d}_k(i) \right| \\
&\leq \max_i \frac{\boldsymbol{\lambda}^\star(i)}{\boldsymbol{\Phi}^\mathsf{T} \mathbf{d}_k(i)} \left\| \tilde{\boldsymbol{\lambda}}_k - \boldsymbol{\Phi}^\mathsf{T} \mathbf{d}_k \right\|_1 \\
&\leq \frac{1}{\beta} \left\| \tilde{\boldsymbol{\lambda}}_k - \boldsymbol{\Phi}^\mathsf{T} \mathbf{d}_k \right\|_1 \\
&\leq \frac{1}{\beta} ( \left\| \tilde{\boldsymbol{\lambda}}_k - \boldsymbol{\lambda}_k \right\|_1 + \| \boldsymbol{\lambda}_k^\star - \boldsymbol{\lambda}_k \|_1 + \| \boldsymbol{\lambda}_k^\star - \boldsymbol{\Phi}^\mathsf{T} \mathbf{d}_k \|_1 ) \\
&\leq \frac{1}{\beta} ( \sqrt{2 D(\tilde{\boldsymbol{\lambda}}_k || \boldsymbol{\lambda}_k)} + \sqrt{2 D(\boldsymbol{\lambda}_k^\star || \boldsymbol{\lambda}_k)} + \| \boldsymbol{\lambda}_k^\star - \boldsymbol{\Phi}^\mathsf{T} \mathbf{d}_k \|_1 ) \\
&\leq \frac{1}{\beta} ( 2 \sqrt{2 D(\boldsymbol{\lambda}_k^\star || \boldsymbol{\lambda}_k)} + \| \boldsymbol{\Phi}^\mathsf{T} \mathbf{d}_k^\star - \boldsymbol{\Phi}^\mathsf{T} \mathbf{d}_k \|_1 ) \\
&\leq \frac{1}{\beta} \left( \sqrt{8\eta(\epsilon_k + \langle \boldsymbol{\rho}_{\boldsymbol{\Phi}}(\widehat{\pi}_{\mathrm{E}}) - \boldsymbol{\Phi}^\mathsf{T} \mathbf{d}_k^\star, \mathbf{w}_k^\star - \mathbf{w}_k \rangle)} \right. \\
&\qquad \left. + \| \boldsymbol{\Phi} \|_\infty \| \mathbf{d}_k^\star - \mathbf{d}_k \|_1 \right),
\end{aligned}
$$

where we used $\max_i \frac{\boldsymbol{\lambda}^\star(i)}{\boldsymbol{\Phi}^\mathsf{T} \mathbf{d}_k(i)} \leq \frac{1}{\beta}$ thanks to Lemma 5 while in the last line we use the fact that $H(\mathbf{d}_k^\star || \mathbf{d}_k)$ is positive and the equality in Lemma 2. To bound the $\ell_1$-norm, we apply Pinkser's inequality and Lemma 2 in [14] to get that

$$
\| \mathbf{d}_k - \mathbf{d}_k^\star \| \leq \sqrt{2 D(\mathbf{d}_k || \mathbf{d}_k^\star)} \leq \sqrt{2 \frac{H(\mathbf{d}_k || \mathbf{d}_k^\star)}{1 - \gamma}} \leq \sqrt{\frac{2\alpha}{1 - \gamma} (\epsilon_k + \langle \boldsymbol{\rho}_{\boldsymbol{\Phi}}(\widehat{\pi}_{\mathrm{E}}) - \boldsymbol{\Phi}^\mathsf{T} \mathbf{d}_k^\star, \mathbf{w}_k^\star - \mathbf{w}_k \rangle)}.
$$

Plugging the last derivation in Equation (22) gives

$$
\begin{aligned}
\frac{1}{K} \sum_k \langle \boldsymbol{\rho}_{\boldsymbol{\Phi}}(\widehat{\pi}_{\mathrm{E}}) - \mathbf{d}^\star, \boldsymbol{\Phi} \mathbf{w}_k \rangle - \min_{\mathbf{w} \in \mathcal{W}} \langle \boldsymbol{\rho}_{\boldsymbol{\Phi}}(\widehat{\pi}_{\mathrm{E}}) - \mathbf{d}_k, \boldsymbol{\Phi} \mathbf{w} \rangle &\leq \frac{D(\boldsymbol{\lambda}^\star || \boldsymbol{\lambda}_0)}{K\eta} + \frac{H(\mathbf{d}^\star || \mathbf{d}_0)}{K\alpha} \\
&\quad + \frac{C}{K} \sum_k \left( \sqrt{\epsilon_k + \langle \boldsymbol{\rho}_{\boldsymbol{\Phi}}(\widehat{\pi}_{\mathrm{E}}) - \boldsymbol{\lambda}_k^\star, \mathbf{w}_k^\star - \mathbf{w}_k \rangle} \right) + \frac{\sum_k \epsilon_k}{K}. \quad (23)
\end{aligned}
$$

Finally, using Lemma 3 we have that the term $\langle \boldsymbol{\rho}_{\boldsymbol{\Phi}}(\widehat{\pi}_{\mathrm{E}}) - \boldsymbol{\lambda}_k^\star, \mathbf{w}_k^\star - \mathbf{w}_k \rangle$ is non positive. Therefore,

$$
\begin{aligned}
\frac{1}{K} \sum_k \langle \boldsymbol{\rho}_{\boldsymbol{\Phi}}(\widehat{\pi}_{\mathrm{E}}) - \boldsymbol{\Phi}^\mathsf{T} \mathbf{d}^\star, \mathbf{w}_k \rangle - \min_{\mathbf{w} \in \mathcal{W}} \langle \boldsymbol{\rho}_{\boldsymbol{\Phi}}(\widehat{\pi}_{\mathrm{E}}) - \boldsymbol{\Phi}^\mathsf{T} \mathbf{d}_k, \mathbf{w} \rangle &\leq \frac{D(\boldsymbol{\lambda}^\star || \boldsymbol{\lambda}_0)}{K\eta} + \frac{H(\mathbf{d}^\star || \mathbf{d}_0)}{K\alpha} \\
&\quad + \frac{C}{K} \sum_k \sqrt{\epsilon_k} + \frac{\sum_k \epsilon_k}{K},
\end{aligned}
$$

where $C = \frac{1}{\beta\eta}(\sqrt{\frac{2\alpha}{1-\gamma}} + \sqrt{8\eta}) + 3\sqrt{\frac{2\alpha}{1-\gamma}}$. $\qquad\square$

Finally, we need a Lemma that provides a concentration for the estimated expert feature expectation vector.

**Lemma 6** ([111]). *Let $\mathcal{D}_{\pi_E} \triangleq \{(s_0^\ell, a_0^\ell, s_1^\ell, a_1^\ell, \ldots, s_H^\ell, a_H^\ell)\}_{\ell=1}^{n_\mathrm{E}} \sim \pi_E$ be a finite set of i.i.d. truncated sample trajectories. We consider the empirical expert feature expectation vector $\boldsymbol{\rho}_{\boldsymbol{\Phi}}(\widehat{\pi}_E)$*

*by taking sample averages, i.e.,*

$$\boldsymbol{\rho}_{\boldsymbol{\Phi}}(\widehat{\pi_E}) \triangleq (1-\gamma)\frac{1}{n_{\mathrm{E}}}\sum_{t=0}^{H}\sum_{\ell=1}^{N}\gamma^t\phi_i(s_t^\ell, a_t^\ell), \ \forall \ i \in [m].$$

*Suppose the trajectory length is $H \geq \frac{1}{1-\gamma}\log(\frac{1}{\varepsilon})$, and the number of of expert trajectories is $n_{\mathrm{E}} \geq \frac{2\log(\frac{2m}{\delta})}{\varepsilon^2}$. Then, with probability at least $1-\delta$, it holds that $\|\boldsymbol{\rho}_{\boldsymbol{\Phi}}(\pi_E) - \boldsymbol{\rho}_{\boldsymbol{\Phi}}(\widehat{\pi_E})\|_\infty \leq \varepsilon$.*

At this point, Theorem 1 is proven from the results of Theorem 3,Lemma 6 and Lemma 1.

## H   Biased Stochastic Gradients and their Properties

In order to estimate the gradient $\nabla_{\boldsymbol{\theta}} G(\mathbf{w}, \boldsymbol{\theta})$, we define the policy $\pi_{k,\boldsymbol{\theta}}(a|s) \propto \pi_k(a|s)e^{-\alpha Q_{\boldsymbol{\theta}}(s,a)}$, for all $k \in \mathbb{N}$, and for all $\boldsymbol{\theta} \in \mathbb{R}^m$. Then, by standard computations we get that for all $(\mathbf{w}, \boldsymbol{\theta})$, and for all $j \in [m]$,

$\nabla_{\boldsymbol{\theta},j} G(\mathbf{w}, \boldsymbol{\theta})$
$$= \sum_{i=1}^m (\boldsymbol{\Phi}^\mathsf{T}\mathbf{d}_{k-1})(i)\mathbf{B}_{\mathbf{w},\boldsymbol{\theta}}^k(i)\big[\gamma\boldsymbol{\Gamma}_k(i,j) - \mathbb{1}\{i=j\}\big] + (1-\gamma)\sum_s \boldsymbol{\nu}_0(s)\sum_a \pi_{k-1,\boldsymbol{\theta}}(a|s)\phi_i(s,a)$$
$$= \mathbb{E}_{(s,a)\sim\mathbf{d}_{k-1}, i\sim\phi(s,a)}\Big[\mathbf{B}_{\mathbf{w},\boldsymbol{\theta}}^k(i)\big[\gamma\boldsymbol{\Gamma}_k(i,j) - \mathbb{1}\{i=j\}\big]\Big] + (1-\gamma)\,\mathbb{E}_{s_0\sim\boldsymbol{\nu}_0, a_0\sim\pi_{k-1,\boldsymbol{\theta}}(\cdot|s_0)}\Big[\phi_i(s_0, a_0)\Big],$$

where $\mathbf{B}_{\mathbf{w},\boldsymbol{\theta}}^k(i) \triangleq \frac{\exp\left(-\eta\boldsymbol{\delta}_{\mathbf{w},\boldsymbol{\theta}}^k(i)\right)}{Z_k}$, $Z_k \triangleq \sum_{i=1}^m \exp\left(-\eta\boldsymbol{\delta}_{\mathbf{w},\boldsymbol{\theta}}^k(i)\right)\boldsymbol{\rho}_{\boldsymbol{\Phi}}(\pi_{k-1})(i)$, and $\boldsymbol{\Gamma}_k(i,j) \triangleq \sum_{s',a'} \mathbf{M}_{i,s'}\pi_{k-1,\boldsymbol{\theta}}(a'|s')\phi_j(s',a')$, . Similarly, for the gradient $\nabla_{\mathbf{w}} G(\mathbf{w}, \boldsymbol{\theta})$, we can write

$$\nabla_{\mathbf{w},j} G(\mathbf{w}, \boldsymbol{\theta}) = -\boldsymbol{\rho}_{\boldsymbol{\Phi}}(\widehat{\pi_{\mathrm{E}}})(j) + \sum_{i=1}^m (\boldsymbol{\Phi}^\mathsf{T}\mathbf{d}_{k-1})(i)\mathbf{B}_{\mathbf{w},\boldsymbol{\theta}}^k(i)\mathbb{1}\{i=j\}$$
$$= -\boldsymbol{\rho}_{\boldsymbol{\Phi}}(\widehat{\pi_{\mathrm{E}}})(j) + \mathbb{E}_{(s,a)\sim\mathbf{d}_{k-1}, i\sim\phi(s,a)}\Big[\mathbf{B}_{\mathbf{w},\boldsymbol{\theta}}^k(i)\mathbb{1}\{i=j\}\Big]$$

Note that the following estimators of $\nabla_{\boldsymbol{\theta}} G_k(\mathbf{w}, \boldsymbol{\theta})$ and $\nabla_{\mathbf{w}} G_k(\mathbf{w}, \boldsymbol{\theta})$ are unbiased: Sample $(s',a') \sim \mathbf{d}_{k-1}$, $i' \sim \phi(s',a')$, $s_0 \sim \boldsymbol{\nu}_0$, and $a_0 \sim \pi_{k-1,\boldsymbol{\theta}}(\cdot|s_0)$, then define

$$\widetilde{\nabla}_{\mathbf{w},j}\mathcal{G}_k(\mathbf{w}, \boldsymbol{\theta}) = -\boldsymbol{\rho}_{\boldsymbol{\Phi}}(\widehat{\pi_{\mathrm{E}}})(j) + \mathbf{B}_{\mathbf{w},\boldsymbol{\theta}}^k(i')\mathbb{1}\{i'=j\}, \tag{24}$$
$$\widetilde{\nabla}_{\boldsymbol{\theta},j}\mathcal{G}_k(\mathbf{w}, \boldsymbol{\theta}) = \mathbf{B}_{\mathbf{w},\boldsymbol{\theta}}^k(i')\Big[\gamma\boldsymbol{\Gamma}_k(i',j) - \mathbb{1}\{i'=j\}\Big] + (1-\gamma)\phi_j(s_0, a_0). \tag{25}$$

These expressions give rise to the Biased Stochastic Gradient Estimator subroutine (BSGE) given in Algorithm 2, where we plug-in estimators $\widehat{\mathbf{B}}_{\mathbf{w},\boldsymbol{\theta}}^k \in \mathbb{R}^m$ and $\widehat{\boldsymbol{\Gamma}}_k \in \mathbb{R}^{m\times m}$ to Equations (24) and (25). It remains to show how to maintain good estimators $\widehat{\mathbf{B}}_{\mathbf{w},\boldsymbol{\theta}}^k$ and $\widehat{\boldsymbol{\Gamma}}_k$ by using the linear MDP Assumption 1. While the estimator $\widehat{\boldsymbol{\Gamma}}_k \in \mathbb{R}^m$ is a standard ridge regression estimator, the construction of $\widehat{\mathbf{B}}_{\mathbf{w},\boldsymbol{\theta}}^k$ is more involved. In particular, we first need to build an estimator for the product $\mathbf{MV}_{\boldsymbol{\theta}}^k$ via ridge regression. Then, the estimator for $\widehat{\mathbf{B}}_{\mathbf{w},\boldsymbol{\theta}}^k$ is derived by plugging-in the estimator of $\mathbf{MV}_{\boldsymbol{\theta}}^k$, and the estimator for the feature expectation vector $\boldsymbol{\rho}_{\boldsymbol{\Phi}}(\pi_{k-1})$ to equation $\mathbf{B}_{\mathbf{w},\boldsymbol{\theta}}^k(i) \triangleq \frac{\exp\left(-\eta\boldsymbol{\delta}_{\mathbf{w},\boldsymbol{\theta}}^k(i)\right)}{Z_k}$. The reasoning and analysis is inspired by [52, 89].

### H.1   Ridge estimators

This section leverages ridge regression [53] to build estimators $\widehat{\mathbf{B}}_{\mathbf{w},\boldsymbol{\theta}}^k$ and $\widehat{\boldsymbol{\Gamma}}_k \in \mathbb{R}^{m\times m}$. We work under the Assumption 2 which ensures that every iterate covers the features space. We recall that by Lemma 5, Assumption 2 implies that $\boldsymbol{\Phi}^\mathsf{T}\mathbf{d}_k(s,a) \geq \beta$, for all $k \in [K]$.

---
**Algorithm 2** Biased Stochastic Gradient Estimator: BSGE($k, \mathbf{w}, \boldsymbol{\theta}, N$)
---

**Input:** Policy evaluation step $k$, reference points $(\mathbf{w}, \boldsymbol{\theta})$, number of samples $N$

Compute empirical estimators $\widehat{\boldsymbol{\delta}}^{\,k}_{\mathbf{w},\boldsymbol{\theta}} \in \mathbb{R}^m$, $\widehat{\boldsymbol{\Gamma}}_k \in \mathbb{R}^{m\times m}$, $\boldsymbol{\rho}_{\boldsymbol{\Phi}}(\widehat{\pi_{k-1}}) \in \mathbb{R}^m$ using the first $N$ samples $\{(s^{(n)}_{k-1}, a^{(n)}_{k-1}, s'^{(n)}_{k-1})\}^N_{n=1}$ from the buffer $\mathcal{B}_k$

**for** $i = 1, \ldots, m$ **do**

$\quad$ Compute $\widehat{\mathbf{B}}^k_{\mathbf{w},\boldsymbol{\theta}}(i) = \dfrac{\exp\left(-\eta\widehat{\boldsymbol{\delta}}^{\,k}_{\mathbf{w},\boldsymbol{\theta}}(i)\right)}{\widehat{Z}_k}$, Where $\widehat{Z}_k = \sum^m_{i=1}\exp\left(-\eta\widehat{\boldsymbol{\delta}}^{\,k}_{\mathbf{w},\boldsymbol{\theta}}(i)\right)\boldsymbol{\rho}_{\boldsymbol{\Phi}}(\widehat{\pi_{k-1}})(i)$

**end for**

Sample $(s^{(N+1)}_{k-1}, a^{(N+1)}_{k-1}) \sim \boldsymbol{\mu}_{\pi_{k-1}}, i^{(N+1)}_{k-1} \sim \boldsymbol{\phi}(s^{(N+1)}_{k-1}, a^{(N+1)}_{k-1})$

Sample $s^{(0)}_{k-1} \sim \boldsymbol{\nu}_0$, and $a^{(0)}_{k-1} \sim \pi_{k-1,\boldsymbol{\theta}}(\cdot|s_0)$

Compute

$$\widehat{\nabla}_{\mathbf{w},j}\mathcal{G}_k(\mathbf{w}, \boldsymbol{\theta}) = -\boldsymbol{\rho}_{\boldsymbol{\Phi}}(\widehat{\pi_{\mathrm{E}}})(j) + \widehat{\mathbf{B}}^k_{\mathbf{w},\boldsymbol{\theta}}(i^{(N+1)}_{k-1})\mathbb{1}\{i^{(N+1)}_{k-1} = j\}$$

$$\widehat{\nabla}_{\boldsymbol{\theta},j}\mathcal{G}_k(\mathbf{w}, \boldsymbol{\theta}) = \widehat{\mathbf{B}}^k_{\mathbf{w},\boldsymbol{\theta}}(i^{(N+1)}_{k-1})\left[\gamma\widehat{\boldsymbol{\Gamma}}_k(i^{(N+1)}_{k-1}, j) - \mathbb{1}\{i^{(N+1)}_{k-1} = j\}\right] + (1-\gamma)\boldsymbol{\phi}_j(s^{(0)}_{k-1}, a^0_{k-1})$$

**Output:** $(\widehat{\nabla}_{\mathbf{w}}\mathcal{G}_k(\mathbf{w}, \boldsymbol{\theta}), \widehat{\nabla}_{\boldsymbol{\theta}}\mathcal{G}_k(\mathbf{w}, \boldsymbol{\theta}))$

---

### H.1.1 Estimator for $\mathbf{MV}^k_\theta$

We first construct an estimator for $\mathbf{M}_k\mathbf{V}^k_\theta$. We can start noticing that we can rewrite $\mathbf{M}_k\mathbf{V}^k_\theta$ using the feature covariance matrix $\bar{\boldsymbol{\Lambda}}_k \triangleq \mathbb{E}_{(s,a)\sim\mathbf{d}_{k-1}}\left[\boldsymbol{\phi}(s,a)\boldsymbol{\phi}(s,a)^\mathsf{T}\right]$ as showed by the next lemma.

**Lemma 7.** *It holds that* $\mathbf{MV}^k_\theta = \bar{\boldsymbol{\Lambda}}^{-1}_k \mathbb{E}_{(s,a)\sim\mathbf{d}_{k-1}, s'\sim P(\cdot|s,a)}\left[\boldsymbol{\phi}(s,a)V^k_\theta(s')\right].$

*Proof.*

$$\begin{aligned}
\mathbf{MV}^k_\theta &= \bar{\boldsymbol{\Lambda}}^{-1}_k \bar{\boldsymbol{\Lambda}}_k \mathbf{MV}^k_\theta \\
&= \bar{\boldsymbol{\Lambda}}^{-1}_k \mathbb{E}_{(s,a)\sim\mathbf{d}_{k-1}}\left[\boldsymbol{\phi}(s,a)\boldsymbol{\phi}(s,a)^\mathsf{T}\mathbf{MV}^k_\theta\right] \\
&= \bar{\boldsymbol{\Lambda}}^{-1}_k \mathbb{E}_{(s,a)\sim\mathbf{d}_{k-1}}\left[\boldsymbol{\phi}(s,a)\boldsymbol{\phi}(s,a)^\mathsf{T}\sum_{s'}\mathbf{M}_{:s'}V^k_\theta(s')\right] \\
&= \bar{\boldsymbol{\Lambda}}^{-1}_k \mathbb{E}_{(s,a)\sim\mathbf{d}_{k-1}}\left[\boldsymbol{\phi}(s,a)\sum_{s'}\boldsymbol{\phi}(s,a)^\mathsf{T}\mathbf{M}_{:s'}V^k_\theta(s')\right] \\
&= \bar{\boldsymbol{\Lambda}}^{-1}_k \mathbb{E}_{(s,a)\sim\mathbf{d}_{k-1}}\left[\boldsymbol{\phi}(s,a)\sum_{s'}P(s'|s,a)V^k_\theta(s')\right] \\
&= \bar{\boldsymbol{\Lambda}}^{-1}_k \mathbb{E}_{(s,a)\sim\mathbf{d}_{k-1}, s'\sim P(\cdot|s,a)}\left[\boldsymbol{\phi}(s,a)V^k_\theta(s')\right].
\end{aligned}$$

$\square$

It follows that $\mathbf{MV}^k_\theta = \arg\min_{\mathbf{z}} \mathbb{E}_{(s,a)\sim\mathbf{d}_{k-1}, s'\sim P(\cdot|s,a)}\left[\left(\boldsymbol{\phi}(s,a)^\mathsf{T}\mathbf{z} - V^k_\theta(s')\right)^2\right].$

Now, we move to the problem of estimating $\widehat{\mathbf{MV}^k_\theta}$ with a finite amount of environment interactions sampled i.i.d from $\mathbf{d}_{k-1}$. We define

$$\widehat{\mathbf{MV}^k_\theta} \triangleq \arg\min_{\mathbf{z}} \frac{1}{N}\sum^N_{n=1}\left(\boldsymbol{\phi}(s^{(n)}_k, a^{(n)}_k)^\mathsf{T}\mathbf{z} - V^k_\theta(s'^{(n)}_k)\right)^2 + \chi\|\mathbf{z}\|^2_2.$$

By optimality conditions, we can obtain a closed-form expression for $\widehat{\mathbf{MV}^k_\theta}$.

**Lemma 8.** *It holds that*

$$\widehat{\mathbf{MV}_\theta^k} = \frac{1}{N} \left(\mathbf{\Lambda}_{k,N} + \chi\mathbf{I}\right)^{-1} \sum_{n=1}^{N} \phi(s_{k-1}^{(n)}, a_{k-1}^{(n)}) V_\theta^k(s_{k-1}'^{(n)}),$$

*where $\mathbf{\Lambda}_{k,N} \triangleq \frac{1}{N}\sum_{n=1}^{N} \phi(s_{k-1}^{(n)}, a_{k-1}^{(n)})\phi(s_{k-1}^{(n)}, a_{k-1}^{(n)})^\mathsf{T}$ is the empirical covariance matrix.*

*Proof.* Let $\mathcal{L}(\mathbf{z}) \triangleq \frac{1}{N}\sum_{n=1}^{N} \left(\phi(s_{k-1}^{(n)}, a_{k-1}^{(n)})^\mathsf{T}\mathbf{z} - V_\theta^k(s_{k-1}'^{(n)})\right)^2 + \chi\|\mathbf{z}\|_2^2$. The first derivative is given by

$$\frac{1}{2}\nabla_{\mathbf{z}}\mathcal{L}(\mathbf{z}) = \frac{1}{N}\sum_{n=1}^{N} \phi(s_{k-1}^{(n)}, a_{k-1}^{(n)}) \left(\phi(s_{k-1}^{(n)}, a_{k-1}^{(n)})^\mathsf{T}\mathbf{z} - V_\theta^k(s_{k-1}'^{(n)})\right) + \chi\mathbf{z}. \tag{26}$$

Since $\mathcal{L}(\cdot)$ is convex in $\mathbf{z}$, by first-order optimality conditions, we get

$$\frac{1}{N}\sum_{n=1}^{N} \phi(s_{k-1}^{(n)}, a_{k-1}^{(n)}) \left(\phi(s_{k-1}^{(n)}, a_{k-1}^{(n)})^\mathsf{T}\widehat{\mathbf{MV}_\theta^k} - V_\theta^k(s_{k-1}'^{(n)})\right) + \chi\widehat{\mathbf{MV}_\theta^k} = 0$$

The statement follows from rearranging the terms. $\qquad\square$

**Remark 1.** *Note that when $\chi = 0$, and $\phi(s,a)$ is one-hot vector for every $(s,a)$, then we obtain the tabular estimators $\mathbf{W}_v$ proposed in [89].*

We invoke Theorem 2 in [53] to derive an upper bound for $\left\|\mathbf{MV}_\theta^k - \widehat{\mathbf{MV}_\theta^k}\right\|_{\bar{\mathbf{\Lambda}}_k}^2$.

**Lemma 9.** *Fix some $\chi > 0$ and take $N \geq \mathcal{O}(\frac{\log(\frac{m}{\delta})}{\chi\beta})$. Then, with probability at least $1 - \delta$, we have*

$$\left\|\mathbf{MV}_\theta^k - \widehat{\mathbf{MV}_\theta^k}\right\|_{\bar{\mathbf{\Lambda}}_k}^2 \leq \mathcal{O}\left(\frac{m\chi^2}{\beta^3}D^2 + \frac{1}{N}\frac{m\chi}{\beta^4}D^2\log\left(\frac{1}{\delta}\right) + \frac{D^2 m}{N}\log\left(\frac{1}{\delta}\right)\right),$$

*where $D \triangleq \frac{1+\log\left(\frac{1}{\beta}\right)}{1-\gamma} \geq 1$ is the upper bound of $\left\|\mathbf{V}_\theta^k\right\|_\infty$ derived in Proposition 3.*

*Proof.* We introduce the following auxiliary quantities:

$$\mathbf{M}_\chi\mathbf{V}_\theta^k = \arg\min_{\mathbf{z}} \mathop{\mathbb{E}}_{(s,a)\sim\mathbf{d}_{k-1},s'\sim P(\cdot|s,a)} \left[\phi(s,a)^\mathsf{T}\mathbf{z} - V_\theta^k(s')\right] + \chi\|\mathbf{z}\|_2^2$$

$$= \left(\bar{\mathbf{\Lambda}}_k + \chi\mathbf{I}\right)^{-1} \mathop{\mathbb{E}}_{(s,a)\sim\mathbf{d}_{k-1},s'\sim P(\cdot|s,a)} \left[\phi(s,a)V_\theta^k(s')\right],$$

and the conditional expectation

$$\bar{\mathbf{M}}\mathbf{V}_\theta^k = \mathbb{E}\left[\widehat{\mathbf{MV}_\theta^k}\Big|\mathcal{F}_n\right] = \frac{1}{N}\left(\mathbf{\Lambda}_{k,N} + \chi\mathbf{I}\right)^{-1}\sum_{n=1}^{N}\phi(s_{k-1}^{(n)}, a_{k-1}^{(n)}) \mathop{\mathbb{E}}_{s'\sim P(\cdot|s_{k-1}^{(n)}, a_{k-1}^{(n)})}\left[V_\theta^k(s')\right]$$

with $\mathcal{F}_n$ being the filtration $\mathcal{F}_n = \{s_{k-1}^{(i)}, a_{k-1}^{(i)}\}_{i=0}^{n}$. Then applying the general random design decomposition in ([53], Proposition 3) we obtain:

$$\left\|\mathbf{MV}_\theta^k - \widehat{\mathbf{MV}_\theta^k}\right\|_{\bar{\mathbf{\Lambda}}_k}^2 \leq 3\underbrace{\left\|\mathbf{MV}_\theta^k - \mathbf{M}_\chi\mathbf{V}_\theta^k\right\|_{\bar{\mathbf{\Lambda}}_k}^2}_{\triangleq \epsilon_{\mathrm{rg}}} + 3\underbrace{\left\|\mathbf{M}_\chi\mathbf{V}_\theta^k - \bar{\mathbf{M}}\mathbf{V}_\theta^k\right\|_{\bar{\mathbf{\Lambda}}_k}^2}_{\triangleq \epsilon_{\mathrm{bs}}} + 3\underbrace{\left\|\bar{\mathbf{M}}\mathbf{V}_\theta^k - \widehat{\mathbf{MV}_\theta^k}\right\|_{\bar{\mathbf{\Lambda}}_k}^2}_{\triangleq \epsilon_{\mathrm{vr}}},$$

$$\tag{27}$$

where similarly to [53], we define $\epsilon_{\mathrm{rg}}$ as the ridge error, $\epsilon_{\mathrm{bs}}$ the ridge estimator bias and with $\epsilon_{\mathrm{vr}}$ the ridge estimator variance. By choosing $N \geq \mathcal{O}(6\rho_\chi^2 d_{1,\chi}(\log\max(1, d_{1,\chi}) + \log\frac{1}{\delta})) = \mathcal{O}(\frac{1}{\beta\chi}\log\frac{m}{\delta})$, we ensure that the conditions in Theorem 2 in [53] are satisfied. We next bound each term separately.

**Ridge error.** In [53], the bound derived for the ridge error is a function of the regularization parameter $\chi$, the eigenvalues of the covariance matrix $\bar{\boldsymbol{\Lambda}}_k$ denoted as $\{\sigma_j\}_{j=1}^m$ and the corresponding eigenvectors $\{\mathbf{v}_j\}_{j=1}^m$. In particular, we have

$$
\begin{aligned}
\epsilon_{\mathrm{rg}} &\leq \sum_{j=1}^m \frac{\sigma_j}{(\frac{\sigma_j}{\chi}+1)^2}(\mathbf{v}_j^\mathsf{T}\mathbf{M}\mathbf{V}_\theta^k)^2 \\
&= \sum_{j=1}^m \frac{\sigma_j}{(\frac{\sigma_j}{\chi}+1)^2}\left(\mathop{\mathbb{E}}_{(s,a)\sim\mathbf{d}_{k-1},s'\sim P(\cdot|s,a)}\left[\phi(s,a)V_\theta^k(s')\right]^\mathsf{T}\bar{\boldsymbol{\Lambda}}_k^{-1}\mathbf{v}_j\right)^2 \\
&= \sum_{j=1}^m \frac{1}{(\frac{\sigma_j}{\chi}+1)^2\sigma_j}\left(\mathop{\mathbb{E}}_{(s,a)\sim\mathbf{d}_{k-1},s'\sim P(\cdot|s,a)}\left[\phi(s,a)V_\theta^k(s')\right]^\mathsf{T}\mathbf{v}_j\right)^2 \\
&\leq \sum_{j=1}^m \frac{1}{(\frac{\sigma_j}{\chi}+1)^2\sigma_j}\left\|\mathbf{V}_\theta^k\right\|_\infty^2 \\
&\leq \sum_{j=1}^m \frac{1}{(\frac{\beta}{\chi}+1)^2\beta}D^2 \\
&= \frac{m\chi^2}{(\beta+\chi)^2\beta}D^2 \\
&\leq \frac{m\chi^2}{\beta^3}D^2,
\end{aligned}
$$

where in the first inequality we used bullet (3) of Theorem 2 in [53].

**Bias.** It holds that

$$
\epsilon_{\mathrm{bs}} \leq \mathcal{O}\left(\frac{\rho_\chi^2 d_{1,\chi}\mathop{\mathbb{E}}_{(s,a)\sim\mathbf{d}_{k-1}}[\mathrm{approx}(s,a)] + (1+\rho_\chi^2 d_{1,\chi})\epsilon_{\mathrm{rg}}}{N}\log\left(\frac{1}{\delta}\right)\right),
$$

where we used the notation

$$
\begin{aligned}
\mathop{\mathbb{E}}_{(s,a)\sim\mathbf{d}_{k-1}}[\mathrm{approx}(s,a)] &\triangleq \mathop{\mathbb{E}}_{(s,a)\sim\mathbf{d}_{k-1}}\left[\mathop{\mathbb{E}}_{s'\sim P(\cdot|s,a)}\left[V_\theta^k(s')\right]-\phi(s,a)^\mathsf{T}\mathbf{M}\mathbf{V}_\theta^k\right] \\
&= \mathop{\mathbb{E}}_{(s,a)\sim\mathbf{d}_{k-1}}\left[\mathop{\mathbb{E}}_{s'\sim P(\cdot|s,a)}\left[V_\theta^k(s')\right]-\mathbf{P}\mathbf{V}_\theta^k(s,a)\right] \\
&= \mathop{\mathbb{E}}_{(s,a)\sim\mathbf{d}_{k-1}}\left[\mathop{\mathbb{E}}_{s'\sim P(\cdot|s,a)}\left[V_\theta^k(s')\right]-\mathop{\mathbb{E}}_{s'\sim P(\cdot|s,a)}\left[V_\theta^k(s')\right]\right]=0.
\end{aligned}
$$

Moreover,

$$
d_{1,\chi} \triangleq \sum_{j=1}^m \frac{\sigma_j}{\sigma_j+\chi} \leq m
$$

Finally, according to Remark 2 in [53], we have that $\rho_\chi$ is bounded as follows

$$
\rho_\chi^2 \leq \frac{\|\phi(s,a)\|_2^2}{\chi d_{1,\chi}} \leq \frac{1+\chi}{\chi\beta m} \leq \frac{2}{\chi\beta m},
$$

where the last inequality follows from noticing that $d_{1,\chi} \geq \frac{\beta m}{1+\chi}$. Therefore, we can conclude that:

$$
\begin{aligned}
\epsilon_{\mathrm{bs}} &\leq \mathcal{O}\left(\frac{(1+\frac{2}{\chi\beta})\epsilon_{\mathrm{rg}}}{N}\log\left(\frac{1}{\delta}\right)\right) \\
&= \mathcal{O}\left(\frac{\epsilon_{\mathrm{rg}}}{\chi\beta N}\log\left(\frac{1}{\delta}\right)\right) \\
&= \mathcal{O}\left(\frac{1}{N}\frac{m\chi}{\beta^4}D^2\log\left(\frac{1}{\delta}\right)\right),
\end{aligned}
$$

**Variance.** From the bullet (5) in [53] it follows that

$$\epsilon_{\text{vr}} = \mathcal{O}\left(\frac{\text{Var}\left[\mathbf{V}_\theta^k(s') \mid s, a\right] d_{2,\chi}}{N} \log\left(\frac{1}{\delta}\right)\right).$$

We have $\text{Var}\left[\mathbf{V}_\theta^k(s') \mid s, a\right] \leq \left\|\mathbf{V}_\theta^k\right\|_\infty^2 \leq D^2$. Finally, bounding $d_{2,\chi}$ we obtain that

$$d_{2,\chi} = \sum_{j=1}^m \left(\frac{\sigma_j}{\sigma_j + \chi}\right)^2 \leq m.$$

Hence we can conclude

$$\epsilon_{\text{vr}} = \mathcal{O}\left(\frac{D^2 m}{N} \log\left(\frac{1}{\delta}\right)\right).$$

**Final bound.** By combining the above bounds with Equation (27), we get the final bound

$$\left\|\mathbf{MV}_\theta^k - \widehat{\mathbf{MV}_\theta^k}\right\|_{\bar{\mathbf{\Lambda}}_k}^2 \leq \mathcal{O}\left(\frac{m\chi^2}{\beta^3}D^2 + \frac{1}{N}\frac{m\chi}{\beta^4}D^2 \log\left(\frac{1}{\delta}\right) + \frac{D^2 m}{N} \log\left(\frac{1}{\delta}\right)\right).$$

$\square$

The bound above is minimized by choosing $\chi$ as small as allowed. This is made precise in the next corollary.

**Corollary 3.** *Let* $\chi = \mathcal{O}(\frac{\log \frac{m}{\delta}}{\beta N})$. *With probability at least* $1 - \delta$, *it holds that*

$$\left\|\mathbf{MV}_\theta^k - \widehat{\mathbf{MV}_\theta^k}\right\|_{\bar{\mathbf{\Lambda}}_k}^2 \leq \mathcal{O}\left(\frac{D^2 m}{\beta^5 N^2}\left(\log\left(\frac{m}{\delta}\right)\right)^2 + \frac{mD^2}{N} \log\left(\frac{1}{\delta}\right)\right).$$

In order to upper bound $\left\|\mathbf{MV}_\theta^k - \widehat{\mathbf{MV}_\theta^k}\right\|_2^2$ we need the next lemma. Hence, to bound $\left\|\mathbf{MV}_\theta^k - \widehat{\mathbf{MV}_\theta^k}\right\|_2^2$, we can directly apply Theorem 2 in [53] that leads to the following lemma.

**Lemma 10.** *Given a matrix* $\mathbf{A} \in \mathbb{R}^{m \times m}$ *and a vector* $\mathbf{x} \in \mathbb{R}^m$, *we have that* $\|\mathbf{x}\|_{\mathbf{A}} \geq \lambda_{\min}(\mathbf{A}) \|\mathbf{x}\|_2$.

*Proof.* We have that $\mathbf{A} - \lambda_{\min}(\mathbf{A})\mathbf{I} \geq 0$ that implies $\mathbf{x}^\intercal \mathbf{A}\mathbf{x} \geq \lambda_{\min}(\mathbf{A})\mathbf{x}^\intercal \mathbf{x}$. $\square$

**Corollary 4.** *Let* $\chi = \mathcal{O}(\frac{\log \frac{m}{\delta}}{\beta N})$. *With probability at least* $1 - \delta$, *it holds that*

$$\left\|\mathbf{MV}_\theta^k - \widehat{\mathbf{MV}_\theta^k}\right\|_2 \leq \mathcal{O}\left(\frac{D\sqrt{m}}{\beta^3 N} \log\left(\frac{m}{\delta}\right) + \frac{D\sqrt{m}}{\sqrt{N}\beta}\sqrt{\log\left(\frac{1}{\delta}\right)}\right). \tag{28}$$

**Corollary 5.** *Let* $\chi = \mathcal{O}\left(\frac{\log \frac{m}{\delta}}{\beta N}\right)$, *and* $N \geq \max\left(\frac{\gamma^2 mD^2}{\beta\epsilon^2} \log(1/\delta), \frac{\gamma\sqrt{m}D}{\beta^3\epsilon} \log(m/\delta)\right)$. *Then, with probability at least* $1 - \delta$, *it holds that* $\left\|\mathbf{MV}_\theta^k - \widehat{\mathbf{MV}_\theta^k}\right\|_2 \leq \frac{\epsilon}{\gamma}$.

### H.1.2 Estimators for $\mathbf{\Gamma}_k$

Recall that we introduced $\mathbf{\Gamma}_k(i,j) \triangleq \sum_{s',a'} \mathbf{M}_{i,s'} \pi_{k-1,\theta}(a'|s')\phi_j(s',a')$. We can equivalently rewrite it as

$$\mathbf{\Gamma}_k(\cdot, j) = \mathbf{M}\underbrace{\sum_{a'} \pi_{k-1,\theta}(a'|s')\phi_j(s',a')}_{h_{k,j}(s')}$$

$$= \bar{\mathbf{\Lambda}}_k^{-1} \mathbb{E}_{(s,a)\sim\mathbf{d}_{k-1}, s'\sim P(\cdot|s,a)}\left[\phi(s,a)h_{k,j}(s')\right],$$

where the last equality is obtained with manipulations analogous to Lemma 7.

Similarly, We can estimate $\boldsymbol{\Gamma}_k(i,j)$ with a finite amount of environment interactions sampled i.i.d. from $\mathbf{d}_{k-1}$, by solving the following ridge regression problem:

$$\widehat{\boldsymbol{\Gamma}}_k(\cdot,j) = \arg\min_{\mathbf{z}} \frac{1}{N} \sum_{n=1}^{N} \left( \phi(s_{k-1}^{(n)}, a_{k-1}^{(n)})^\mathsf{T} \mathbf{z} - h_{k,j}(s_{k-1}'^{(n)}) \right)^2 + \chi \|\mathbf{z}\|_2^2$$

**Lemma 11.** *By optimality conditions, we can obtain a closed form for $\widehat{\boldsymbol{\Gamma}}_k$ as*

$$\widehat{\boldsymbol{\Gamma}}_k(\cdot,j) = \frac{1}{N} (\boldsymbol{\Lambda}_{k,N} + \chi\mathbf{I})^{-1} \sum_{n=1}^{N} \phi(s_{k-1}^{(n)}, a_{k-1}^{(n)}) h_{k,j}(s_{k-1}'^{(n)}).$$

By noting that $\|\mathbf{h}_{k-1,j}\|_\infty \le 1$ for any $k$, it follows that

**Corollary 6.** *For $\chi = \mathcal{O}(\frac{\log \frac{m}{\delta}}{\beta N})$, with probability at least $1 - \delta$, it holds that*

$$\left\| \boldsymbol{\Gamma}_k(\cdot,j) - \widehat{\boldsymbol{\Gamma}}_k(\cdot,j) \right\|_2 \le \mathcal{O}\left( \frac{\sqrt{m}}{\sqrt{N\beta}} \sqrt{\log\left(\frac{1}{\delta}\right)} + \frac{\sqrt{m}}{\beta^3 N} \log\left(\frac{m}{\delta}\right) \right). \tag{29}$$

**Corollary 7.** *For $\chi = \mathcal{O}(\frac{\log \frac{m}{\delta}}{\beta N})$, and $N \ge \max\left( \mathcal{O}\left(\frac{m}{\beta\epsilon^2}\log(1/\delta)\right), \mathcal{O}\left(\frac{\sqrt{m}}{\beta^3\epsilon}\log(m/\delta)\right)\right)$, with probability at least $1 - \delta$, it holds that $\left\| \boldsymbol{\Gamma}_k(\cdot,j) - \widehat{\boldsymbol{\Gamma}}_k(\cdot,j) \right\|_2 \le \epsilon$.*

### H.1.3 Estimator for feature expectation vector $\boldsymbol{\rho}_{\boldsymbol{\Phi}}(\pi_{k-1})$

The goal is to estimate $\boldsymbol{\rho}_{\boldsymbol{\Phi}}(\pi_{k-1})$. Consider the sample transitions $\{s_{k-1}^{(n)}, a_{k-1}^{(n)}\}_{n=1}^{N} \sim \mathbf{d}_{k-1}^N$. Then we estimate $\boldsymbol{\rho}_{\boldsymbol{\Phi}}(\pi_{k-1}) = \boldsymbol{\Phi}^\mathsf{T}\mathbf{d}_{k-1}$ by $\boldsymbol{\rho}_{\boldsymbol{\Phi}}(\widehat{\pi_{k-1}}) \triangleq \frac{1}{N} \sum_{n=1}^{N} \phi(s_{k-1}^{(n)}, a_{k-1}^{(n)})$.

In the next lemma, we provide a useful concentration result.

**Lemma 12.** *With probability at least $1 - \delta$, for all $N \ge \frac{1.4 \log\log(2N) + \log\frac{10.4m}{\delta}}{\beta\epsilon^2}$, and for all $i \in [m]$ simultaneously, it holds that*

$$\left| \boldsymbol{\rho}_{\boldsymbol{\Phi}}(\widehat{\pi_{k-1}})(i) - \boldsymbol{\rho}_{\boldsymbol{\phi}}(\pi_{k-1})(i) \right| \le 2.26\epsilon \boldsymbol{\rho}_{\boldsymbol{\phi}}(\pi_{k-1})(i) \tag{30}$$

*Proof.* Consider the martingale difference sequence $Z_i(n) = \phi_i(s_{k-1}^{(n)}, a_{k-1}^{(n)}) - \boldsymbol{\rho}_{\boldsymbol{\Phi}}(\pi_{k-1})(i)$ with the variance process $V_i(n) = \sum_{j=1}^{n} \mathbb{E}\left[ Z_i^2(j) | \mathcal{F}_{j-1} \right]$, where $\mathcal{F}_{j-1}$ being the filtration up to the state action pair $(s_{k-1}^{(j)}, a_{k-1}^{(j)})$. We have,

$$
\begin{aligned}
V_i(n) &= \sum_{j=1}^{n} \mathbb{E}\left[ Z_i^2(j) | \mathcal{F}_{j-1} \right] \\
&= \sum_{j=1}^{n} \mathbb{E}_{(s,a)\sim\mathbf{d}_{k-1}} \left[ (\phi_i(s,a) - \boldsymbol{\rho}_{\boldsymbol{\Phi}}(\pi_{k-1})(i))^2 | \mathcal{F}_{j-1} \right] \\
&= \sum_{j=1}^{n} \mathbb{E}_{(s,a)\sim\mathbf{d}_{k-1}} \left[ \phi_i^2(s,a) - 2\phi_i(s,a)\boldsymbol{\rho}_{\boldsymbol{\Phi}}(\pi_{k-1})(i) + \boldsymbol{\rho}_{\boldsymbol{\Phi}}(\pi_{k-1})(i)^2 | \mathcal{F}_{j-1} \right] \\
&\le \sum_{j=1}^{n} \mathbb{E}_{(s,a)\sim\mathbf{d}_{k-1}} \left[ \phi_i(s,a) | \mathcal{F}_{j-1} \right] - n\boldsymbol{\rho}_{\boldsymbol{\Phi}}(\pi_{k-1})(i)^2 \\
&= n\left( \boldsymbol{\rho}_{\boldsymbol{\Phi}}(\pi_{k-1})(i) - \boldsymbol{\rho}_{\boldsymbol{\Phi}}(\pi_{k-1})(i)^2 \right) \le n\boldsymbol{\rho}_{\boldsymbol{\Phi}}(\pi_{k-1})(i).
\end{aligned}
$$

The martingale difference sequence $Z_i(j)$ satisfies the sub-$\psi_P$ condition of [52] (see Bennet case in their Table 3) with constant $c = 2$. Therefore, by Lemma 13 in [89] with $m = \boldsymbol{\rho}_{\boldsymbol{\Phi}}(\pi_{k-1})(i)$, with

probability at least $1 - \frac{\delta}{2m}$, for all $N \geq \frac{1.4 \log \log (2N) + \log \frac{10.4m}{\delta}}{\beta \epsilon^2}$ simultaneously, it holds that

$$N \boldsymbol{\rho}_{\boldsymbol{\Phi}}(\widehat{\pi_{k-1}})(i) \geq N \boldsymbol{\rho}_{\boldsymbol{\Phi}}(\pi_{k-1})(i) - 1.44 \sqrt{\boldsymbol{\rho}_{\boldsymbol{\Phi}}(\pi_{k-1})(i) N \left( \log \log 2N + \frac{10.4m}{\delta} \right)}$$

$$- 0.82 \left( 1.4 \log \log 2N + \frac{10.4m}{\delta} \right)$$

$$\geq N \boldsymbol{\rho}_{\boldsymbol{\Phi}}(\pi_{k-1})(i) - 1.44 \sqrt{\boldsymbol{\rho}_{\boldsymbol{\Phi}}(\pi_{k-1})(i)^2 N^2 \epsilon^2} - 0.82 N \beta \epsilon^2$$

$$\geq N \boldsymbol{\rho}_{\boldsymbol{\Phi}}(\pi_{k-1})(i) - 2.26 \boldsymbol{\rho}_{\boldsymbol{\Phi}}(\pi_{k-1})(i) N \epsilon.$$

Similarly, with probability at least $1 - \frac{\delta}{2m}$, for all $N \geq \frac{1.4 \log \log (2N) + \log \frac{10.4m}{\delta}}{\beta \epsilon^2}$ simultaneously, it holds that $\boldsymbol{\rho}_{\boldsymbol{\Phi}}(\widehat{\pi_{k-1}})(i) \leq \boldsymbol{\rho}_{\boldsymbol{\Phi}}(\pi_{k-1})(i) + 2.26 \boldsymbol{\rho}_{\boldsymbol{\Phi}}(\pi_{k-1})(i) N \epsilon$. A union bound concludes the proof. $\qquad \square$

### H.1.4   Estimators for $\widehat{\mathbf{B}}_{\mathbf{w}, \boldsymbol{\theta}}^k$

We can directly invoke Lemma 17 in [89] to get guarantees for the estimator $\widehat{\mathbf{B}}_{\mathbf{w}, \boldsymbol{\theta}}^k(i)$. In particular, we obtain the following result.

**Lemma 13.** *Let* $\left\| \mathbf{MV}_{\theta}^k - \widehat{\mathbf{MV}}_{\theta}^k \right\|_{\infty} \leq \frac{\epsilon}{\gamma}$ *and* $\left| \boldsymbol{\rho}_{\boldsymbol{\Phi}}(\widehat{\pi_{k-1}})(i) - \boldsymbol{\rho}_{\boldsymbol{\Phi}}(\pi_{k-1})(i) \right| \leq 2.26 \epsilon \boldsymbol{\rho}_{\boldsymbol{\Phi}}(\widehat{\pi_{k-1}})(i)$. *Then, it holds that* $\left| \widehat{\mathbf{B}}_{\mathbf{w}, \boldsymbol{\theta}}^k(i) - \mathbf{B}_{\mathbf{w}, \boldsymbol{\theta}}^k(i) \right| \leq 38 \eta \epsilon \mathbf{B}_{\mathbf{w}, \boldsymbol{\theta}}^k(i) \leq 38 \frac{\eta \epsilon}{\beta}$.

*Proof.* First, we notice that $\left\| \mathbf{MV}_{\theta}^k - \widehat{\mathbf{MV}}_{\theta}^k \right\|_{\infty} \leq \frac{\epsilon}{\gamma}$ implies that $\widehat{\boldsymbol{\delta}}_{\mathbf{w}, \boldsymbol{\theta}}^k(i) - \boldsymbol{\delta}_{\mathbf{w}, \boldsymbol{\theta}}^k(i) \leq \epsilon$. Therefore, by Lemma 17 in [89] we get $\left| \widehat{\mathbf{B}}_{\mathbf{w}, \boldsymbol{\theta}}^k(i) - \mathbf{B}_{\mathbf{w}, \boldsymbol{\theta}}^k(i) \right| \leq 38 \eta \epsilon \mathbf{B}_{\mathbf{w}, \boldsymbol{\theta}}^k(i)$. Moreover, it holds that

$$\mathbf{B}_{\mathbf{w}, \boldsymbol{\theta}}^k(i) = \frac{e^{-\eta \boldsymbol{\delta}_{\mathbf{w}, \boldsymbol{\theta}}^k(i)}}{\sum_i^m \rho_{\phi_i}(\pi_{k-1}) e^{-\eta \boldsymbol{\delta}_{\mathbf{w}, \boldsymbol{\theta}}^k(i)}} \leq \frac{e^{-\eta \boldsymbol{\delta}_{\mathbf{w}, \boldsymbol{\theta}}^k(i)}}{\beta \sum_i^m e^{-\eta \boldsymbol{\delta}_{\mathbf{w}, \boldsymbol{\theta}}^k(i)}} \leq \frac{1}{\beta}.$$

Therefore,

$$\left| \widehat{\mathbf{B}}_{\mathbf{w}, \boldsymbol{\theta}}^k(i) - \mathbf{B}_{\mathbf{w}, \boldsymbol{\theta}}^k(i) \right| \leq \frac{38 \eta \epsilon}{\beta}, \quad \text{and} \quad \widehat{\mathbf{B}}_{\mathbf{w}, \boldsymbol{\theta}}^k(i) \leq \mathbf{B}_{\mathbf{w}, \boldsymbol{\theta}}^k(i) (1 + 38 \eta \epsilon) \leq \frac{1}{\beta} (1 + 38 \eta \epsilon).$$

$\qquad \square$

**Corollary 8.** *Let* $N_1 \geq \max \left( \mathcal{O} \left( \frac{\gamma^2 m D^2}{\beta \epsilon^2} \log(2/\delta) \right), \mathcal{O} \left( \frac{\gamma \sqrt{m} D}{\beta^3 \epsilon} \log(2m/\delta) \right) \right)$ *and* $N_2 \geq \frac{1.4 \log \log (2N_2) + \log \frac{20.8m}{\delta}}{\beta \epsilon^2}$. *Then, for* $\chi = \mathcal{O} \left( \frac{\log \frac{2m}{\delta}}{\beta N} \right)$, *and for* $N \geq \max (N_1, N_2)$, *with probability at least* $1 - \delta$, *it holds that* $\left| \widehat{\mathbf{B}}_{\mathbf{w}, \boldsymbol{\theta}}^k(i) - \mathbf{B}_{\mathbf{w}, \boldsymbol{\theta}}^k(i) \right| \leq 38 \frac{\eta \epsilon}{\beta}$, *for all* $i \in [m]$.

*Proof.* By Corollary 5, we have that with $N \geq N_1$ it holds that $\left\| \mathbf{MV}_{\theta}^k - \widehat{\mathbf{MV}}_{\theta}^k \right\|_{\infty} \leq \frac{\epsilon}{\gamma}$, with probability $1 - \delta/2$. Furthermore, Lemma 12 gives that for $N \geq \frac{1.4 \log \log (2N) + \log \frac{20.8m}{\delta}}{\beta \epsilon^2}$, it holds with probability $1 - \delta/2$ that $\left| \boldsymbol{\rho}_{\boldsymbol{\Phi}}(\widehat{\pi_{k-1}})(i) - \boldsymbol{\rho}_{\boldsymbol{\Phi}}(\pi_{k-1})(i) \right| \leq 2.26 \epsilon \boldsymbol{\rho}_{\boldsymbol{\Phi}}(\widehat{\pi_{k-1}})(i)$, for all $i \in [m]$ simultaneously.

Therefore, a union bound gives that for $N \geq \max (N_1, N_2)$, with probability $1 - \delta$, we have that $\left\| \mathbf{MV}_{\theta}^k - \widehat{\mathbf{MV}}_{\theta}^k \right\|_{\infty} \leq \frac{\epsilon}{\gamma}$, and $\left| \boldsymbol{\rho}_{\boldsymbol{\Phi}}(\widehat{\pi_{k-1}})(i) - \boldsymbol{\rho}_{\boldsymbol{\Phi}}(\pi_{k-1})(i) \right| \leq 2.26 \epsilon \boldsymbol{\rho}_{\boldsymbol{\Phi}}(\widehat{\pi_{k-1}})(i)$, for all $i \in [m]$. An application of Lemma 13 concludes the proof. $\qquad \square$

### H.1.5   Estimators for $\mathbf{B}_{\mathbf{w}, \boldsymbol{\theta}}^k(i) \boldsymbol{\Gamma}_k(i, j)$

We obtain an estimator for $\mathbf{B}_{\mathbf{w}, \boldsymbol{\theta}}^k(i) \boldsymbol{\Gamma}_k(i, j)$ simply as $\widehat{\mathbf{B}}_{\mathbf{w}, \boldsymbol{\theta}}^k(i) \widehat{\boldsymbol{\Gamma}}_k(i, j)$. The next lemma gives guarantees for such an estimator.

**Lemma 14.** *Assume that for any $(i,j) \in [m]^2$, it holds that $\left|\widehat{\mathbf{B}}^k_{\mathbf{w},\boldsymbol{\theta}}(i) - \mathbf{B}^k_{\mathbf{w},\boldsymbol{\theta}}(i)\right| \leq \frac{38\eta\epsilon}{\beta}$ and $\left|\widehat{\boldsymbol{\Gamma}}_k(i,j) - \boldsymbol{\Gamma}_k(i,j)\right| \leq \epsilon$. Then, $\left|\mathbf{B}^k_{\mathbf{w},\boldsymbol{\theta}}(i)\boldsymbol{\Gamma}_k(i,j) - \widehat{\mathbf{B}}^k_{\mathbf{w},\boldsymbol{\theta}}(i)\widehat{\boldsymbol{\Gamma}}_k(i,j)\right| \leq \frac{\epsilon}{\beta}(1 + (1+\epsilon)38\eta)$, for all $(i,j) \in [m]^2$.*

*Proof.* We have that

$$
\begin{aligned}
\left|\mathbf{B}^k_{\mathbf{w},\boldsymbol{\theta}}(i)\boldsymbol{\Gamma}_k(i,j) - \widehat{\mathbf{B}}^k_{\mathbf{w},\boldsymbol{\theta}}(i)\widehat{\boldsymbol{\Gamma}}_k(i,j)\right| &\leq \mathbf{B}^k_{\mathbf{w},\boldsymbol{\theta}}(i)\left|\widehat{\boldsymbol{\Gamma}}_k(i,j) - \boldsymbol{\Gamma}_k(i,j)\right| + \widehat{\boldsymbol{\Gamma}}_k(i,j)\left|\widehat{\mathbf{B}}^k_{\mathbf{w},\boldsymbol{\theta}}(i) - \mathbf{B}^k_{\mathbf{w},\boldsymbol{\theta}}(i)\right| \\
&\leq \frac{1}{\beta}\left|\widehat{\boldsymbol{\Gamma}}_k(i,j) - \boldsymbol{\Gamma}_k(i,j)\right| + (1+\epsilon)\left|\widehat{\mathbf{B}}^k_{\mathbf{w},\boldsymbol{\theta}}(i) - \mathbf{B}^k_{\mathbf{w},\boldsymbol{\theta}}(i)\right| \\
&\leq \frac{\epsilon}{\beta} + (1+\epsilon)\frac{38\eta\epsilon}{\beta} = \frac{\epsilon}{\beta}(1 + (1+\epsilon)38\eta),
\end{aligned}
$$

where we used the bound $\widehat{\boldsymbol{\Gamma}}_k(i,j) \leq \boldsymbol{\Gamma}_k(i,j) + \epsilon \leq 1 + \epsilon$. $\qquad\square$

**Lemma 15.** *For $\chi = \mathcal{O}\left(\frac{\log \frac{m}{\delta}}{\beta N}\right)$, choose $N \geq \max\left(N_1, N_2, \mathcal{O}\left(\frac{m}{\beta\epsilon^2}\log(m/\delta)\right), \mathcal{O}\left(\frac{\sqrt{m}}{\beta^3\epsilon}\log(m^2/\delta)\right)\right)$ with $N_1$ and $N_2$ as defined in Corollary 8, then with probability $1 - \delta$, for all $(i,j) \in [m]^2$ simultaneously:*

$$
\left|\mathbf{B}^k_{\mathbf{w},\boldsymbol{\theta}}(i)\boldsymbol{\Gamma}_k(i,j) - \widehat{\mathbf{B}}^k_{\mathbf{w},\boldsymbol{\theta}}(i)\widehat{\boldsymbol{\Gamma}}_k(i,j)\right| \leq \frac{\epsilon}{\beta}(1 + (1+\epsilon)38\eta).
$$

*Proof.* By Corollary 8, when $\chi = \mathcal{O}\left(\frac{\log \frac{m}{\delta}}{\beta N}\right)$, and $N \geq \max(N_1, N_2)$, it holds with probability at least $1 - \delta$ that

$$
\left|\widehat{\mathbf{B}}^k_{\mathbf{w},\boldsymbol{\theta}}(i) - \mathbf{B}^k_{\mathbf{w},\boldsymbol{\theta}}(i)\right| \leq \frac{38\eta\epsilon}{\beta}, \text{ for all } i \in [m].
$$

Moreover, when $N \geq \max\left(\mathcal{O}\left(\frac{m}{\beta\epsilon^2}\log(m/\delta)\right), \mathcal{O}\left(\frac{\sqrt{m}}{\beta^3\epsilon}\log(m^2/\delta)\right)\right)$, by Corollary 7, with probability at least $1 - \delta$, it holds that $\left\|\boldsymbol{\Gamma}_k(\cdot,j) - \widehat{\boldsymbol{\Gamma}}_k(\cdot,j)\right\|_2 \leq \epsilon$, for all $j \in [m]$ simultaneously.

Finally, a union bound and Lemma 14 give that with probability at least $1 - \delta$, it holds that $\left|\mathbf{B}^k_{\mathbf{w},\boldsymbol{\theta}}(i)\boldsymbol{\Gamma}_k(i,j) - \widehat{\mathbf{B}}^k_{\mathbf{w},\boldsymbol{\theta}}(i)\widehat{\boldsymbol{\Gamma}}_k(i,j)\right| \leq \frac{\epsilon}{\beta}(1 + (1+\epsilon)38\eta)$. $\qquad\square$

### H.2 Properties of Stochastic gradients

**Lemma 16.** *Let $N \geq \max\left(N_1, N_2, \mathcal{O}\left(\frac{m}{\beta\epsilon^2}\log(m/\delta)\right), \mathcal{O}\left(\frac{\sqrt{m}}{\beta^3\epsilon}\log(m^2/\delta)\right)\right)$ with $N_1$ and $N_2$ with $N_1$ and $N_2$ as defined in Corollary 8. Then, with probability $1 - \delta$, the following bounds on the stochastic gradient variance hold simultaneously:*

$$
\left\|\widehat{\nabla}_{\boldsymbol{\theta}}\mathcal{G}_k(\mathbf{w},\boldsymbol{\theta}) - \mathbb{E}_{i^{(N+1)}_{k-1}}\left[\widehat{\nabla}_{\boldsymbol{\theta}}\mathcal{G}_k(\mathbf{w},\boldsymbol{\theta})|\mathcal{F}_N\right]\right\|_\infty \leq 2\frac{(1 + 38\epsilon\eta)}{\beta}(2 + \epsilon) + 2(1 - \gamma),
$$

$$
\left\|\widehat{\nabla}_{\mathbf{w}}\mathcal{G}_k(\mathbf{w},\boldsymbol{\theta}) - \mathbb{E}_{i^{(N+1)}_{k-1}}\left[\widehat{\nabla}_{\mathbf{w}}\mathcal{G}_k(\mathbf{w},\boldsymbol{\theta})|\mathcal{F}_N\right]\right\|_\infty \leq 2\left(1 + \frac{1 + 38\eta\epsilon}{\beta}\right).
$$

*Furthermore, with probability at least $1 - \delta$, the following bounds on the stochastic gradient bias hold simultaneously:*

$$
\left\|\mathbb{E}\left[\widehat{\nabla}_{\boldsymbol{\theta},j}\mathcal{G}_k(\mathbf{w},\boldsymbol{\theta})|\mathcal{F}_N\right] - \nabla_{\boldsymbol{\theta},j}\mathcal{G}_k(\mathbf{w},\boldsymbol{\theta})\right\|_1 \leq m\frac{\epsilon}{\beta}(\gamma + 38\eta(1 + \gamma(1+\epsilon))),
$$

$$
\left\|\nabla_{\mathbf{w},j}\mathcal{G}_k(\mathbf{w},\boldsymbol{\theta}) - \mathbb{E}\left[\widehat{\nabla}_{\mathbf{w},j}\mathcal{G}_k(\mathbf{w},\boldsymbol{\theta})|\mathcal{F}_N\right]\right\|_1 \leq \frac{38\eta\epsilon}{\beta}.
$$

*Proof.* **Variance for gradient wrt $\boldsymbol{\theta}$.** Recall that by definition of the stochastic gradient we have that

$$\widehat{\nabla}_{\boldsymbol{\theta},j}\mathcal{G}_k(\mathbf{w},\boldsymbol{\theta}) - (1-\gamma)\phi_j(s_{k-1}^{(0)}, a_{k-1}^{(0)}) = \widehat{\mathbf{B}}_{\mathbf{w},\boldsymbol{\theta}}^k(i_{k-1}^{(N+1)})\Big[\gamma\widehat{\boldsymbol{\Gamma}}_k(i_{k-1}^{(N+1)},j) - \mathbb{1}\{i_{k-1}^{(N+1)} = j\}\Big].$$

It then follows that

$$\left|\widehat{\nabla}_{\boldsymbol{\theta}}\mathcal{G}_k(\mathbf{w},\boldsymbol{\theta}) - (1-\gamma)\phi_j(s_{k-1}^{(0)}, a_{k-1}^{(0)})\right| \leq \gamma\left|\widehat{\mathbf{B}}_{\mathbf{w},\boldsymbol{\theta}}^k(i_{k-1}^{(N+1)})\widehat{\boldsymbol{\Gamma}}_k(i_{k-1}^{(N+1)},j)\right| + \left\|\widehat{\mathbf{B}}_{\mathbf{w},\boldsymbol{\theta}}^k(j)\right\|_\infty$$

Invoking Lemma 15, we have that if $N \geq \max\left(N_1, N_2, \mathcal{O}\left(\frac{m}{\beta\epsilon^2}\log(m/\delta)\right), \mathcal{O}\left(\frac{\sqrt{m}}{\beta^3\epsilon}\log(m^2/\delta)\right)\right)$ with $N_1$ and $N_2$ as defined in Corollary 8, then with probability $1 - \delta$,

$$\left|\widehat{\mathbf{B}}_{\mathbf{w},\boldsymbol{\theta}}^k(i_{k-1}^{(N+1)})\widehat{\boldsymbol{\Gamma}}_k(i_{k-1}^{(N+1)},j)\right| \leq \frac{1}{\beta} + \frac{\epsilon}{\beta}(1 + 38(1+\epsilon)\eta) = \frac{1}{\beta}(1 + \epsilon(1 + 38(1+\epsilon)\eta)).$$

Similarly, by Corollary 8, for $N \geq \max(N_1, N_2)$, we have that with probability $1 - \delta$,

$$\widehat{\mathbf{B}}_{\mathbf{w},\boldsymbol{\theta}}^k(i_{k-1}^{(N+1)}) \leq \mathbf{B}_{\mathbf{w},\boldsymbol{\theta}}^k(i_{k-1}^{(N+1)})(1 + 38\eta\epsilon) \leq \frac{1}{\beta}(1 + 38\eta\epsilon).$$

Hence, a union bound gives that with probability $1 - \delta$,

$$\left|\widehat{\nabla}_{\boldsymbol{\theta},j}\mathcal{G}_k(\mathbf{w},\boldsymbol{\theta}) - (1-\gamma)\phi_j(s_{k-1}^{(0)}, a_{k-1}^0)\right| \leq \gamma\frac{(1 + 38\epsilon\eta)}{\beta}(1 + \epsilon) + \frac{(1 + 38\epsilon\eta)}{\beta}.$$

This implies that

$$\left|\widehat{\nabla}_{\boldsymbol{\theta},j}\mathcal{G}_k(\mathbf{w},\boldsymbol{\theta})\right| \leq \gamma\frac{(1 + 38\epsilon\eta)}{\beta}(1 + \epsilon) + \frac{(1 + 38\epsilon\eta)}{\beta} + (1 - \gamma).$$

Therefore, by introducing a filtration $\mathcal{F}_N = \sigma\left(\{(s_{k-1}^{(n)}, a_{k-1}^{(n)}, s_{k-1}'^{(n)})\}_{n=1}^N\right)$, and noticing that $\widehat{\mathbf{B}}_{\mathbf{w},\boldsymbol{\theta}}^k$ and $\widehat{\boldsymbol{\Gamma}}_k$ are $\mathcal{F}_N$-measurable, we get

$$\left|\mathop{\mathbb{E}}_{i_{k-1}^{(N+1)}}\left[\widehat{\nabla}_{\boldsymbol{\theta},j}\mathcal{G}_k(\mathbf{w},\boldsymbol{\theta})|\mathcal{F}_N\right]\right| \leq \mathop{\mathbb{E}}_{i_{k-1}^{(N+1)}}\left[\left|\widehat{\nabla}_{\boldsymbol{\theta},j}\mathcal{G}_k(\mathbf{w},\boldsymbol{\theta})\right||\mathcal{F}_N\right]$$

$$\leq \gamma\frac{(1 + 38\epsilon\eta)}{\beta}(1 + \epsilon) + \frac{(1 + 38\epsilon\eta)}{\beta} + (1 - \gamma)$$

At this point, we can simply notice that

$$\left|\widehat{\nabla}_{\boldsymbol{\theta}}\mathcal{G}_k(\mathbf{w},\boldsymbol{\theta}) - \mathop{\mathbb{E}}_{i_{k-1}^{(N+1)}}\left[\widehat{\nabla}_{\boldsymbol{\theta}}\mathcal{G}_k(\mathbf{w},\boldsymbol{\theta})|\mathcal{F}_N\right]\right| \leq 2\left[\gamma\frac{(1 + 38\epsilon\eta)}{\beta}(1 + \epsilon) + \frac{(1 + 38\epsilon\eta)}{\beta} + 2(1 - \gamma)\right]$$

$$\leq 2\frac{(1 + 38\epsilon\eta)}{\beta}(2 + \epsilon) + 2(1 - \gamma).$$

Therefore, with probability $1 - \delta$, it holds that

$$\left\|\widehat{\nabla}_{\boldsymbol{\theta}}\mathcal{G}_k(\mathbf{w},\boldsymbol{\theta}) - \mathop{\mathbb{E}}_{i_{k-1}^{(N+1)}}\left[\widehat{\nabla}_{\boldsymbol{\theta}}\mathcal{G}_k(\mathbf{w},\boldsymbol{\theta})|\mathcal{F}_N\right]\right\|_\infty \leq 2\frac{(1 + 38\epsilon\eta)}{\beta}(2 + \epsilon) + 2(1 - \gamma).$$

**Variance for gradient wrt w.** Similarly with Corollary 8, we obtain that if $N \geq \max(N_1, N_2)$, then with probability at least $1 - \delta$,

$$\left\|\widehat{\nabla}_{\mathbf{w}}\mathcal{G}_k(\mathbf{w},\boldsymbol{\theta})\right\|_\infty \leq 1 + \frac{1 + 38\eta\epsilon}{\beta}.$$

This implies that

$$\left\|\widehat{\nabla}_{\mathbf{w}}\mathcal{G}_k(\mathbf{w},\boldsymbol{\theta}) - \mathop{\mathbb{E}}_{i_{k-1}^{(N+1)}}\left[\widehat{\nabla}_{\mathbf{w}}\mathcal{G}_k(\mathbf{w},\boldsymbol{\theta})|\mathcal{F}_N\right]\right\|_\infty \leq 2\left(1 + \frac{1 + 38\eta\epsilon}{\beta}\right).$$

**Bias for gradient wrt $\boldsymbol{\theta}$.** By using the unbiased estimator $\widetilde{\nabla}_{\boldsymbol{\theta},j}\mathcal{G}_k(\mathbf{w},\boldsymbol{\theta})$ in Equation (25), we get

$$\left|\widetilde{\nabla}_{\boldsymbol{\theta},j}\mathcal{G}_k(\mathbf{w},\boldsymbol{\theta}) - \widehat{\nabla}_{\boldsymbol{\theta},j}\mathcal{G}_k(\mathbf{w},\boldsymbol{\theta})\right| \leq \left|\gamma\left(\widehat{\mathbf{B}}_{\mathbf{w},\boldsymbol{\theta}}^k(i_{k-1}^{(N+1)})\widehat{\mathbf{\Gamma}}_k(i_{k-1}^{(N+1)},j) - \mathbf{B}_{\mathbf{w},\boldsymbol{\theta}}^k(i_{k-1}^{(N+1)})\mathbf{\Gamma}_k(i_{k-1}^{(N+1)},j)\right)\right|$$

$$+ \left|\mathbb{1}\{i_{k-1}^{(N+1)}=j\}\left(\widehat{\mathbf{B}}_{\mathbf{w},\boldsymbol{\theta}}^k(i_{k-1}^{(N+1)}) - \mathbf{B}_{\mathbf{w},\boldsymbol{\theta}}^k(i_{k-1}^{(N+1)})\right)\right|$$

$$\leq \gamma\left|\widehat{\mathbf{B}}_{\mathbf{w},\boldsymbol{\theta}}^k(i_{k-1}^{(N+1)})\widehat{\mathbf{\Gamma}}_k(i_{k-1}^{(N+1)},j) - \mathbf{B}_{\mathbf{w},\boldsymbol{\theta}}^k(i^{(N+1)})\mathbf{\Gamma}_k(i_{k-1}^{(N+1)},j)\right|.$$

By choosing $\chi$ and $N$ as in Lemma 15 and Corollary 8, and by a union bound, we have that with probability $1-\delta$,

$$\left|\widetilde{\nabla}_{\boldsymbol{\theta},j}\mathcal{G}_k(\mathbf{w},\boldsymbol{\theta}) - \widehat{\nabla}_{\boldsymbol{\theta},j}\mathcal{G}_k(\mathbf{w},\boldsymbol{\theta})\right| \leq \gamma\frac{\epsilon}{\beta}(1+(1+\epsilon)38\eta) + \frac{38\epsilon\eta}{\beta}$$

$$= \frac{\epsilon}{\beta}\left(\gamma + 38\eta\left(1+\gamma(1+\epsilon)\right)\right).$$

Using that $\widetilde{\nabla}_{\boldsymbol{\theta},j}\mathcal{G}_k(\mathbf{w},\boldsymbol{\theta})$ is an unbiased estimator of $\nabla_{\boldsymbol{\theta},j}\mathcal{G}_k(\mathbf{w},\boldsymbol{\theta})$, we get

$$\left|\mathbb{E}\left[\widehat{\nabla}_{\boldsymbol{\theta},j}\mathcal{G}_k(\mathbf{w},\boldsymbol{\theta})|\mathcal{F}_N\right] - \nabla_{\boldsymbol{\theta},j}\mathcal{G}_k(\mathbf{w},\boldsymbol{\theta})\right| = \left|\mathbb{E}\left[\widehat{\nabla}_{\boldsymbol{\theta},j}\mathcal{G}_k(\mathbf{w},\boldsymbol{\theta}) - \widetilde{\nabla}_{\boldsymbol{\theta},j}\mathcal{G}_k(\mathbf{w},\boldsymbol{\theta})\right]\right|$$

$$\leq \mathbb{E}\left[\left|\widehat{\nabla}_{\boldsymbol{\theta},j}\mathcal{G}_k(\mathbf{w},\boldsymbol{\theta}) - \widetilde{\nabla}_{\boldsymbol{\theta},j}\mathcal{G}_k(\mathbf{w},\boldsymbol{\theta})\right|\right]$$

$$\leq \frac{\epsilon}{\beta}\left(\gamma + 38\eta\left(1+\gamma(1+\epsilon)\right)\right).$$

Hence, we have that $\left\|\mathbb{E}\left[\widehat{\nabla}_{\boldsymbol{\theta},j}\mathcal{G}_k(\mathbf{w},\boldsymbol{\theta})|\mathcal{F}_N\right] - \nabla_{\boldsymbol{\theta},j}\mathcal{G}_k(\mathbf{w},\boldsymbol{\theta})\right\|_1 \leq m\frac{\epsilon}{\beta}\left(\gamma + 38\eta\left(1+\gamma(1+\epsilon)\right)\right).$

**Bias bound for the gradient wrt w.** Similarly, we can notice that with probability at least $1-\delta$, it holds that

$$\left|\widetilde{\nabla}_{\mathbf{w},j}\mathcal{G}_k(\mathbf{w},\boldsymbol{\theta}) - \widehat{\nabla}_{\mathbf{w},j}\mathcal{G}_k(\mathbf{w},\boldsymbol{\theta})\right| = \left|\mathbb{1}\{i_{k-1}^{(N+1)}=j\}\left(\widehat{\mathbf{B}}_{\mathbf{w},\boldsymbol{\theta}}^k(i_{k-1}^{(N+1)}) - \mathbf{B}_{\mathbf{w},\boldsymbol{\theta}}^k(i_{k-1}^{(N+1)})\right)\right|$$

$$\leq \frac{38\eta\epsilon}{\beta}.$$

Since we have only one non-zero element, and by the unbiasedness of $\widetilde{\nabla}_{\mathbf{w},j}\mathcal{G}_k(\mathbf{w},\boldsymbol{\theta})$, we get

$$\left\|\nabla_{\mathbf{w},j}\mathcal{G}_k(\mathbf{w},\boldsymbol{\theta}) - \mathbb{E}\left[\widehat{\nabla}_{\mathbf{w},j}\mathcal{G}_k(\mathbf{w},\boldsymbol{\theta})|\mathcal{F}_N\right]\right\|_1 \leq \frac{38\eta\epsilon}{\beta}.$$

$\square$

# I  Proof of Theorem 2

We first prove a generalization of the Azuma-Hoeffding inequality (Theorem 3.14 in [75]) that holds when the martingale difference sequence is bounded with high probability but not almost surely.

**Lemma 17** (Modified Azuma-Hoeffding). *Let $\{Y_i\}_i^n$ be a martingale difference sequence adapted to $\mathcal{F}_i$, such that for each $i$, $|Y_i| \leq c_i$ with probability at least $1-\delta_2$. Then, it holds that*

$$\mathbb{P}\left[\sum_{i=1}^n Y_i \geq \epsilon\right] \leq \exp\left(-\frac{2\epsilon^2}{\sum_{i=1}^n c_i^2}\right) + n\delta_2. \tag{31}$$

*Proof.* Define the events $E_i = \{Y_i \leq c_i\}$ and the intersection $E = \cap_{i=1}^n\{E_i\}$, and notice that $\mathbb{P}[E^c] = \mathbb{P}[\cup_{i=1}^n\{E_i^c\}] \leq \sum_{i=1}^n \mathbb{P}[E_i^c] = n\delta_2$. We then have the following decomposition:

$$\mathbb{P}\left[\sum_{i=1}^n Y_i \geq \epsilon\right] = \mathbb{P}\left[\{\sum_{i=1}^n Y_i \geq \epsilon\} \cap E\right] + \mathbb{P}\left[\{\sum_{i=1}^n Y_i \geq \epsilon\} \cap E^c\right]$$

$$\leq \mathbb{P}\left[\{\sum_{i=1}^n Y_i \geq \epsilon\} \cap E\right] + \mathbb{P}[E^c]$$

$$\leq \mathbb{P}\left[\sum_{i=1}^n Y_i \geq \epsilon\Big|E\right]\underbrace{\mathbb{P}[E]}_{\leq 1} + n\delta_2 \leq \exp\left(-\frac{2\epsilon^2}{\sum_{i=1}^n c_i^2}\right) + n\delta_2,$$

where in the last step we noticed that under the event $E$, the martingale difference sequence is bounded almost surely, therefore we can apply the standard Azuma-Hoeffding inequality. $\qquad\square$

**Corollary 9.** *Let $\{Y_i\}_i^n$ be a martingale difference sequence adapted to $\mathcal{F}_i$, such that for each $i$, $|Y_i| \leq c_i$ with probability at least $1 - \delta_2$. Then, with probability $1 - \delta_1$ (with $\delta_1 > n\delta_2$), it holds that*

$$\mathbb{P}\left[\sum_{i=1}^n Y_i \geq \sqrt{\frac{\left(\sum_{i=1}^n c_i^2\right)\log\left(1/\left(\delta_1 - n\delta_2\right)\right)}{2}}\right] \leq \delta_1 + n\delta_2.$$

*Proof of Theorem 2.* We fix a policy evaluation step $k \in [K]$, i.e., we study the $k$-th iteration of the outer loop of Algorithm 1. Similarly to the proof of Lemma 19 in [89], the biased SGD subroutine can be seen as an inexact gradient ascent scheme with updates

$$\mathbf{w}_{t+1}^k = \Pi_{\mathcal{W}}\left(\mathbf{w}_t^k + \beta_t(\nabla_{\mathbf{w}} f(\mathbf{w}_t^k, \boldsymbol{\theta}_t^k) + b_{\mathbf{w},t}^k + \epsilon_{\mathbf{w},t}^k)\right), \tag{32}$$

$$\boldsymbol{\theta}_{t+1}^k = \Pi_{\Theta}\left(\boldsymbol{\theta}_t^k + \beta_t(\nabla_{\boldsymbol{\theta}} f(\mathbf{w}_t^k, \boldsymbol{\theta}_t^k) + b_{\boldsymbol{\theta},t}^k + \epsilon_{\boldsymbol{\theta},t}^k)\right), \tag{33}$$

with

$$\epsilon_{\boldsymbol{\theta},t}^k \triangleq \widehat{\nabla}_{\boldsymbol{\theta}} \mathcal{G}_k(\mathbf{w}_t^k, \boldsymbol{\theta}_t^k) - \mathbb{E}\left[\widehat{\nabla}_{\boldsymbol{\theta}} \mathcal{G}_k(\mathbf{w}_t^k, \boldsymbol{\theta}_t^k) \mid \mathcal{F}_{t-1}\right], \tag{34}$$

$$\epsilon_{\mathbf{w},t}^k \triangleq \widehat{\nabla}_{\mathbf{w}} \mathcal{G}_k(\mathbf{w}_t^k, \boldsymbol{\theta}_t^k) - \mathbb{E}\left[\widehat{\nabla}_{\mathbf{w}} \mathcal{G}_k(\mathbf{w}_t^k, \boldsymbol{\theta}_t^k) \mid \mathcal{F}_{t-1}\right], \tag{35}$$

$$b_{\boldsymbol{\theta},t}^k \triangleq \mathbb{E}\left[\widehat{\nabla}_{\boldsymbol{\theta}} \mathcal{G}_k(\mathbf{w}_t^k, \boldsymbol{\theta}_t^k) \mid \mathcal{F}_{t-1}\right] - \nabla_{\boldsymbol{\theta}} \mathcal{G}_k(\mathbf{w}_t^k, \boldsymbol{\theta}_t^k), \tag{36}$$

$$b_{\mathbf{w},t}^k \triangleq \mathbb{E}\left[\widehat{\nabla}_{\mathbf{w}} \mathcal{G}_k(\mathbf{w}_t^k, \boldsymbol{\theta}_t^k) \mid \mathcal{F}_{t-1}\right] - \nabla_{\mathbf{w}} \mathcal{G}_k(\mathbf{w}_t^k, \boldsymbol{\theta}_t^k). \tag{37}$$

By Lemma 16, and a union bound, we get that for $n(t) \geq \max\left\{\mathcal{O}\left(\frac{\gamma^2 m D^2}{\beta \xi_t^2}\log(\frac{Tm}{\delta})\right), \mathcal{O}\left(\frac{m}{\beta \xi_t^2}\log(\frac{Tm}{\delta})\right)\right\}$, with probability at least $1 - \delta/2$, for all $t = 1, \ldots, T$ simultaneously, it holds that

$$\|\epsilon_{\boldsymbol{\theta},t}^k\|_1 \leq 2m\frac{(1+38\xi_t\eta)}{\beta}(2+\xi_t) + 2(1-\gamma) \leq \frac{6m}{\beta}(1+38\eta) + 2, \tag{38}$$

$$\|\epsilon_{\mathbf{w},t}^k\|_1 \leq 2m\left(1 + \frac{1+38\eta\xi_t}{\beta}\right) \leq 2m(1 + \frac{1+38\eta}{\beta}), \tag{39}$$

$$\|b_{\boldsymbol{\theta},t}^k\|_1 \leq m\frac{\xi_t}{\beta}\left(\gamma + 38\eta\left(1 + \gamma(1+\xi_t)\right)\right) \leq \frac{m}{\beta}(1 + 114\beta), \tag{40}$$

$$\|b_{\mathbf{w},t}^k\|_1 \leq \frac{38\eta\xi_t}{\beta} \leq \frac{38\eta}{\beta}, \tag{41}$$

where we used that $\{\xi_t\}_{t=1}^T \cup \{\gamma\} \subset (0,1)$.

Moreover, by Hölder's inequality, we get

$$|\langle \epsilon_{\boldsymbol{\theta},t}^k, \boldsymbol{\theta}_t^k - \boldsymbol{\theta}_k^\star \rangle| \leq \|\epsilon_{\boldsymbol{\theta},t}^k\|_1 \|\boldsymbol{\theta}_t^k - \boldsymbol{\theta}_k^\star\|_\infty \leq \frac{12Dm}{\beta}(1+38\eta) + 2 \triangleq M_1, \tag{42}$$

$$|\langle \epsilon_{\mathbf{w},t}^k, \mathbf{w}_t^k - \mathbf{w}_k^\star \rangle| \leq \|\epsilon_{\mathbf{w},t}^k\|_1 \|\mathbf{w}_t^k - \mathbf{w}_k^\star\|_\infty \leq 2m(1 + \frac{1+38\eta}{\beta}) \triangleq M_2, \tag{43}$$

where we used that by the triangle inequality and Proposition 3, it holds that $\|\boldsymbol{\theta}_t^k - \boldsymbol{\theta}_k^\star\|_\infty \leq 2\frac{1+|\log\beta|}{1-\gamma} \triangleq 2D$. We recall that $D \triangleq \frac{1+\log\left(\frac{1}{\beta}\right)}{1-\gamma} \geq 1$.

Since $\left\{X_{\boldsymbol{\theta},t}^k \triangleq \langle \epsilon_{\boldsymbol{\theta},t}^k, \boldsymbol{\theta}_t^k - \boldsymbol{\theta}_k^\star \rangle\right\}_{t=1}^\infty$ and $\left\{X_{\mathbf{w},t}^k \triangleq \langle \epsilon_{\mathbf{w},t}^k, \mathbf{w}_t^k - \mathbf{w}_k^\star \rangle\right\}_{t=1}^\infty$ are martingale differences, by using Corollary 9 and a simple union bound, we get that with probability at least $1 - \delta/2$,

$$-\sum_{t=1}^T \langle \epsilon_{\boldsymbol{\theta},t}^k, \boldsymbol{\theta}_t^k - \boldsymbol{\theta}_k^\star \rangle \leq 2M_1\sqrt{T\log(\frac{16T}{\delta})}, \tag{44}$$

$$-\sum_{t=1}^T \langle \epsilon_{\mathbf{w},t}^k, \mathbf{w}_t^k - \mathbf{w}_k^\star \rangle \leq 2M_2\sqrt{T\log(\frac{16T}{\delta})}. \tag{45}$$

Furthermore, note that $\mathcal{G}_k$ is $\eta + \alpha$-smooth with respect to the $\|\cdot\|_\infty$-norm, and so by Lemma 12 in [89], we can bound the $\|\cdot\|_1$-norm of its gradients. In particular, we have

$$\|\nabla_{\boldsymbol{\theta}}\mathcal{G}(\mathbf{w}_t^k, \boldsymbol{\theta}_t^k)\|_1 + \|\nabla_{\mathbf{w}}\mathcal{G}(\mathbf{w}_t^k, \boldsymbol{\theta}_t^k)\|_1 \leq 2(\eta + \alpha)(D + 1). \tag{46}$$

This in turn implies that

$$\|\nabla_{\boldsymbol{\theta}}\mathcal{G}(\mathbf{w}_t^k, \boldsymbol{\theta}_t^k)\|_2^2 + \|\nabla_{\mathbf{w}}\mathcal{G}(\mathbf{w}_t^k, \boldsymbol{\theta}_t^k)\|_2^2 \leq 4(\eta + \alpha)^2(D + 1)^2. \tag{47}$$

By smoothness and concavity of the objective $\mathcal{G}_k$, we can apply Lemma 9 in [89]. In particular, by Equations (38)–(47), a union bound, and by summing over $t$ in the bound of Lemma 9 in [89], we have the following guarantee for our inexact gradient scheme:

If $n(t) \geq \max\left\{\mathcal{O}\left(\frac{\gamma^2 m D^2}{\beta \xi_t^2}\log(\frac{Tm}{\delta}),\right), \mathcal{O}\left(\frac{m}{\beta \xi_t^2}\log(\frac{Tm}{\delta}),\right)\right\}$, and $\beta_t \leq \frac{2}{\alpha + \eta}$, then with probability at least $1 - \delta$, it holds that

$$\sum_{t=1}^{T}\left(\mathcal{G}_k(\mathbf{w}_k^\star, \boldsymbol{\theta}_k^\star) - \mathcal{G}_k(\mathbf{w}_t^k, \boldsymbol{\theta}_t^k)\right) \tag{48}$$

$$\leq \sum_{t=1}^{T}\frac{\|\mathbf{w}_t^k - \mathbf{w}_k^\star\|_2^2 + \|\mathbf{w}_{t+1}^k - \mathbf{w}_k^\star\|_2^2}{2\beta_t} + \sum_{t=1}^{T}\frac{\|\boldsymbol{\theta}_t^k - \boldsymbol{\theta}_k^\star\|_2^2 + \|\boldsymbol{\theta}_{t+1}^k - \boldsymbol{\theta}_k^\star\|_2^2}{2\beta_t} \tag{49}$$

$$+ 2\sum_{t=1}^{T}\beta_t\left(\|\nabla_{\boldsymbol{\theta}}\mathcal{G}(\mathbf{w}_t^k, \boldsymbol{\theta}_t^k)\|_2^2 + \|\nabla_{\mathbf{w}}\mathcal{G}(\mathbf{w}_t^k, \boldsymbol{\theta}_t^k)\|_2^2\right) \tag{50}$$

$$+ 5\sum_{t=1}^{T}\beta_t\left(\|b_{\mathbf{w},t}^k\|_2^2 + \|b_{\boldsymbol{\theta},t}^k\|_2^2 + \|\epsilon_{\mathbf{w},t}^k\|_2^2 + \|\epsilon_{\boldsymbol{\theta},t}^k\|_2^2\right) \tag{51}$$

$$+ \sum_{t=1}^{T}\left(\|b_{\mathbf{w},t}^k\|_1 + \|b_{\boldsymbol{\theta},t}^k\|_1\right)\max\{\|\mathbf{w}_t^k - \mathbf{w}_k^\star\|_\infty, \|\boldsymbol{\theta}_t^k - \boldsymbol{\theta}_k^\star\|_\infty\} \tag{52}$$

$$- \sum_{t=1}^{T}\left\langle\epsilon_{\boldsymbol{\theta},t}^k, \boldsymbol{\theta}_t^k - \boldsymbol{\theta}_k^\star\right\rangle - \sum_{t=1}^{T}\left\langle\epsilon_{\mathbf{w},t}^k, \mathbf{w}_t^k - \mathbf{w}_k^\star\right\rangle \tag{53}$$

$$\leq \sum_{t=1}^{T}\frac{\|\mathbf{w}_t^k - \mathbf{w}_k^\star\|_2^2 + \|\mathbf{w}_{t+1}^k - \mathbf{w}_k^\star\|_2^2}{2\beta_t} + \sum_{t=1}^{T}\frac{\|\boldsymbol{\theta}_t^k - \boldsymbol{\theta}_k^\star\|_2^2 + \|\boldsymbol{\theta}_{t+1}^k - \boldsymbol{\theta}_k^\star\|_2^2}{2\beta_t} \tag{54}$$

$$+ \sum_{t=1}^{T}\left(\beta_t L_1 + 2DL_2\xi_t\right) + 2(M_1 + M_2)\sqrt{T\log(\frac{16T}{\delta})}, \tag{55}$$

where

$$L_1 = \mathcal{O}\left((\eta + \alpha)^2 D^2 + \frac{\max\{\eta, 1\}^2 m^2}{\beta^2}\right), \tag{56}$$

$$L_2 = \mathcal{O}\left(\frac{\eta + m}{\beta}\right), \tag{57}$$

$$M_1 = \mathcal{O}\left(\frac{\max\{\eta, 1\}m}{\beta}\right), \tag{58}$$

$$M_2 = \mathcal{O}\left(\frac{\max\{\eta, 1\}Dm}{\beta}\right), \tag{59}$$

$$\tag{60}$$

We choose $\beta_t = \frac{L}{\sqrt{t}}$, for some constant $L$. Then a telescoping sum gives

$$\sum_{t=1}^{T}\left(\frac{\|\mathbf{w}_t^k - \mathbf{w}_k^\star\|_2^2 + \|\mathbf{w}_{t+1}^k - \mathbf{w}_k^\star\|_2^2}{2\beta_t} + \frac{\|\boldsymbol{\theta}_t^k - \boldsymbol{\theta}_k^\star\|_2^2 + \|\boldsymbol{\theta}_{t+1}^k - \boldsymbol{\theta}_k^\star\|_2^2}{2\beta_t}\right) \leq \frac{1}{2L}(D^2 + 1)\sqrt{T}. \tag{61}$$

Moreover, $\sum_{t=1}^{T} \beta_t L_1 \leq 2L_1 L \sqrt{T}$. By combining this inequality with Equations (55) and (61), we get that

$$\sum_{t=1}^{T} \left( \mathcal{G}_k(\mathbf{w}_k^\star, \boldsymbol{\theta}_k^\star) - \mathcal{G}_k(\mathbf{w}_t^k, \boldsymbol{\theta}_t^k) \right) \leq \frac{1}{2L}(D^2 + 1)\sqrt{T} + 2L_1 L \sqrt{T} + 2DL_2 \sum_{t=1}^{T} \xi_t \quad (62)$$

$$+ 2(M_1 + M_2)\sqrt{T \log(\frac{16T}{\delta})} \quad (63)$$

The optimal choice for $L$ is $L = \frac{\sqrt{1+D^2}}{2\sqrt{L_1}}$. In addition, by setting $\xi_t = \sqrt{\frac{L_1}{t}}$, we conclude that

$$\sum_{t=1}^{T} \left( \mathcal{G}_k(\mathbf{w}_k^\star, \boldsymbol{\theta}_k^\star) - \mathcal{G}_k(\mathbf{w}_t^k, \boldsymbol{\theta}_t^k) \right) \leq 4 \max \left\{ \sqrt{1+D^2}, 2DL_2 \right\} \sqrt{L_1}\sqrt{T} \quad (64)$$

$$+ 2(M_1 + M_2)\sqrt{T \log(\frac{16T}{\delta})}. \quad (65)$$

Therefore, by combining Equations (56)–(59) and Equation (64), and by Jensen's inequality, we get that if $n(t) \geq \max \left( \mathcal{O}\left( \frac{\gamma^2 mDt}{(\eta+\alpha)^2 \beta} \log \frac{Tm}{\delta} \right), \mathcal{O}\left( \frac{mt}{\beta} \log \frac{Tm}{\delta} \right) \right)$, and $\beta_t = \mathcal{O}(\frac{1}{\sqrt{t}})$, then, with probability at least $1 - \delta$, it holds that $\mathcal{G}_k(\mathbf{w}_k^\star, \boldsymbol{\theta}_k^\star) - \mathcal{G}_k(\mathbf{w}_k, \boldsymbol{\theta}_k) \leq \mathcal{O}\left( \frac{\max\{\eta,1\}mD}{\beta\sqrt{T}} \right)$.

$\square$

## I.1   Proof of Corollary 1

*Proof of Corollary 1.* We plug the upper bound for $\epsilon_k$ given by Theorem 2 in the error propagation analysis of Theorem 1. In particular, from Theorem 1, with probability at least $1 - \delta_1$, it holds that

$$d_{\mathcal{C}}(\widehat{\pi}, \pi_{\mathrm{E}}) \leq \frac{1}{K}\left( \frac{D(\boldsymbol{\lambda}^*||\boldsymbol{\Phi}^\mathsf{T}\mathbf{d}_0)}{\eta} + \frac{H(\mathbf{d}^*||\mathbf{d}_0)}{\alpha} + C(\eta,\alpha) \sum_k \sqrt{\epsilon_k} + \sum_k \epsilon_k \right) + \varepsilon.$$

where we replaced we made explicit the fact the constant (wrt to $K$ and $T$) $C$ depends on $\alpha$ and $\eta$ (See Theorem 1 for the exact expression). By plugging in the bound for $\epsilon_k$ given by Theorem 2, and a union bound, we get that and if we use $n(t) \geq \max \left( \mathcal{O}\left( \frac{\gamma^2 mDt}{\beta} \log \frac{Tm}{\delta_2} \right), \mathcal{O}\left( \frac{mt}{\beta} \log \frac{Tm}{\delta_2} \right) \right)$ samples per iteration, then with probability at least $1 - \delta_1 - \delta_2$, it holds that

$$d_{\mathcal{C}}(\widehat{\pi}, \pi_{\mathrm{E}}) \leq \frac{1}{K}\left( \frac{D(\boldsymbol{\lambda}^*||\boldsymbol{\Phi}^\mathsf{T}\mathbf{d}_0)}{\eta} + \frac{H(\mathbf{d}^*||\mathbf{d}_0)}{\alpha} + C(\eta,\alpha) \sum_k \mathcal{O}\left( \sqrt{\frac{\eta mD}{\beta\sqrt{T}}} \right) + \sum_k \mathcal{O}\left( \frac{\eta mD}{\beta\sqrt{T}} \right) \right) + \varepsilon.$$

Setting $\eta = \alpha = 1$, letting $C_1 \triangleq C(1,1)$ and keeping only the dominant terms, we obtain

$$d_{\mathcal{C}}(\widehat{\pi}, \pi_{\mathrm{E}}) \leq \frac{D(\boldsymbol{\lambda}^*||\boldsymbol{\Phi}^\mathsf{T}\mathbf{d}_0) + H(\mathbf{d}^*||\mathbf{d}_0)}{K} + \mathcal{O}\left( C_1 \frac{mD}{\beta\sqrt[4]{T}} \right) + \varepsilon$$

Then, choosing $K = \frac{D(\boldsymbol{\lambda}^*||\boldsymbol{\Phi}^\mathsf{T}\mathbf{d}_0) + H(\mathbf{d}^*||\mathbf{d}_0)}{\epsilon}$ and $T = \Omega\left( \frac{m^4 D^4}{\beta^4 C_1^4 \epsilon^4} \right)$, we can ensure that $d_{\mathcal{C}}(\widehat{\pi}, \pi_{\mathrm{E}}) \leq \epsilon + \varepsilon$. The overall sample complexity is $Kn(T) = \Omega(KT) = \Omega(\epsilon^{-5})$. Notice that the corollary improves upon the sample complexity bound of $\Omega(\epsilon^{-8})$ derived in [89].   $\square$

## J   Offline imitation learning version

Inspecting Equation (5), one can notice that estimating the empirical logistic Bellman evaluation objective $\mathcal{G}_k$ or its gradients requires sampling from $\mathbf{d}_{k-1}$. Hence, the algorithm needs interactions with the environment at every iteration $k$. It is possible to alleviate this requirement, changing the center point for the relative entropy. This is akin to smoothing [82] choosing a convenient center point. In particular, we replace Equation (2) with the following update:

$$(\boldsymbol{\lambda}_1, \mathbf{d}_1) = \underset{\boldsymbol{\lambda}\in\Delta_{[m]}, \mathbf{d}\in\Delta_{\mathcal{S}\times\mathcal{A}}}{\arg\min} \left\langle \mathbf{y}^\star, \mathbf{A}\begin{bmatrix} \boldsymbol{\lambda} \\ \mathbf{d} \end{bmatrix} + \widehat{\mathbf{b}} \right\rangle + \frac{1}{\eta}D(\boldsymbol{\lambda}||\boldsymbol{\Phi}^\mathsf{T}\boldsymbol{\mu}_{\pi_{\mathrm{E}}}) + \frac{1}{\alpha}H(\mathbf{d}||\mathbf{d}_0). \quad (66)$$

**Algorithm 3** Offline Proximal Point Imitation Learning ($\mathtt{OP^2IL}$)

---

**Input:** Feature matrix $\boldsymbol{\Phi}$, number of iterations $K$, step sizes $\eta$, $\alpha$, and $\beta$

**Input:** Expert demonstrations $\mathcal{D}_{\mathrm{E}}^{n_{\mathrm{E}},H}$

○ Initialize $\pi_0$ as uniform distribution over $\mathcal{A}$, and set $\mathbf{w}_0 = \frac{1}{m}\mathbf{1}$

○ Compute the empirical FEV $\widehat{\boldsymbol{\rho}_{\boldsymbol{\Phi}}}(\pi_{\mathrm{E}})$ using expert demonstrations

○ Sample $\{(s^{(n)}, a^{(n)}, s'^{(n)})\}_{n=1}^N$ with $s^{(n)}, a^{(n)}$ sampled i.i.d. from $\boldsymbol{\mu}_{\pi_E}$ and $s'^{(n)} \sim P(\cdot|s^{(n)}, a^{(n)})$ and compute the empirical offline logistic Bellman error by

$$\widehat{\mathcal{G}}(\mathbf{w}, \boldsymbol{\theta}) = -\langle \boldsymbol{\rho}_{\boldsymbol{\Phi}}(\widehat{\pi_{\mathrm{E}}}), \mathbf{w} \rangle - \frac{1}{\eta} \log\left(\frac{1}{N}\sum_{n=1}^N e^{-\eta\widehat{\boldsymbol{\delta}}_{\mathbf{w},\boldsymbol{\theta}}(s^{(n)}, a^{(n)}, s'^{(n)})}\right) + (1-\gamma)\langle \boldsymbol{\nu}_0, V_{\boldsymbol{\theta}}\rangle$$

`// policy evaluation & cost update`
○ Find an approximate maximizer of the negative empirical logistic Bellman error

$$(\mathbf{w}_1, \boldsymbol{\theta}_1) \approx \operatorname{argmax}_{\mathbf{w},\boldsymbol{\theta}} \widehat{\mathcal{G}}(\mathbf{w}, \boldsymbol{\theta})$$

`// policy improvement`
Policy update:
$$\pi_{\mathbf{d}_1}(a|s) \propto \pi_{\mathbf{d}_0}(a|s)\, e^{-\alpha Q_{\boldsymbol{\theta}_1}(s,a)}$$

**Output:** Policy $\pi_{\mathbf{d}_1}$

---

Note that we have removed the iteration index $k$, since the offline version does not require to iteratively collect new samples from the environment. Changing the reference distribution from $\boldsymbol{\Phi}^\intercal \mathbf{d}_k$ to $\boldsymbol{\Phi}^\intercal \boldsymbol{\mu}_{\pi_E}$ gives Algorithm 3. In this case, the logistic Bellman evaluation objective takes the form

$$\mathcal{G}(\mathbf{w}, \boldsymbol{\theta}) \triangleq -\frac{1}{\eta}\log\sum_{i=1}^m (\boldsymbol{\Phi}^\intercal \boldsymbol{\mu}_{\pi_E})(i) e^{-\eta\delta_{\mathbf{w},\boldsymbol{\theta}}(i)} + (1-\gamma)\langle \boldsymbol{\nu}_0, \mathbf{V}_{\boldsymbol{\theta}}\rangle - \langle \boldsymbol{\rho}_{\boldsymbol{\Phi}}(\widehat{\pi_{\mathrm{E}}}), \mathbf{w}\rangle, \qquad (67)$$

The difference with the online variant is that in the first term we have the expert occupancy measure instead of the occupancy measure induced by the current policy. We describe the corresponding empirical estimate in Algorithm 3. Furthermore, we suppress the index $k$, since the offline algorithm does not require multiple iterations.

### J.1 Theoretical guarantees for the offline case

With minor modifications of the error propagation analysis given in Theorem 1, one can prove the following result.

**Theorem 4.** *Under the same assumptions as in Theorem 1, and by choosing* $\alpha = \left(\frac{2H(\mathbf{d}^\star\|\mathbf{d}_0)}{3w_{\max}}\sqrt{\frac{1-\gamma}{2\epsilon}}\right)^{2/3}$, *we obtain*

$$d_{\mathcal{C}}(\pi_{\mathrm{E}}, \pi_{\mathbf{d}_1}) - d_{\mathcal{C}}(\pi_{\mathrm{E}}, \pi_{\mathbf{d}^\star}) \le \frac{D(\boldsymbol{\lambda}^\star\|\boldsymbol{\Phi}^\intercal\boldsymbol{\mu}_{\pi_E})}{\eta} + \left(\frac{243 H(\mathbf{d}^\star\|\mathbf{d}_0)w_{\max}^2}{2(1-\gamma)}\right)^{1/3}\epsilon_1^{1/3} + \epsilon_1 + \varepsilon. \quad (68)$$

*where $\epsilon_1$ is the error in the maximization of the logistic Bellman error, i.e.* $\epsilon = \max_{\mathbf{w}\in\mathcal{W},\boldsymbol{\theta}}\mathcal{G}(\mathbf{w}, \boldsymbol{\theta}) - \mathcal{G}(\mathbf{w}_1, \boldsymbol{\theta}_1)$ *and $\varepsilon$ is the error in estimating the expert feature expectation vector as in Lemma 6.*

*Proof.* Following exactly the same steps in the proof of Theorem 1 for the special case of $K = 1$, we get

$$d_{\mathcal{C}}(\pi_E, \pi_{\mathbf{d}_1}) - d_{\mathcal{C}}(\pi_E, \pi_{\mathbf{d}^\star}) \le \frac{D(\boldsymbol{\lambda}^\star\|\boldsymbol{\Phi}^\intercal\boldsymbol{\mu}_{\pi_E})}{\eta} + \frac{H(\mathbf{d}^\star\|\mathbf{d}_0)}{\alpha} + 3w_{\max}\|\mathbf{d}_1 - \mathbf{d}_1^\star\|_1 + \epsilon_1 + \varepsilon. \quad (69)$$

By using the bound $\|\mathbf{d}_1 - \mathbf{d}_1^\star\|_1 \le \sqrt{\frac{2\alpha\epsilon_1}{1-\gamma}}$, we have

$$d_{\mathcal{C}}(\pi_E, \pi_{\mathbf{d}_1}) - d_{\mathcal{C}}(\pi_E, \pi_{\mathbf{d}^\star}) \le \frac{D(\boldsymbol{\lambda}^\star\|\boldsymbol{\Phi}^\intercal\boldsymbol{\mu}_{\pi_E})}{\eta} + \frac{H(\mathbf{d}^\star\|\mathbf{d}_0)}{\alpha} + 3w_{\max}\sqrt{\frac{2\alpha\epsilon_1}{1-\gamma}} + \epsilon_1 + \varepsilon. \quad (70)$$

Therefore, by choosing $\alpha$ as stated in the theorem we conclude the proof. □

Notice that if the expert is nearly optimal, the step size $\eta$ can be taken small, ensuring low bias in the gradients. This allows to use the original empirical logistic Bellman error analysis, proposed in [14], where one can control the bias by choosing $\eta$ appropriately small. To this end, we need to relate the logistic bellman error in the feature space to the one in the state-action space. As we will show, this introduces an additional bias of order $\mathcal{O}(\eta)$. The statement is made precise in Theorem 5. Thanks to this result and Theorem 2 in [14], we have that $\epsilon \leq (8 + e)\eta B^2 + 56\sqrt{\frac{m \log (1+4BN)\delta}{N}}$ where $N$ is the number of expert transitions in the dataset. We have the following result.

**Corollary 10.** *Let* $C_1 = \left(\frac{243H(\mathbf{d}^\star||\mathbf{d}_0)w_{\max}^2}{2(1-\gamma)}\right)^{1/3}$, $\eta = \mathcal{O}\left(\frac{D(\boldsymbol{\lambda}^\star||\boldsymbol{\Phi}^\intercal\boldsymbol{\mu}_{\pi_E})^{3/4}}{(C_1 B)^{1/4}}\right)$ *and* $N = \tilde{\mathcal{O}}(m\epsilon^{-6}\log(1/\delta))$. *Then, with probability* $1 - \delta$, *it holds that*

$$d_{\mathcal{C}}(\pi_E, \pi_{\mathbf{d}_1}) - d_{\mathcal{C}}(\pi_E, \pi_{\mathbf{d}^\star}) \leq \mathcal{O}\left(C_1^{1/4}B^{1/4}D(\boldsymbol{\lambda}^\star||\boldsymbol{\Phi}^\intercal\boldsymbol{\mu}_{\pi_E})^{1/4}\right) + \mathcal{O}(\epsilon). \tag{71}$$

**Remark 2.** *We notice that the optimal choice of* $\eta$ *is smaller as the expert is closely optimal, i.e.* $D(\boldsymbol{\lambda}^\star||\boldsymbol{\Phi}^\intercal\boldsymbol{\mu}_{\pi_E})$ *is small. In this condition, we can use the empirical objective estimator proposed in[14] ensuring small bias. This means that estimating the objective from sample is feasible in the offline setting. It is still an open question if this is viable for the online setting improving the error propagation analysis.*

Next, we present an important result showing that it is possible to replace the minimization of $\mathcal{G}$, with its counterpart in the state-action space defined as

$$\mathcal{G}^{\mathcal{S},\mathcal{A}}(\boldsymbol{\theta}, \mathbf{w}) = -\frac{1}{\eta}\log\sum_{s,a}\boldsymbol{\mu}_{\pi_E}(s,a)e^{-\eta\delta_{\mathbf{w},\boldsymbol{\theta}}^{\mathcal{S},\mathcal{A}}(s,a)} + (1-\gamma)\langle\boldsymbol{\nu}_0, \mathbf{V}_{\boldsymbol{\theta}}\rangle - \langle\boldsymbol{\rho}_{\boldsymbol{\Phi}}(\widehat{\pi_E}), \mathbf{w}\rangle,$$

where we introduced $\boldsymbol{\delta}_{\mathbf{w},\boldsymbol{\theta}}^{\mathcal{S},\mathcal{A}} = \boldsymbol{\Phi}\boldsymbol{\delta}_{\mathbf{w},\boldsymbol{\theta}}$.

**Theorem 5.** *Let* $B \triangleq 1 + 2\frac{1+|\log\beta|}{1-\gamma}$. *Suppose* $\eta$ *is chosen such that* $\eta B \leq 1$. *Then, it holds that*

$$\left|\mathcal{G}(\boldsymbol{\theta}, \mathbf{w}) - \mathcal{G}^{\mathcal{S},\mathcal{A}}(\boldsymbol{\theta}, \mathbf{w})\right| \leq e\eta B^2.$$

*Proof.* From Proposition 3, we have that $\|\boldsymbol{\theta}\|_\infty \leq \frac{1+|\log\beta|}{1-\gamma}$ and $\|\mathbf{V}_{\boldsymbol{\theta}}\|_\infty \leq \frac{1+|\log\beta|}{1-\gamma}$, for all $\boldsymbol{\theta} \in \mathbb{R}^m$. It follows that for any $(\mathbf{w}, \boldsymbol{\theta}) \in \mathcal{W} \times \mathbb{R}^m$, it holds that $\|\boldsymbol{\delta}_{\boldsymbol{\theta},\mathbf{w}}\|_\infty = \|\mathbf{w} + \gamma\mathbf{M}\mathbf{V}_{\boldsymbol{\theta}} - \boldsymbol{\theta}\|_\infty \leq 1 + 2\frac{1+|\log\beta|}{1-\gamma} = B$. Hence, it holds that $\eta\|\boldsymbol{\delta}_{\boldsymbol{\theta},\mathbf{w}}\|_\infty \leq \eta B \leq 1$. First, we recall the assumption that the rows of $\boldsymbol{\Phi}$ are probability distributions, i.e., $\phi(s,a) \in \Delta_{[m]}$, for all $(s,a)$. We then have

$$\boldsymbol{\delta}_{\mathbf{w},\boldsymbol{\theta}}^{\mathcal{S},\mathcal{A}}(s,a) = (\boldsymbol{\Phi}\boldsymbol{\delta}_{\mathbf{w},\boldsymbol{\theta}})(s,a) = \sum_{i=1}^m\phi_i(s,a)\boldsymbol{\delta}_{\mathbf{w},\boldsymbol{\theta}}(i) = \mathop{\mathbb{E}}_{i\sim\phi(s,a)}[\boldsymbol{\delta}_{\mathbf{w},\boldsymbol{\theta}}(i)]. \tag{72}$$

Moreover, we have

$$\mathcal{G}(\mathbf{w}, \boldsymbol{\theta}) - \mathcal{G}^{\mathcal{S},\mathcal{A}}(\mathbf{w}, \boldsymbol{\theta}) = \underbrace{-\frac{1}{\eta}\log\left(\sum_{i=1}^m(\boldsymbol{\Phi}^\intercal\boldsymbol{\mu}_{\pi_E})(i)e^{-\eta\delta_{\mathbf{w},\boldsymbol{\theta}}(i)}\right)}_{\triangleq W} + \underbrace{\frac{1}{\eta}\log\left(\sum_{s,a}\boldsymbol{\mu}_{\pi_E}(s,a)e^{-\eta\delta_{\mathbf{w},\boldsymbol{\theta}}^{\mathcal{S},\mathcal{A}}(s,a)}\right)}_{\triangleq W^{\mathcal{S},\mathcal{A}}}$$

We can then lower bound $W$ as

$$\begin{aligned}
W &= \frac{1}{\eta}\log\left(\sum_{i=1}^m\sum_{s,a}\phi_i(s,a)\boldsymbol{\mu}_{\pi_E}(s,a)e^{-\eta\delta_{\mathbf{w},\boldsymbol{\theta}}(i)}\right) \\
&= \frac{1}{\eta}\log\left(\mathop{\mathbb{E}}_{(s,a)\sim\boldsymbol{\mu}_{\pi_E}}\left[\mathop{\mathbb{E}}_{i\sim\phi(s,a)}\left[e^{-\eta\delta_{\mathbf{w},\boldsymbol{\theta}}(i)}\right]\right]\right) \\
&\geq \frac{1}{\eta}\log\left(\mathop{\mathbb{E}}_{(s,a)\sim\boldsymbol{\mu}_{\pi_E}}\left[e^{-\eta\mathop{\mathbb{E}}_{i\sim\phi(s,a)}[\boldsymbol{\delta}_{\mathbf{w},\boldsymbol{\theta}}(i)]}\right]\right), \\
&= W^{\mathcal{S},\mathcal{A}},
\end{aligned}$$

where the inequality follows by Jensen's inequality for expectations.

We will now upper bound the term $W$. Thanks to the choice of $\eta$ such that $\eta B \leq 1$, we have that $\eta \leq \frac{1}{B} \leq \frac{1}{|\boldsymbol{\delta}_{\mathbf{w},\boldsymbol{\theta}}(i)|}$, for all $i$. Therefore, we can apply the inequality $e^x \leq 1 + x + x^2$ for $x = -\eta \boldsymbol{\delta}_{\mathbf{w},\boldsymbol{\theta}}^{\mathcal{SA}}(i) \leq 1$ and obtain

$$
\begin{aligned}
\mathop{\mathbb{E}}_{i\sim\phi(s,a)}\left[e^{-\eta\boldsymbol{\delta}_{\mathbf{w},\boldsymbol{\theta}}(i)}\right] &\leq \mathop{\mathbb{E}}_{i\sim\phi(s,a)}\left[1 - \eta\boldsymbol{\delta}_{\mathbf{w},\boldsymbol{\theta}}(i) + (\eta\boldsymbol{\delta}_{\mathbf{w},\boldsymbol{\theta}}(i))^2\right] \\
&\leq 1 - \mathop{\mathbb{E}}_{i\sim\phi(s,a)}\left[\eta\boldsymbol{\delta}_{\mathbf{w},\boldsymbol{\theta}}(i)\right] + (\eta B)^2 \\
&= 1 - \eta\boldsymbol{\delta}_{\mathbf{w},\boldsymbol{\theta}}^{\mathcal{SA}}(s,a) + (\eta B)^2 \\
&\leq e^{-\eta\boldsymbol{\delta}_{\mathbf{w},\boldsymbol{\theta}}^{\mathcal{SA}}(s,a)} + (\eta B)^2,
\end{aligned}
$$

where in the third line we used Equation (72), and in the last line we used the inequality $1 - x \leq e^{-x}$ for $x = \boldsymbol{\delta}_{\mathbf{w},\boldsymbol{\theta}}^{\mathcal{SA}}(s,a)$. By taking expectations with respect to $\boldsymbol{\mu}_{\pi_E}$ and logarithms on both sides, we get

$$
W^{\mathcal{SA}} \leq W \leq \frac{1}{\eta}\log\mathop{\mathbb{E}}_{(s,a)\sim\boldsymbol{\mu}_{\pi_E}}\left[e^{-\eta\boldsymbol{\delta}_{\mathbf{w},\boldsymbol{\theta}}^{\mathcal{SA}}(s,a)} + (\eta B)^2\right].
$$

Subtracting $W$ yields

$$
\begin{aligned}
0 \leq W - W^{\mathcal{SA}} &\leq \frac{1}{\eta}\log\mathop{\mathbb{E}}_{(s,a)\sim\boldsymbol{\mu}_{\pi_E}}\left[e^{-\eta\boldsymbol{\delta}_{\mathbf{w},\boldsymbol{\theta}}^{\mathcal{SA}}(s,a)} + (\eta B)^2\right] - W^{\mathcal{SA}} \\
&= \frac{1}{\eta}\log\left(1 + \frac{(\eta B)^2}{\mathop{\mathbb{E}}_{(s,a)\sim\boldsymbol{\mu}_{\pi_E}}\left[e^{-\eta\boldsymbol{\delta}_{\mathbf{w},\boldsymbol{\theta}}^{\mathcal{SA}}(s,a)}\right]}\right) \\
&\leq \frac{\eta B^2}{\mathop{\mathbb{E}}_{(s,a)\sim\boldsymbol{\mu}_{\pi_E}}\left[e^{-\eta\boldsymbol{\delta}_{\mathbf{w},\boldsymbol{\theta}}^{\mathcal{SA}}(s,a)}\right]} \\
&\leq \frac{\eta B^2}{\mathop{\mathbb{E}}_{(s,a)\sim\boldsymbol{\mu}_{\pi_E}}\left[e^{-\eta B}\right]} \\
&\leq e\eta B^2,
\end{aligned}
$$

where in the third line we used the inequality $\log(1 + x) \leq x$ for $x = \frac{(\eta B)^2}{\mathop{\mathbb{E}}_{(s,a)\sim\boldsymbol{\mu}_{\pi_E}}\left[e^{-\eta\boldsymbol{\delta}_{\mathbf{w},\boldsymbol{\theta}}^{\mathcal{SA}}(s,a)}\right]}$, while

in the last line we used that $\eta B \leq 1$. This concludes the proof. $\square$

After having established with Theorem 5 that $\mathcal{G}^{\mathcal{S},\mathcal{A}}$ can be used as biased estimate of $\mathcal{G}$, we can proceed as in [14]. In particular, we maximize the empirical objective $\widehat{\mathcal{G}}$ (see Algorithm 3) that is a biased estimate of $\mathcal{G}^{\mathcal{S},\mathcal{A}}$ ([14, Theorem 2]). Then, we compute unbiased gradients of $\widehat{\mathcal{G}}$, recurring to the Donsker-Varadhan formula [18, Corollary 4.15] that implies the following result.

**Theorem 6.** *Given a batch of expert data* $\{\widetilde{S}_n, \widetilde{A}_n, \widetilde{S}_n'\}_{n=1}^N \sim \boldsymbol{\mu}_{\pi_E} \times \mathbf{P}$, *the following is true:*

$$
\max_{\theta}\max_{w}\widehat{\mathcal{G}}(\theta, w) = \max_{\theta}\max_{w}\min_{z}\mathcal{S}(\theta, w, z) \tag{73}
$$

*with:*

$$
\mathcal{S}(\theta, w, z) = -\frac{1}{N}\sum_{n=1}^N \boldsymbol{\mu}_{\pi_E}(\widetilde{S}_n, \widetilde{A}_n)\sum_{i=1}^m \mathbf{w}_i\phi_i(\widetilde{S}_n, \widetilde{A}_n) \tag{74}
$$

$$
+ \frac{1}{N}\sum_{n=1}^N z(n)\left(\widehat{\delta}_{\mathbf{w},\boldsymbol{\theta}}(\widetilde{S}_n, \widetilde{A}_n, \widetilde{S}_n') + \frac{1}{\eta}\log(Nz(n))\right) \tag{75}
$$

$$
+ (1 - \gamma)\langle\boldsymbol{\nu}_0, \mathbf{V}_{\boldsymbol{\theta}}\rangle \tag{76}
$$

*and the minimum attained at* $z^\star \propto \frac{1}{N}e^{-\eta\widehat{\delta}_{\mathbf{w},\boldsymbol{\theta}}(\widetilde{S}_n, \widetilde{A}_n, \widetilde{S}_n')}$

Hence, in the deep learning implementation we update the cost and the value networks backpropagating through $\mathcal{S}(\boldsymbol{\theta}, \mathbf{w}, z^\star)$.

## J.2 Practical implementation

We test a practical relaxation of Algorithm 3 that uses two separate neural networks for cost and value function approximation. We use a two layers neural network with $128$ units per layer with `ReLu` activation for the `CartPole-v1` environment. Whereas, for `Acrobot-v1` and `LunarLander-v2` we used a 3 layers architecture with $64$ units per layer.

# K Mirror Descent versus Proximal Point

To highlight an important message of our work, in this section, we briefly discuss a mirror descent scheme with alternating updates, and we compare it to our proximal point algorithm in Figure 6. Note that in contrast to the classical RL setting, where proximal point and mirror descent coincide because of the linear objective, in imitation learning this is not the case.

The updates for the mirror descent scheme involve alternation between updating the occupancy measure $\mathbf{d}_k$ and the feature expectation vector $\boldsymbol{\lambda}_k$ in one stage and the cost weights in a second stage. That is,

$$(\boldsymbol{\lambda}_k, \mathbf{d}_k) = \underset{(\boldsymbol{\lambda}, \mathbf{d}) \in \mathcal{M}_{\boldsymbol{\Phi}}}{\arg\min} \langle \boldsymbol{\mu}, \mathbf{c}_{\mathbf{w}_k} \rangle + \tfrac{1}{\eta} D(\boldsymbol{\lambda} || \boldsymbol{\Phi}^{\mathsf{T}} \mathbf{d}_{k-1}) + \tfrac{1}{\alpha} H(\mathbf{d} || \mathbf{d}_{k-1}), \tag{77}$$

$$\mathbf{w}_{k+1} = \underset{\mathbf{w} \in \Delta_{[m]}}{\arg\min} \langle \boldsymbol{\mu}_{\pi_{\mathrm{E}}} - \mathbf{d}_k, \mathbf{c}_{\mathbf{w}} \rangle + \tfrac{1}{\beta} D(\mathbf{w} || \mathbf{w}_k). \tag{78}$$

One can notice that the update in Equation (77) corresponds to one update of Logistic $Q$-Learning [14]. Therefore, it can be implemented by maximizing the negative logistic Bellman error that is now a function only of the variable $\boldsymbol{\theta}$ and not of both $(\boldsymbol{\theta}, \mathbf{w})$ as in PPM. The next proposition is the counterpart of Proposition 2 for the mirror descent scheme.

**Proposition 5.** *For a parameter* $\boldsymbol{\theta} \in \mathbb{R}^m$, *we define the state-action logistic value function* $\mathbf{Q}_{\boldsymbol{\theta}} \in \mathbb{R}^{|\mathcal{S}||\mathcal{A}|}$ *by* $\mathbf{Q}_{\boldsymbol{\theta}} \triangleq \boldsymbol{\Phi}\boldsymbol{\theta}$, *and the* $k$*-step state logistic value function* $\mathbf{V}_{\boldsymbol{\theta}}^k \in \mathbb{R}^{|\mathcal{S}|}$ *by*

$$V_{\boldsymbol{\theta}}^k(s) \triangleq -\frac{1}{\alpha} \log \left( \sum_a \pi_{\mathbf{d}_{k-1}}(a|s) e^{-\alpha Q_{\boldsymbol{\theta}}(s,a)} \right).$$

*Moreover, for a fixed cost* $\mathbf{c} = \mathbf{c}_{\mathbf{w}}$, *we define the* $k$*-step Bellman error function* $\delta_{\boldsymbol{\theta},\mathbf{w}}^k$ *by* $\delta_{\boldsymbol{\theta},\mathbf{w}}^k \triangleq \mathbf{w} + \gamma \mathbf{M} \mathbf{V}_{\boldsymbol{\theta}}^k - \boldsymbol{\theta}$. *Then, the unique solution of the aforementioned problem is given by*

$$\lambda_k(i) \propto (\boldsymbol{\Phi}^{\mathsf{T}} \mathbf{d}_{k-1})(i) \, e^{-\eta \delta_{\boldsymbol{\theta}_k,\mathbf{w}_k}^k(i)}, \tag{79}$$

$$\pi_{\mathbf{d}_k}(a|s) \propto \pi_{\mathbf{d}_{k-1}}(a|s) \, e^{-\alpha Q_{\boldsymbol{\theta}_k}(s,a)}, \tag{80}$$

$$w_{k+1,i} \propto w_{k,i} \, e^{-\beta \langle \boldsymbol{\phi}_i, \boldsymbol{\mu}_{\pi_{\mathrm{E}}} - \mathbf{d}_k \rangle}, \tag{81}$$

*where* $\boldsymbol{\theta}_k$ *is the maximizer of the negative* $k$*-step logistic Bellman error function*

$$\mathcal{G}_k(\boldsymbol{\theta}) \triangleq -\frac{1}{\eta} \log \sum_{i=1}^m (\boldsymbol{\Phi}^{\mathsf{T}} \mathbf{d}_{k-1})(i) e^{-\eta \delta_{\boldsymbol{\theta},\mathbf{w}_k}^k(i)} + (1 - \gamma) \langle \boldsymbol{\nu}_0, \mathbf{V}_{\boldsymbol{\theta}}^k \rangle.$$

Proposition 5 leads to an actor critic scheme that has three separate and alternating updates: (i) policy update stage, (ii) policy evaluation update, and (iii) cost weights update. Similar actor critic-schemes for different MDP models, and different policy evaluation objectives (e.g., minimizing the squared Bellman error) have been also proposed in [122, 70, 105]. Contrary to these schemes, in our proximal imitation learning algorithm, the policy evaluation step involves optimization of a single objective over both cost and $Q$-functions. In this way, we avoid instability or poor convergence in optimization due to nested policy evaluation and cost update steps. In section L.5, we verify numerically that PPM outperforms Mirror Descent in simple tabular environments (see Figure 6).

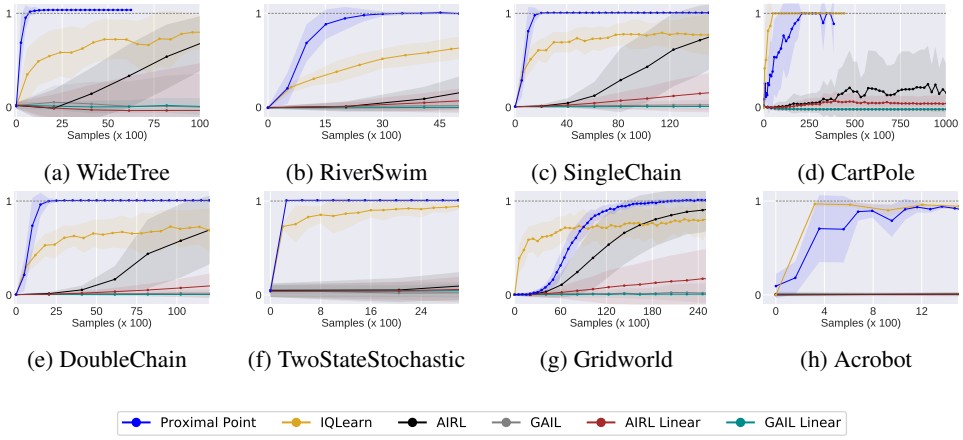

Figure 5: **Extended Online IL Experiments**. We show the total returns vs the number of env steps. We report the results of some environments omitted in the main text.

## L    Experimental Details

### L.1    Refereences for environments description

In the tabular case we used the environments (`DoubleChain` [39], `SingleChain` [39], `RiverSwim` [108], `WideTree` [8], `Two States Deterministic` [9], `Two States Stochastic` [14] and `WindyGrid` [110]). While for the offline setting, we used the environments `CartPole` [12], `Acrobot` [42] and `LunarLander` [19]. The curves are averaged over 50 seeds. For the environments `Cartpole` and `Acrobot`, we used a three layer neural network to approximate the value function. In these cases we averaged 5 seeds.

### L.2    Hyperparameters

We report the hyperparameters for the tabular online experiments in Table 1 and for the offline experiments in Table 2

| Environment | n-trajs | lr w | lr $\boldsymbol{\theta}$ | $\eta$ | $\alpha$ | optimizer |
|---|---|---|---|---|---|---|
| **TwoStateStochastic-v0** | 25 | 0.5 | 0.5 | 10 | 1 | FoRB |
| **TwoStateStochastic-v0** | 25 | 0.5 | 0.5 | 10 | 1 | Adam |
| **WideTree-v0** | 25 | 0.5 | 0.5 | 10 | 1 | FoRB |
| **RiverSwim-v0** | 50 | 0.2 | 0.2 | 10 | 1 | FoRB |
| **WindyGrid-v0** | 50 | 0.5 | 0.01 | 10 | 1 | FoRB |
| **SingleChainProblem-v0** | 50 | 0.3 | 0.005 | 10 | 1 | Adam |
| **DoubleChainProblem-v0** | 50 | 0.5 | 0.005 | 10 | 1 | Adam |

Table 1: Hyperparameters for proximal point imitation learning in tabular experiments. `FoRB` stands for Forward Reflected Backward [72].

### L.3    On the data sampling

In all the experiments, we perform a relaxation of our theoretical scheme. In particular, to increase the sample efficiency we sample state action pairs from the Markovian stream of experience. Analyzing this setting is an open problem.

| Environment | lr w | lr $\boldsymbol{\theta}$ | $\eta$ | $\alpha$ | optimizer |
|---|---|---|---|---|---|
| **CartPole-v1** | $5e-3$ | $5e-3$ | 10 | 1 | Adam |
| **Acrobot-v1** | $5e-3$ | $5e-3$ | 10 | 1 | Adam |
| **LunarLander-v2** | $1e-4$ | $1e-4$ | 10 | 0.01 | Adam |

Table 2: Hyperparameters for offline experiments

| Environment | n-trajs | lr **w** | lr $\boldsymbol{\theta}$ | $\eta$ | $\alpha$ |
|---|---|---|---|---|---|
| **TwoStateStochastic-v0** | 25 | 0.5 | 0.5 | 10 | 1 |
| **TwoStateProblem-v0** | 25 | 0.5 | 0.5 | 10 | 1 |
| **WideTree-v0** | 25 | 0.5 | 0.5 | 10 | 1 |
| **RiverSwim-v0** | 25 | 0.5 | 0.01 | 10 | 1 |
| **WindyGrid-v0** | 50 | 0.5 | 0.0006 | 10 | 1 |
| **SingleChainProblem-v0** | 50 | 0.03 | 0.05 | 10 | 1 |
| **DoubleChainProblem-v0** | 50 | 0.03 | 0.025 | 10 | 1 |

Table 3: Hyperparameters for primal dual mirror descent imitation learning in tabular experiments. As optimizer, we used OGD in all cases.

## L.4  Offline experiments setting

We consider a training environment and a test environment with different random seeds. We train both IQLearn and Proximal Point for $2e5$ environment steps and we evaluate the policy running 10 episodes on the evaluation environment every $1e3$ steps. We report the maximum evaluation result achieved at the end of training. We average the seeds from 0 to 10 for the results shown in Figure 2. We use two separate instances of the same architecture as function approximation for the $Q$-values and cost respectively. Finally, since the algorithm operates offline it has no access to the distribution $\boldsymbol{\nu}_0$. In order to approximate the term $\langle \boldsymbol{\nu}_0, \mathbf{V} \rangle$, we use the Bellman flow constraints and the fact that the expert occupancy measure is feasible, i.e. $(1-\gamma)\langle \boldsymbol{\nu}_0, \mathbf{V} \rangle = \langle \boldsymbol{\mu}_{\pi_E}, -\gamma \mathbf{PV} + \mathbf{BV} \rangle$ where the last term can be estimated from the expert samples.

## L.5  Comparison with mirror descent

We designed also a mirror descent scheme with alternating updates for imitation learning, briefly described in Appendix K. The best hyperparameters are given in Table 3. Furthermore, we show a comparison with our proximal point scheme in Figure 6. It is interesting to notice that mirror descent and proximal point have been used interchangeably in the RL literature. Indeed, in that case the objective is linear therefore the two algorithms coincide. However, when considering the max-form objective in imitation learning the equivalence between mirror descent and proximal point does not longer hold true. We verify numerically that PPM outperforms mirror descent in simple tabular environments (see Figure 6).

## L.6  Hyperparameters for Pong (Atari)

We use a convolutional neural network to learn the $Q$ values instead of the linear function approximation class we considered in the theoretical analysis. We set the parameter $\alpha$ to $1e-3$ and $\eta$ to $8e-2$, we used expert samples to approximate expectation with respect to the initial distribution. For optimizing the network we used Adam [61] with learning rate $1e-4$ and defaults value for $\beta_1, \beta_2$ Instead of hard constraints on the euclidean norm of the elements of $\mathcal{W}$ we consider a $\ell_2$ penalty to the loss function. As expert trajectories we used the dataset released by [40]. This is the only hyperparameters configuration we tried using a single seed (using the seed 0) on our method because of the high computation requirements of this environment.

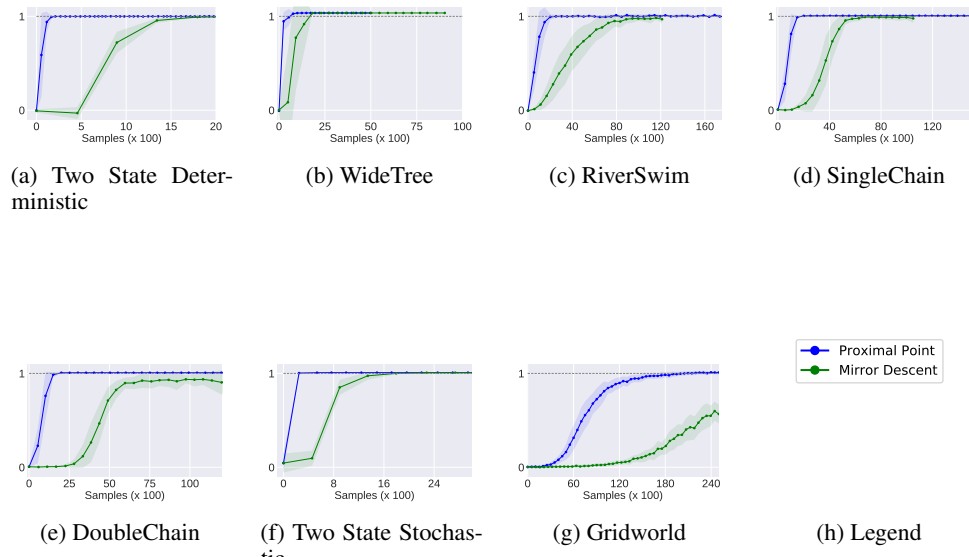

Figure 6: **Proximal Point vs Mirror Descent.** Comparison of proximal point and mirror descent in tabular domains. Averages of 10 seeds.

## L.7  Hyperparameters for MuJoCo (continuous control)

The policy network outputs a distribution over continuous action and is parametrized by independent gaussian distributions for every component of the continuous action vector. We use a three layer neural network to estimate their means and variances. We used as center point in the divergence $D$ the expert feature expectation vector. With further modifications our method can extend also to continuous control tasks in MuJoCo [113]. The main challenge is that the policy improvement step can not be computed in closed form. We therefore approximate it with a SAC architecture as proposed in [40]. We set $\alpha$ to $1e-3$, $\eta$ to $8e-2$, the SAC actor learning rate to $3e-5$ using Adam as optimizer using default values of $\beta_1, \beta_2$, for the critic we used again Adam with learning rate $3e-4$ and default values for $\beta_1, \beta_2$. The actor training of SAC is performed using a transition buffer containing expert and learner data in equal proportion. We used samples from the expert policy to estimate expectations wrt the initial distribution. We avoid using target networks. We tested our algorithm on both the environment `Ant` and `HalfCheetah` using either the data provided in [40] or fresh expert data that we generated training experts with PPO [101]. The results are averaged across 5 seeds. For `Hopper`, we used a larger SAC actor learning rate equal to $2e-4$ and $\alpha = 1e-2$. In addition, we notice that for this environment having a large $\beta_1$ in Adam was harmful. Hence, we used $\beta_1 = 0$.

For `Walker`, we set the actor learning to $1e-4$.

## L.8  Acknowledging existing assets and license.

We built on the code and expert data provided in [40]. They are open sourced for academic scope according to their GitHub page `https://github.com/Div99/IQ-Learn/blob/main/LICENSE.md`.

## L.9  On the importance of the dataset

We observed that the performance of our imitation learning algorithm and IQ-Learn can be affected by the choice of the expert data. In particular, in Appendix L.9, we show that IQ-Learn works better with the expert data provided in [40].

## L.10  Hardware

We ran the experiments on our internal cluster.

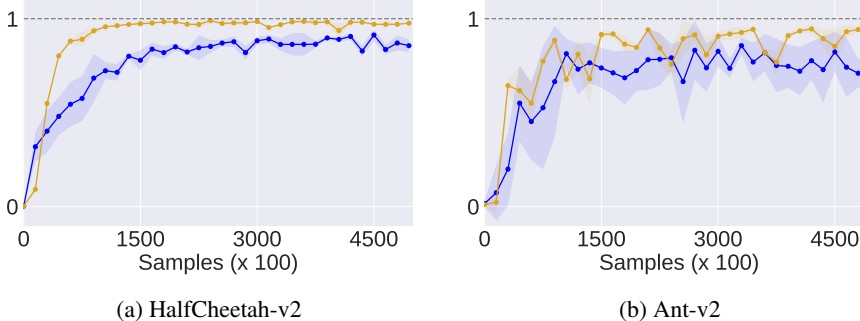

(a) HalfCheetah-v2          (b) Ant-v2

Figure 7: Experiments in the MuJoCo environments with the expert data provided by [40]. The blue line is proximal point while the yellow line is IQLearn.

## M   Recovered Costs

A unique algorithmic feature of the proposed methodology is that we can explicitly recover a cost along with the $Q$-function without requiring adversarial training. In Figures 8 and 3, we visualize our recovered costs in several simple tabular environments (River Swim, Single Chain, Double Chain, and Gridworld, respectively). Most importantly, we verify that the recovered costs induce nearly optimal policies w.r.t. the unknown true cost function. Compared to IQ-Learn, the we do not require knowledge or further interaction with the environment. Therefore, the recovered cost functions show promising transfer capability to new dynamics.

We experimented with a transfer reward setting on a Gridworld (Figure 4). We consider two different Gridworld MDP environments, say $M$ and $\widetilde{M}$, with opposite action effects. This means that action Down in $\widetilde{M}$ corresponds to action Left in $M$ and vice versa. Similarly, the effects of Up and Right are swapped between $\widetilde{M}$ and $M$. We denote by $\mathbf{V}^{\pi}_{\widetilde{M},\mathbf{c}_{\text{true}}}$ (resp. $\mathbf{V}^{\star}_{\widetilde{M},\mathbf{c}_{\text{true}}}$) the value function of policy $\pi$ (resp. optimal value function) in the MDP environment $\widetilde{M}$ with cost function $\mathbf{c}_{\text{true}}$. Moreover, we denote by $\pi^{\star}_{M,\mathbf{c}}$ the optimal policy in the MDP environment $M$ under cost function $\mathbf{c}$. We notice that the recovered cost induces an optimal policy for the new dynamics while the imitating policy fails. Albeit, cost transfer is successful in this experiment we do not expect this fact to be true in general because we do not tackle the issue of cost shaping [87].

### M.1   Preliminary theoretical arguments

We have some preliminary theoretical arguments justifying the near optimality of the recovered costs/rewards. We present briefly the reasoning.

For brevity, we consider the case $\mathcal{W} = B_1^m$. Then $\pi_{\text{E}}$ is optimal for the IL problem. Moreover, for simplicity, we consider the case $\mathbf{\Phi} = \mathbf{I}$. Otherwise, in the following derivations, we replace $\mathbf{Q}$-values by parameterized $\mathbf{Q}_{\boldsymbol{\theta}}$.

Let $(\widehat{\mathbf{w}}_K, \widehat{\mathbf{Q}}_K)$ be the output (average iterate) of $\text{P}^2\text{IL}$ after $K$ outer loop iterations. We give a sketch of proof that $\widehat{\mathbf{w}}_K$ converges to an optimal solution to the inverse problem as $K \to \infty$, i.e., $\widehat{\mathbf{w}}_K$ converges to some $\mathbf{w}_{\text{A}} \in \mathcal{W}$ such that $\pi_{\text{E}}$ is optimal for $\mathbf{c}_{\mathbf{w}_{\text{A}}}$. To this end, we first introduce the following definition.

**Definition 1.** *We say that $\mathbf{w} \in \mathcal{W}$ is $\varepsilon_1$-optimal and $\varepsilon_2$-feasible for the (Dual) program if-f there exists $\mathbf{V} \in \mathbb{R}^{|\mathcal{S}|}$, such that*

$$\langle \boldsymbol{\mu}_{\pi_{\text{E}}}, \mathbf{c}_{\mathbf{w}} \rangle - (1-\gamma)\langle \boldsymbol{\nu}_0, \mathbf{V} \rangle \quad \leq \quad \varepsilon_1, \tag{82}$$

$$\mathbf{c}_{\mathbf{w}} - (\mathbf{B} - \gamma\mathbf{P})\mathbf{V} \quad \geq \quad -\varepsilon_2 \mathbf{1}. \tag{83}$$

*In this case, $\mathbf{V} \in \mathbb{R}^{|\mathcal{S}|}$ is called a certificate.*

Note that the definition of $\varepsilon_1$-optimality for the (Dual) program follows from the fact that the dual optimal value is $\zeta^{\star} = 0$. Moreover, in the definition of $\varepsilon_2$-feasibility we have relaxed the nonnegativity constraint in the dual program (Dual). We make the following conjecture.

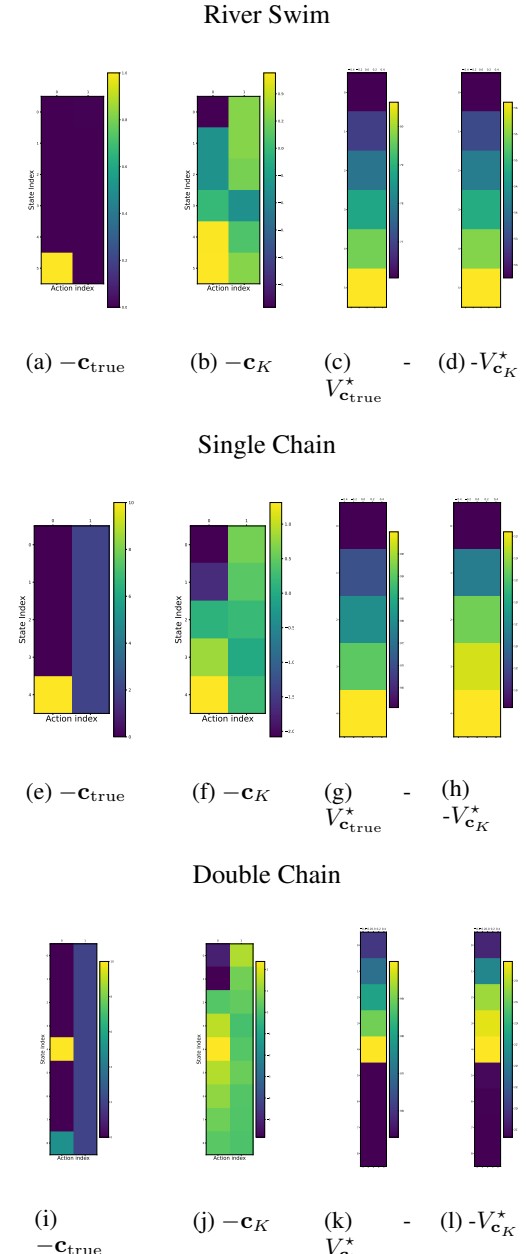

Figure 8: **Recovered Costs.** Comparison between the true cost $\mathbf{c}_{\text{true}}$ and the cost $\mathbf{c}_K$ recovered by $\mathrm{P}^2\mathrm{IL}$. We notice that the optimal value functions $V^\star_{\mathbf{c}_{\text{true}}}$ and $V^\star_{\mathbf{c}_K}$ present the same pattern. Hence, the optimal policy with respect to $\mathbf{c}_K$ is nearly optimal with respect to $\mathbf{c}_{\text{true}}$.

**Conjecture:** For a sufficiently large number of samples $N = \mathcal{O}\Big(\text{poly}\big(\frac{1}{\varepsilon}, \log(\frac{1}{\delta}), m\big)\Big)$, with probability at least $1 - \delta$, the output cost weight $\widehat{\mathbf{w}}_K$ is $\varepsilon$-optimal and $\varepsilon$-feasible for the (Dual) program, with certificate the corresponding logistic value function $\mathbf{V}_{\widehat{\mathbf{Q}}_K}$.

This is easy to show for the exact PPM updates, since $(\mathbf{d}_{\widehat{\pi}_K}, \mathbf{d}_{\widehat{\pi}_K}, \widehat{\mathbf{w}}_K, \mathbf{V}_{\widehat{\mathbf{Q}}_K}, \widehat{\mathbf{Q}}_K)$ is a saddle-point of the (SPP). The proof needs much more effort for the inexact updates used in the sampling-based algorithm.

**Lemma 18.** *Assume that $\widetilde{\mathbf{w}}$ is $\varepsilon_1$-optimal and $\varepsilon_2$-feasible for the (Dual) program. Then, $\pi_{\mathrm{E}}$ is $(\varepsilon_1 + \varepsilon_2)$-optimal for $\mathbf{c}_{\widetilde{\mathbf{w}}}$.*

*Proof.* There exists $\widetilde{\mathbf{V}} \in \mathbb{R}^{|\mathcal{S}|}$, such that

$$
\begin{align}
\langle \boldsymbol{\mu}_{\pi_{\mathrm{E}}}, \mathbf{c}_{\widetilde{\mathbf{w}}} \rangle - (1-\gamma)\langle \boldsymbol{\nu}_0, \widetilde{\mathbf{V}} \rangle &\leq \varepsilon_1, \tag{84}\\
\mathbf{c}_{\widetilde{\mathbf{w}}} - (\mathbf{B} - \gamma\mathbf{P})\widetilde{\mathbf{V}} &\geq -\varepsilon_2\mathbf{1}. \tag{85}
\end{align}
$$

Let $\widetilde{\pi}$ be an optimal policy for $\mathbf{c}_{\widetilde{\mathbf{w}}}$. Then, we have that

$$
\left\langle \boldsymbol{\mu}_{\widetilde{\pi}}, \mathbf{c}_{\widetilde{\mathbf{w}}} - (\mathbf{B} - \gamma\mathbf{P})\widetilde{\mathbf{V}} \right\rangle \geq -\varepsilon_2 \langle \boldsymbol{\mu}_{\widetilde{\pi}}, \mathbf{1} \rangle = -\varepsilon_2.
$$

By using that $(\mathbf{B} - \gamma\mathbf{P})^{\mathsf{T}}\boldsymbol{\mu}_{\widetilde{\pi}} = (1-\gamma)\boldsymbol{\nu}_0$, we equivalently that

$$
\langle \boldsymbol{\mu}_{\widetilde{\pi}}, \mathbf{c}_{\widetilde{\mathbf{w}}} \rangle - (1-\gamma)\langle \boldsymbol{\nu}_0, \widetilde{\mathbf{V}} \rangle \geq -\varepsilon_2.
$$

Therefore,

$$
\langle \boldsymbol{\mu}_{\mathrm{E}}, \mathbf{c}_{\widetilde{\mathbf{w}}} \rangle \leq (1-\gamma)\langle \boldsymbol{\nu}_0, \widetilde{\mathbf{V}} \rangle + \varepsilon_1 \leq \langle \boldsymbol{\mu}_{\widetilde{\pi}}, \mathbf{c}_{\widetilde{\mathbf{w}}} \rangle + \varepsilon_1 + \varepsilon_2.
$$

Thus, $\pi_{\mathrm{E}}$ is $(\varepsilon_1 + \varepsilon_2)$-optimal for $\mathbf{c}_{\widetilde{\mathbf{w}}}$. $\qquad\square$

**Claim:** As $K \to \infty$ one may approach as closely as desired an optimal solution to the inverse problem.

*Proof for the ideal PPM updates.* We recall that by Proposition 4, the set of such solutions is characterized as the set of $\mathbf{w}$-optimizers to (Dual).

Let $\widehat{\mathbf{V}}_K = \mathbf{V}_{\widehat{\mathbf{Q}}_K}$. By the conjecture, for all $K$, we have

$$
\begin{align}
\langle \boldsymbol{\mu}_{\pi_{\mathrm{E}}}, \mathbf{c}_{\widehat{\mathbf{w}}_K} \rangle - (1-\gamma)\langle \boldsymbol{\nu}_0, \widehat{\mathbf{V}}_K \rangle &\leq \varepsilon_K, \tag{86}\\
\mathbf{c}_{\widehat{\mathbf{w}}_K} - (\mathbf{B} - \gamma\mathbf{P})\widehat{\mathbf{V}}_K &\geq -\varepsilon_K\mathbf{1}, \tag{87}
\end{align}
$$

for some sequence $\{\varepsilon_K\}_{K=1}^{\infty}$ such that $\lim_{K\to\infty} \varepsilon_K = 0$. The sequence $\{\widehat{\mathbf{w}}_K\}_{K=1}^{\infty} \subset \mathcal{W}$ is bounded and so there exists a subsequence $\{\widehat{\mathbf{w}}_{K_l}\}_{l=1}^{\infty}$, such that $\lim_{l\to\infty} \widehat{\mathbf{w}}_{K_l} = \mathbf{w}_{\mathrm{A}}$, for some $\mathbf{w}_{\mathrm{A}} \in \mathcal{W}$. Similarly, by Proposition 3 the sequence $\{\widehat{\mathbf{V}}_{K_l}\}_{l=1}^{\infty}$ is bounded and so there exists a subsequence $\{\widehat{\mathbf{V}}_{K_{l_n}}\}_{n=1}^{\infty}$, such that $\lim_{n\to\infty} \widehat{\mathbf{V}}_{K_{l_n}} = \mathbf{V}_{\mathrm{A}}$, for some $\mathbf{V}_{\mathrm{A}}$. By Equations(86)–(87), we have that for all $n \in \mathbb{N}$,

$$
\begin{align}
\langle \boldsymbol{\mu}_{\pi_{\mathrm{E}}}, \mathbf{c}_{\widehat{\mathbf{w}}_{K_{l_n}}} \rangle - (1-\gamma)\langle \boldsymbol{\nu}_0, \widehat{\mathbf{V}}_{K_{l_n}} \rangle &\leq \varepsilon_{K_{l_n}}, \tag{88}\\
\mathbf{c}_{\widehat{\mathbf{w}}_{K_{l_n}}} - (\mathbf{B} - \gamma\mathbf{P})\widehat{\mathbf{V}}_{K_{l_n}} &\geq -\varepsilon_{K_{l_n}}\mathbf{1}. \tag{89}
\end{align}
$$

Taking $n \to \infty$, we end up that

$$
\begin{align}
\langle \boldsymbol{\mu}_{\pi_{\mathrm{E}}}, \mathbf{c}_{\mathbf{w}_{\mathrm{A}}} \rangle - (1-\gamma)\langle \boldsymbol{\nu}_0, \mathbf{V}_{\mathrm{A}} \rangle &\leq 0, \tag{90}\\
\mathbf{c}_{\mathbf{w}_{\mathrm{A}}} - (\mathbf{B} - \gamma\mathbf{P})\mathbf{V}_{\mathrm{A}} &\geq 0. \tag{91}
\end{align}
$$

Equivalently,

$$
\begin{align}
\langle \boldsymbol{\mu}_{\pi_{\mathrm{E}}}, \mathbf{c}_{\mathbf{w}_{\mathrm{A}}} \rangle - (1-\gamma)\langle \boldsymbol{\nu}_0, \mathbf{V}_{\mathrm{A}} \rangle &= 0, \tag{92}\\
\mathbf{c}_{\mathbf{w}_{\mathrm{A}}} - (\mathbf{B} - \gamma\mathbf{P})\mathbf{V}_{\mathrm{A}} &\geq 0. \tag{93}
\end{align}
$$

Therefore, by Proposition 4, $\pi_{\mathrm{E}}$ is optimal for $\mathbf{c}_{\mathbf{w}_{\mathrm{A}}}$.

$\qquad\square$