# OpenReview forum: "Proximal Point Imitation Learning"
_NeurIPS.cc/2022/Conference — NeurIPS 2022 Accept_

### Official Review · Reviewer_7EDZ · 2022-07-08

**Rating:** 8
**Confidence:** 4
**Soundness:** 4 excellent
**Presentation:** 4 excellent
**Contribution:** 4 excellent

**Summary:**

The paper presents a novel method for imitation learning that is based on the LP formulation for MDPs. More precisely, the method uses the recently repopularized technique of splitting the occupancy measures to also obtain the Q-function as Lagrangian multiplier, and optimizes this program using a proximal point method with the regularizers used by Q-REPS. The resulting algorithm is similar to the state-of-art IQ-Learn algorithm and differs primarily by using a "logistic Bellman evaluation" objective for optimizing the Q-function (and jointly the reward function).
The paper provides a thorough theoretical analysis in the linear MDP setting (transition matrix and rewards are linear for given features), bounding the suboptimality depending on the evaluation/estimation errors (except for the policy improvements which are assumed optimal), and evaluates the methind in tabular settings, for offline imitation learning, and for continous control problems, where it performs overall better than IQ-Learn.

**Questions:**

1. Compared to IQ-Learn, the proposed method explicitly learns a reward function. Intuitively, I would expect the parametric form of the reward function to perform better than the implicit form (referring to the way IQ-Learn recovers it from Q and Value function). I was a bit suprised, that the paper does not discuss the use of the reward function, given that one of the main algorithmic properties is that an explicit reward function is learned along with Q function. Why?

2. How are the gradients estimated in the deep learning setting?

**Limitations:**

The paper states the theoretical assumptions, but overall the limitations should be better discussed.

**Strengths And Weaknesses:**

## Originality
Imitation learning based on the LP formulation was already introduced in the classical work by Syed & Schapire (2007) (MWAL). However, the proposed algorithm, significantly deviates from MWAL, mainly by building on the recent advances by Bas-Serrano & Neu (2020) to derive a bi-linear saddle point problem that contains the Q-function as Lagrangian multiplier, and by orientating algorithmically on the state-of-art IQ-Learn algorithm by Garg et al. (2021). The resulting work is clearly novel, both algorithmically as well as theoretically.

## Quality
The derivation of the algorithm is presented clearly and I checked the proofs up to proposition 3. That is, I did not check the proofs related to the "biased stochastic gradient estimates" and the theoretical bounds, which are quite involved and not in my field of expertise. However, I have no reason to believe that they are of worse quality compared to the rest of the work.

The empirical analysis is good by including tabular settings where the assumptions can be met, as well as continuous settings, where the practical competitiveness could be demonstrated. However, the latter could be improved by adding more experiments on different environments.

It would be very interesting to evaluate the learned reward function, by--after convergence--discarding the Q-function and policy, and learning a new policy from scratch from the learned reward function. Would the reward succeed in inducing a good policy? Would it even transfer better, compared to Q-function/policy with respect to changes in the dynamics?


## Clarity
I like the presentation of the paper a lot. There is a nice flow starting from the imitation learning objective Eq. 1 in the background section (Section 2) to the presentation of the saddle point problem and the PPM update equations in Section 4, which are the basis of the proposed algorithm. Only the discussion of the "Q-convex" dual (cf. Proposition 2) breaks the flow a bit. I wonder, why the authors chose to discuss the dual formulation at this place. The path "(Primal) -> (Primal') -> (Dual')" would seem more logically to me than "(Primal) -> (Dual) -> (Primal')".

The paper ends quite abruptly after briefly describing the experimental settings and results. I think the paper would be better structured by adding a conclusion for reflecting on the main insights (LP formulation can be used for competitive IL algorithm with nice properties such as joint reward-cost optimization and logistic Bellman errors) and for discussion the limitations of the current work (limited  evaluation in challenging environments).

I also think that the gradient estimates (6), (7) could be presented better. Or actually, I wonder, if the equations of the gradients need to be shown in the main document at all. I briefly looked into the code (thanks for submitting it!) and it seems like the gradients a computed by backpropagating through a different objective than Eq. 5. Would it be possible to state the objective that is used computing the gradients and refer to the appendix for showing that these gradients are biased estimates of the gradients of (5)?

The projections in Algorithm 1, should be briefly discussed in the text.

To save some space for additional discussions, I would consider integrating section 5 in section 4 (only briefly stating the theoretical results / theorems) and maybe removing the discussion of the dual in section 3 (at the cost of slightly extending the introduction to Eq. 2 in Section 4).

## Significance
I found the paper very interesting both from a theoretical and practical perspective, and believe that also many other researchers in the field of imitation learning could be inspired by the work.

## Minor
Line 706: The most important merits of the primal-dual algorithm [...] I wouldn't use "merits" for describing bad properties.

---

> ### Author Response · Authors · 2022-08-02
> **On the presentation**
>
> Thank you for your time and effort in reviewing our paper and for your fruitful suggestions. We are pleased by the overall positive evaluation of our work, and in particular, we appreciate your positive feedback on the novelty, quality, and significance of our contributions. Your comments have been instrumental in improving our manuscript and inspired exciting future research directions. Below we address the main raised points.
>
> **Minor reorganization of the paper** We will make our best effort to take into consideration all of the suggestions and further improve the clarity and presentation of the paper. In particular, in the revised manuscript, we will make the following rearrangements:
>
> -**Discussion of the "Q-convex" dual** Following your suggestion, we will move the discussion of the (Dual) program from Section 3 to Appendix C. We opted to make this suggested rearrangement to keep the paper focused and not break the flow.
>
> -**Conclusion and discussion** The initial manuscript includes a discussion of our work's limitations and several interesting future directions (Appendix B, lines 748-782). Using the extra page allowed in the final version (if accepted) we will add a final section in the main text (7. Discussion and Outlook) to reflect on the main insights and discuss the limitations of the current work. We then plan to refer the reader to Appendix B for a detailed presentation of promising future directions. For completeness, we also present here a possible text for the conclusion. We appreciate any feedback you may have.
>
> "In this work, we studied a proximal point inverse Q-learning (PPIQ-Learn) algorithm with both theoretical guarantees and convincing empirical performance. Our methodology is rooted in classical optimization tools and the LP approach to MDPs. More precisely, the method uses the recently repopularized overparameterization technique to obtain the Q-function as a Lagrangian multiplier and solves the associated program using a PPM with appropriately chosen Bregman divergences. The most significant merits of PPIQ-Learn are the following:
>
> (I) It optimizes a convex and smooth logistic Bellman evaluation objective over both cost and Q-functions. In particular, it avoids instability due to adversarial training and can also recover an explicit cost along with Q function;
>
> (II)In the context of linear MDPs, it comes with efficient resource guarantees and error bounds for the suboptimality of the learned policy (Theorem 2, Corollary 1). In particular, given polynomially many samples in 1/ε, log(1/$\delta$), and feature dimension m, it recovers an $\epsilon$-optimal policy, w.p $1-\delta$. Notably, the bound is independent of the size of the state-action space.
>
> (III) Beyond the linear MDP setting, it can be implemented in a model-free manner, for both online and offline setups, with general function approximation without losing its theoretical specifications. This is justified by providing an error propagation analysis (Theorems 1 and 4), guaranteeing that small optimization errors lead to high-quality output policy;
>
> (IV) It enjoys not only strong theoretical guarantees but also favorable empirical performance.
>
> At the same time, our newly introduced methods bring challenges and open questions. One interesting question is whether one can accelerate the PPM updates and improve the convergence rate. Next, PPIQ-Learn requires approximately solving a small-dimensional optimization program repetitively. Can one avoid the inner optimization loop and take only one gradient ascent step for the cost and $Q$-values at each iteration? Moreover, the theoretical analysis assumes that the policy improvement steps can be implemented precisely. Can one provide an error propagation analysis that also accommodates policy optimization errors? Lastly, is it possible to provide rigorous arguments for the near-optimality of the recovered cost function? We hope our new techniques will be useful to future algorithm designers and lay the foundations for overcoming current limitations and challenges. In Appendix B, we point out in detail a few interesting directions. On the practical side, can PPIQ-Learn perform well in environments more challenging than MuJoCo and Atari ?”
>
> -**Empirical objective** Following your suggestion, we will present the details of the gradient estimates only in Appendix in the revised manuscript. We plan to make this suggested rearrangement to create more space for the discussion and outlook section. Moreover, for completeness of exposition, we will present the complete formula of the empirical logistic policy evaluation objective in Algorithm 3 (line 1144) and state that these biased empirical objectives can be optimized using the same reasoning as in Bas-Serrano et al., 2021 use the Donsker-Varadhan formula. We will provide the complete formulas in the Appendix for self-containment.
>
> -**Projections in Algorithm 1** We will introduce the notation used for the projections in Algorithm 1.

---

> > ### Author Response · Authors · 2022-08-02
> > **On the recovered cost ( Part 1)**
> >
> > **Compared to IQ-Learn, the proposed method explicitly learns a reward function. Intuitively, I would expect the parametric form of the reward function to perform better than the implicit form (referring to the way IQ-Learn recovers it from Q and Value function). I was a bit surprised that the paper does not discuss the use of the reward function, given that one of the main algorithmic properties is that an explicit reward function is learned along with Q function. Why?**
> >
> > Thank you for the suggestion regarding the use of the reward function. In a first place, we focused on providing guarantees for the imitating policy and we did not investigate this direction but you motivated us to look into this property of our algorithm from both an empirical and theoretical side. We elaborate on this in the next sections.
> >
> > **Empirical results**
> >
> > We performed some preliminary experiments in tabular environments, and noticed that the recovered reward function resembles the one optimized by the expert. The fact that they are not identical is not surprising given the reward-shaping phenomenon (Ng et al., 1999).
> > We report a comparison between true and learned reward in Double Chain, Single Chain, River Swim, and Gridworld. Most importantly, we verified that the recovered reward induces a nearly optimal policy with respect to the unknown true reward function. The results can be visualized at the following anonymous link: https://imgur.com/a/13h2BIQ
> > Moreover, compared to IQ-Learn, we do not require the knowledge of the transition dynamics to recover the state-action reward function.
> > Regarding the transfer capability to new dynamics, we experimented on Gridworld and used the recovered reward to learn a policy in a new environment where the effects of actions are swapped. We noticed that the recovered reward induces an optimal policy while the imitating policy fails. The results can be seen at the following anonymous link: https://imgur.com/a/7KdRIlK.
> > We do not expect this to hold for any perturbation of dynamics and in any possible environment because we can not ensure identifying the true reward up to a constant shift. This seems to be generally possible only when given access to multiple experts (Cao et al., 2021).
> > We will add the figures at the end of Section 6 (Experiments) and, for more details, we will refer the reader to Appendix.
> >
> > (Ng et al., 1999) Policy Invariance under reward transformation: theory and applications to reward shaping, ICML, 1999
> >
> > (Cao et al., 2021) Identifiability in Inverse Reinforcement Learning, NeurIPS, 2021.
> >
> > **Theoretical result**
> > We have some preliminary theoretical arguments justifying the near optimality of the recovered costs/rewards. We present the reasoning briefly. ( Due to the space limitation the reasoning will continue in the next answer. )
> >
> > For brevity, we consider the case $\\mathcal{W}=B_1^m$. Then $\\pi_{\\textup{E}}$ is optimal for the IL problem. Moreover, for simplicity, we consider the case $\\boldsymbol{\\Phi}=\\mathbf{I}$. Otherwise, in the following derivations, we replace $\\mathbf{Q}$-values by parameterized $\\mathbf{Q}_{\\boldsymbol{\\theta}}$.
> >
> > Let $(\\hat{\\mathbf{w}}\_K,\\widehat{\\mathbf{Q}}\_K)$ be the output (average iterate) of PPIQ-Learn after $K$ outer loop iterations. We give a sketch of proof that  $\widehat{\mathbf{w}}\_K$ converges to an optimal solution to the inverse problem as $K\rightarrow\infty$, i.e., $\widehat{\mathbf{w}}\_K$ converges to some $\mathbf{w}\_{\textup{A}}\in\mathcal{W}$ such that $\pi\_{\textup{E}}$ is optimal for $\mathbf{c}\_{\mathbf{w}\_{\textup{A}}}$. To this end, we first introduce the following definition.
> >
> > **Definition**
> > We say that $\mathbf{w}\in\mathcal{W}$ is $\varepsilon\_1$-optimal and $\varepsilon\_2$-feasible for the dual program (introduced in the main text) if-f there exists $\mathbf{V}\in\mathbb{R}^{|\mathcal{S}|}$, such that
> > $$\\langle\\boldsymbol{\\mu}\_{\\pi\_{E}},\\mathbf{c}\_{\\mathbf{w}}\rangle-(1-\gamma)\langle\boldsymbol{\nu}\_{0},\mathbf{V}\rangle \le \varepsilon\_1$$
> > $$\mathbf{c}_{\mathbf{w}}-(\mathbf{B}-\gamma\mathbf{P})\mathbf{V} \geq  -\varepsilon\_2\mathbf{1}.$$
> >
> > In this case, $\mathbf{V}\in\mathbb{R}^{|\mathcal{S}|}$ is called a certificate.
> >
> > Note that the definition of $\varepsilon\_1$-optimality for the dual program follows from the fact that the dual optimal value is $\zeta^\star=0$. Moreover, in the definition of $\varepsilon\_2$-feasibility we have relaxed the nonnegativity constraint in the dual dual program . We make the following conjecture.
> >
> > **Conjecture:** For a sufficiently large number of samples $N=\mathcal{O}\Big(\textup{poly}\big(\frac{1}{\varepsilon},\log(\frac{1}{\delta}\big),m)\Big)$, with probability at least $1-\delta$, the output cost weight $\widehat{\mathbf{w}}\_K$ is $\varepsilon$-optimal and $\varepsilon$-feasible for the dual program, with certificate the corresponding logistic value function $\mathbf{V}\_{\widehat{\mathbf{Q}}\_K}$.

---

> > > ### Author Response · Authors · 2022-08-02
> > > **On the recovered cost (Part 2)**
> > >
> > >
> > > The conjecture is easy to show for the exact PPM updates, since $(\mathbf{d}\_{\widehat{\pi}\_K},\mathbf{d}\_{\widehat{\pi}\_K},\widehat{\mathbf{w}}\_K,\mathbf{V}\_{\widehat{\mathbf{Q}}\_K},\widehat{\mathbf{Q}}\_K)$ is a saddle-point of the (SPP). The proof needs much more effort for the case of inexact updates used in the sampling-based algorithm.
> > >
> > > **Lemma**
> > > Assume that $\widetilde{\mathbf{w}}$ is $\varepsilon\_1$-optimal and $\varepsilon\_2$-feasible for the dual program. Then, $\pi\_{\textup{E}}$ is $(\varepsilon\_1+\varepsilon\_2)$-optimal for $\mathbf{c}\_{\widetilde{\mathbf{w}}}$.
> > >
> > > **Proof**
> > > There exists $\mathbf{\widetilde{V}}\in\mathbb{R}^{|\mathcal{S}|}$, such that
> > >
> > > $$\langle\boldsymbol{\mu}\_{\pi\_{\textup{E}}},\mathbf{{c}}\_{\mathbf{\widetilde{w}}}\rangle-(1-\gamma)\langle\boldsymbol{\nu}\_{0},\mathbf{\widetilde{V}}\rangle \le  \varepsilon_1,$$
> > >
> > > $$ \mathbf{c}\_{\mathbf{\widetilde{w}}}-(\mathbf{B}-\gamma\mathbf{P})\mathbf{\widetilde{V}} \geq  -\varepsilon\_2\mathbf{1}.$$
> > >
> > > Let $\widetilde{\pi}$ be an optimal policy for $\mathbf{c}\_{\widetilde{\mathbf{w}}}$.
> > > Then, we have that
> > > $$\big\langle\boldsymbol{\mu}\_{\widetilde{\pi}},\mathbf{c}\_{\mathbf{\widetilde{w}}}-(\mathbf{B}-\gamma\mathbf{P})\mathbf{\widetilde{V}}\big\rangle\geq -\varepsilon\_2 \langle\boldsymbol{\mu}\_{\widetilde{\pi}},\mathbf{1}\rangle=-\varepsilon\_2.$$
> > > By using that $(\mathbf{B}-\gamma\mathbf{P})^\intercal\boldsymbol{\mu}\_{\widetilde{\pi}}=(1-\gamma)\boldsymbol{\nu}\_0$, we equivalently that
> > > $$
> > > \langle\boldsymbol{\mu}\_{\widetilde{\pi}},\mathbf{c}\_{\mathbf{\widetilde{w}}}\rangle-(1-\gamma)\langle\boldsymbol{\nu}\_{0},\mathbf{\widetilde{V}}\rangle\geq-\varepsilon_2.
> > > $$
> > > Therefore,
> > > $$\langle\boldsymbol{\mu}\_{\textup{E}},\mathbf{c}\_{\widetilde{\mathbf{w}}}\rangle\le(1-\gamma)\langle\boldsymbol{\nu}\_{0},\mathbf{\widetilde{V}}\rangle+\varepsilon\_1\le\langle\boldsymbol{\mu}\_{\widetilde{\pi}},\mathbf{c}\_{\mathbf{\widetilde{w}}}\rangle+\varepsilon\_1+\varepsilon\_2.$$
> > > Thus, $\pi\_{\textup{E}}$ is $(\varepsilon\_1+\varepsilon\_2)$-optimal for $\mathbf{c}\_{\widetilde{\mathbf{w}}}$.
> > >
> > >
> > > **Claim:** As $K\rightarrow\infty$ one may approach as closely as desired an optimal solution to the inverse problem.
> > >
> > > **Proof**
> > > We recall that by Proposition 2 (in the main text), the set of such solutions is characterized as the set of $\mathbf{w}$-optimizers to dual.
> > >
> > > Let $\widehat{\mathbf{V}}\_K=\mathbf{V}\_{\widehat{\mathbf{Q}}\_K}$.
> > > By the conjecture, for all $K$, we have
> > > $$\langle\boldsymbol{\mu}\_{\pi\_{\textup{E}}},\mathbf{c}\_{\widehat{\mathbf{w}}\_K}\rangle-(1-\gamma)\langle\boldsymbol{\nu}\_{0},\widehat{\mathbf{V}}\_K\rangle \le \varepsilon\_K.    \text{(Limit 1)}$$
> > >
> > > $$\mathbf{c}\_{\widehat{\mathbf{w}}\_K}-(\mathbf{B}-\gamma\mathbf{P})\widehat{\mathbf{V}}\_K \geq -\varepsilon\_K\mathbf{1}.  \text{(Limit 2)}$$
> > > for some sequence $\\{\varepsilon\_K\\}\_{K=1}^\infty$ such that $\lim\_{K\rightarrow\infty}\varepsilon\_K=0$.
> > > The sequence $\\{\widehat{\mathbf{w}}\_K\\}\_{K=1}^\infty\subset\mathcal{W}$ is bounded and so there exists a subsequence $\\{\widehat{\mathbf{w}}\_{K\_l\}\\}\_{l=1}^\infty$, such that $\lim\_{l\rightarrow\infty}\widehat{\mathbf{w}}\_{K\_l}=\mathbf{w}\_{\textup{A}}$, for some $\mathbf{w}\_{\textup{A}}\in\mathcal{W}$.
> > >
> > > Similarly, by Proposition 4 (in the main text) the sequence $\\{\widehat{\mathbf{V}}\_{K\_l}\\}\_{l=1}^\infty$ is bounded and so there exists a subsequence $\\{\widehat{\mathbf{V}}\_{K\_{l\_n}}\\}\_{n=1}^\infty$, such that $\lim\_{n\rightarrow\infty}\widehat{\mathbf{V}}\_{K\_{l\_n}}=\mathbf{V}\_{\textup{A}}$, for some $\mathbf{V}\_{\textup{A}}$. By Equations(Limit 1 and Limit 2), and taking $n\rightarrow\infty$, we conclude that
> > >
> > > $$
> > > \langle\boldsymbol{\mu}\_{\pi_{\textup{E}}},\mathbf{c}\_{{\mathbf{w}}\_{\textup{A}}}\rangle-(1-\gamma)\langle\boldsymbol{\nu}\_{0},{\mathbf{V}}\_{\textup{A}}\rangle \le 0,$$
> > > $$
> > > \mathbf{c}\_{{\mathbf{w}}\_{\textup{A}}}-(\mathbf{B}-\gamma\mathbf{P}){\mathbf{V}}\_{\textup{A}} \geq  0 .
> > > $$
> > > Equivalently,
> > > $$
> > > \langle\boldsymbol{\mu}\_{\pi\_{\textup{E}}},\mathbf{c}\_{{\mathbf{w}}\_{\textup{A}}}\rangle-(1-\gamma)\langle\boldsymbol{\nu}\_{0},{\mathbf{V}}\_{\textup{A}}\rangle = 0,$$
> > >
> > > $$\mathbf{c}\_{{\mathbf{w}}\_{\textup{A}}}-(\mathbf{B}-\gamma\mathbf{P}){\mathbf{V}}\_{\textup{A}}\geq 0.$$
> > >
> > > Therefore, by Proposition 2, $\pi_{\textup{E}}$ is optimal for $\mathbf{c}\_{\mathbf{w}\_{\textup{A}}}$.
> > >
> > > Although these are some preliminary formal arguments, this approach is quite intriguing,   requires careful design and analysis, and will be the subject of future work. We will add this discussion in Appendix B (Future works) of the revised paper.

---

> > > > ### Author Response · Authors · 2022-08-02
> > > > **On the deep learning implementation and additional experiment in Walker2d**
> > > >
> > > > **How are the gradients estimated in the deep learning setting?**
> > > >
> > > > In the experiment with neural network function approximation, we use the offline estimator of the logistic Bellman error function given in Algorithm 3. In particular, at each epoch we sample a subset of expert state-action pairs from the expert dataset and we compute the empirical objective  $\hat G$ in Algorithm 3. Then, similarly to  Bas-Serrano et al., 2021  we can use the Donsker-Varadhan formula for deriving a max-min objective. equivalent to the optimization of $\hat G $ (See Bas-Serrano et al., 2021 Proposition 2). At this point the gradients are estimated via the backpropagation algorithm on this objective. We add the reference and the complete formulas in Appendix for self-containment.
> > > >
> > > > In Appendix L.7, we write “We used as center point in the divergence expert feature expectation vector”. This implies that we are using the empirical objective $\hat G$ in Algorithm 3 but we will expand Appendix L.7 to make sure that this connection is clear.
> > > >
> > > > **empirical analysis … could be improved by adding more experiments on different environments**
> > > >
> > > > To improve the empirical analysis, we ran an additional experiment on Walker2d, where our PPIQ-Learn is still performing better than IQLearn. The results can be seen at the following anonymous link: https://imgur.com/a/bYu43r7 . We will add the figure in Section 6 (Experiments).
> > > >
> > > > Thanks again for your thorough review, your positive evaluation and the suggestions that helped us improve the quality of the submission.

---

### Official Review · Reviewer_7mcb · 2022-07-12

**Rating:** 5
**Confidence:** 3
**Soundness:** 4 excellent
**Presentation:** 3 good
**Contribution:** 2 fair

**Summary:**

Based on the proximal point method, this paper propose a new algorithm to solve the minimax formulation of the imitation learning problem, which avoids the nested policy evaluation and cost updates. Based on the assumption of linearly parametrized cost classes, the paper further provides a theoretical analysis for the required number of expert transitions, trajectories and number of concave maximization problems to solve. Finally the paper provides an empirical study on several RL environments with both linear and non-linear function approximations.

**Questions:**

See "Strengths And Weaknesses" Section.

**Limitations:**

See "Strengths And Weaknesses" Section. No potential negative societal impact.

**Strengths And Weaknesses:**

Stengths:
- Detailed introduction to related work and background.
- Good writing and presentation quality.
- Solid math derivations.

Comments, Questions & Limitations:
- The motivation / key challenge to solve in this paper is not clear to me. It seems the derivations based on proximal point method may lead to a new algorithm for imitation learning, but what are the unique advantages (especially in terms of practical applications) of the proposed approach and what are the limitations of previous works this work has solved?
- I understand that the resulting algorithm avoids nested optimization / alternating updates. But this does not seem to be a unique feature of the proposed method and many previous IL methods also enjoy that property.
- The theoretical analysis is for linear functions, which may be restrictive. Moreover, I do not see particular insights from the theoretical analysis. If the main take-away message is the estimation error can be made arbitrarily small by increasing the number of expert transitions or computational resources in general, that is not very exciting. Is the sample complexity bound state-of-the-art, and is the bounds meaningful in practice (for example $\epsilon^{-5}$ sample transitions may be too large to be useful in practice)?
- The policy update step $\pi_k \propto \pi_{k-1} e^{- \alpha Q_{\theta_k}}$ seems to be something like density ratio correction of previous policy. Can you elaborate more on that part, and in particular the intuitions?
- From figure 2, the final performance of PPM and IQLearn seems similar in most cases, but PPM is indeed more sample efficient than other methods.

---

> ### Author Response · Authors · 2022-08-02
> **Answers to specific questions ( Part 1)**
>
> Thank you for your time, effort, and constructive comments on our manuscript.
> Below please find our response to the points you raised.
> We hope they will help strengthen your positive opinion about the submission.
> We would like to highlight that our work is mainly theoretical.
> Please let us know if we can provide further clarification to support this statement.
>
>
>
> **“The motivation / key challenge to solve in this paper is not clear to me. It seems the derivations based on proximal point method may lead to a new algorithm for imitation learning, but what are the unique advantages (especially in terms of practical applications) of the proposed approach and what are the limitations of previous works this work has solved?”**
>
>
> Our main motivation is to bridge the gap between theory and practice and design algorithms that enjoy both strong theoretical guarantees and favorable empirical performance.
>
> The first practical contribution is the derivation from first principles of a new objective that jointly learns reward and $Q$-values from expert data. This objective is significantly different from that of IQ-Learn  (c.f., our objective in Equation (5) with Equation (9) in IQLearn) and results in convincing empirical performance. Indeed, PPIQ-Learn that learns $Q$-functions minimizing this objective seems to perform better than IQLearn in the experiments.
> A second practical advantage over other scalable algorithms like IQ Learn, GAIL, ASAF and SQUIL is that we explicitly learn a reward function that may be used to transfer the optimal policy in an environment with perturbed dynamics. We would like to kindly ask you to check the comments of Reviewer 7EDZ and the additional experiments at the following anonymous link: https://imgur.com/a/7KdRIlK
> Importantly, learning a reward function does not require alternating updates. Simply jointly maximizing equation (5) seems to be a unique advantage of our algorithm.
> Apart from these practical advantages, we believe that the main motivation for our work is theoretical. We would kindly ask you to go through the answer to Reviewer s345 titled **On the contribution and the novelty** for a discussion of the theoretical benefits of our algorithm.
>
> **I understand that the resulting algorithm avoids nested optimization / alternating updates. But this does not seem to be a unique feature of the proposed method and many previous IL methods also enjoy that property.**
>
> To the best of our knowledge, this is the first work that proposes an IL policy evaluation objective that optimizes jointly reward and Q-functions. This joint optimization avoids instability due to adversarial training (as in IQ-Learn) and can also recover an explicit reward along with the Q-function without requiring knowledge or interaction with the environment (which is not the case in IQ-Learn).
> This seems a unique algorithmic feature of the proposed methodology.
>
> Indeed, other methods like IQLearn, ASAF (Barde et al., 2020) and SQUIL (Reddy et al., 2020) avoid alternated value function and reward updates, but they do not explicitly recover a reward function. Moreover, algorithms that recover a reward function like OAL [35] and AIRL [5] require alternated updates between reward and value functions. Please see also the detailed answer to Reviewer 7EDZ with title (Extract a policy from the recovered reward) for more insights in this direction.
>
> We would be happy to engage in further discussion and be grateful if you could let us know if you are aware of any other algorithm that enjoys the property of learning both a value and cost function without using alternating updates.
>
> (Barde et al., 2020) Adversarial soft advantage fitting: Imitation learning without policy optimization., NeurIPS, 2020
>
> (Reddy et al., 2020) Squil: Imitation learning via reinforcement learning with sparse reward.,ICLR, 2020.

---

> > ### Author Response · Authors · 2022-08-02
> > **Answers to specific questions ( Part 2 )**
> >
> > **“The theoretical analysis is for linear functions, which may be restrictive. Moreover, I do not see particular insights from the theoretical analysis. If the main take-away message is that the estimation error can be made arbitrarily small by increasing the number of expert transitions or computational resources in general, that is not very exciting. Is the sample complexity bound state-of-the-art, and is the bounds meaningful in practice (for example ϵ−5 sample transitions may be too large to be useful in practice)?”**
> >
> > **“The theoretical analysis for linear functions may be restrictive.”**
> >
> > To the best of our knowledge the linear MDP assumption is widely accepted in the RL theory community (see for example references [83-88] in the main text. In addition, the work of [Ren et al. 2021] suggests that it is possible to learn linear representation of MuJoCo environments. We also stress the fact that our linear MDP setting is different from linear control because the features are allowed to be nonlinear functions. Despite being linear, our transition law can still have infinite degrees of freedom. This is a substantial difference from the recent theoretical works on IL [34,35,36,37,40] which consider either tabular MDPs [35], or a linear quadratic regulator [36], or a linear transition law that can be completely specified by a finite-dimensional matrix [34,40]. In the last case, the degrees of freedom are bounded, and thus mitigate the challenges in estimating the transition model. Indeed, the linear MDP setting studied in [34,40] reduces the unknown dynamics problem to estimating an unknown finite-dimensional matrix, which differs from our nonparametric approach. (lines 170-174 Main Text) (lines 683-688 App.) We also note that [40,41] require the restrictive assumption of bounded concentrability coefficients, while this is not the case for the analysis in our paper (lines 34-36 and 690-693).
> > The only theoretical IL works that consider general MDPs are [70,39]. However, the authors in [70] provide local optimality convergence guarantees, i.e., convergence to a stationary point. On the contrary, our algorithm provides global convergence guarantees for linear MDPs  (Lines 692-695). Moreover our error propagation analysis justifies using our derived actor-critic scheme with general function approximation. A scalable deep reinforcement learning implementation is possible, without losing the theoretical guarantees of Theorems 1 and 4 (lines 287-291). Compared to the primal-dual algorithm [39] (i) we do not require a generative model; (ii) we avoid restrictive coherence assumption (iii) we avoid the problematic occupancy measure approximation (lines 706-714).
> > Finally, our theoretical analysis contains several blocks that may be of independent interest. Mainly the proof of Proposition 4 that derives a contraction operator from the program Primal’ to bound the dual variables. We are also excited about the analysis of the biased stochastic gradient estimation (Appendix H) where we realized that we could use tools from random design analysis of Ridge Regression. This connection seems to be new.
> >
> > **“If the main take-away message is that the estimation error can be made arbitrarily small by increasing the number of expert transitions or computational resources in general, that is not very exciting.”**
> >
> > We highlight that this fundamental theoretical RL question is not addressed by algorithms that do not have theoretical guarantees for stochastic gradient estimates (as IQ-Learn). Moreover, notice that we provide nonasymptotic guarantees, i.e., a finite sample analysis and explicit error bounds on the performance of the extracted policy. That is, we can say how fast the error decreases while computation resources, expert demonstrations and sample transitions increase. In addition, our analysis has the challenge that the gradient estimates are unavoidably biased. And this is the case for the IQ-learn objective as well. In both objectives an expectation with respect to transition dynamics appears inside a nonlinear function. We were able to prove that for the linear MDP case the bias can be controlled (Theorem 2).

---

> > > ### Author Response · Authors · 2022-08-02
> > > **Answers to specific questions ( Part 3 )**
> > >
> > > **“Is the sample complexity bound state-of-the-art, and is the bounds meaningful in practice (for example ϵ−5 sample transitions may be too large to be useful in practice)?”**
> > >
> > >  To our knowledge, such guarantees in this setting are provided for the first time (lines 34-37). Therefore it is state of the art.  By “efficient” (see also lines 23-27 in Section 1) we mean efficient in both runtime and sample complexity (number of expert trajectories and sample transitions). The computational and sample complexities should not depend on the number of states but should depend on an intrinsic complexity measure of the function class (Jin et. al., (2019)). In particular, we show that given polynomially many samples in $1/\varepsilon$, $\log(1/\varepsilon)$, and feature dimension $m$, PPIQ-Learn recovers an $\epsilon$-optimal policy, w.p $1-\delta$. It is an open question if the PPM algorithm can be accelerated or if there exists an algorithm with a better convergence rate. We expect that the answer is affirmative.
> > >
> > > To further address the question regarding the sample complexity of $\varepsilon^{-5}$, we can say that for a widely used algorithm like REPS (Peters et al.,2010), a recent work (Pacchiano et al., 2021) has proven a worst sample complexity bound $\varepsilon^{−8}$. Assuming that this bound is tight and considering the empirical performance of the algorithm, we think that $\varepsilon^{-5}$ is acceptable in practice.
> > >
> > > **“The policy update step $π_k ∝ π_{k−1} e^{−\alpha Q_{\theta_k}}$ seems to be something like density ratio correction of previous policy. Can you elaborate more on that part, and in particular the intuitions?”**
> > >
> > > In the RL context, this update (also called softmax update) appears in regularized MDPs with causal entropy regularization (also known as soft MDPs) to promote safe exploration, induce multimodal risk-sensitive policies, improve empirical performance , or model observed behavior of imperfect agents. See for example (Ziebart et al., 2010, Fox et al, 2016, Zhou et al., 2018, Haarnoja et al., 2018). Intuitively, a softmax policy update in terms of a $Q$-function is a multimodal, “softened” version of the corresponding (deterministic) greedy policy.
> > >
> > > Causal entropy regularization is also used in the objective of IQ-Learn and GAIL. Therefore, the policy updates for discrete actions in IQ-Learn are also softmax updates. In our case, we derive this update as a consequence of the choice of the divergence H.
> > >
> > > In optimization literature, a similar update (also called multiplicative weight update)      arises when using proximal gradient descent with the KL-divergence as Bregman divergence. It first appeared in mirror descent (Nemirovsky & Yudin, 1983) and in online learning (Vovk 1990, Littlestone & Warmuth 1994, Auer 1995).
> > > It is unclear to us how this update can be explained as a density ratio correction for importance sampling and estimation wrt to offline data set. We would be grateful if you could share with us your view from this perspective.
> > >
> > > (Ziebart et al.,  2010) Modeling Interaction via the Principle of Maximum Causal Entropy, ICML, 2010
> > >
> > > (Fox et al., 2016 ), Taming the noise in reinforcement learning via soft updates, UAI,  2016
> > >
> > > (Zhou et al. 2018), Infinite time horizon maximum causal entropy inverse reinforcement learning, TAC, 2018
> > >
> > > (Haarnoja et al., 2018) Soft Actor-Critic: Off-Policy Maximum Entropy Deep Reinforcement Learning with a Stochastic Actor, ICML, 2018
> > >
> > > ( Vovk, 1990). Aggregating Startegies, Computational Learning Theory, 1990
> > >
> > > (Littlestone and Warmuth, 1994) The weighted majority voting algorithm, Information and Computation, 1994.
> > >
> > > (Auer et al., 1995) Gambling in a rigged casino: The adversarial multi-armed bandit problem, Foundations of Computer Science, 1995
> > >
> > > **“From figure 2, the final performance of PPM and IQLearn seems similar in most cases, but PPM is indeed more sample efficient than other methods.”**
> > >
> > > We agree, the final performance is similar to the one reached by IQLearn in most of our experiments. We think that it is reasonable to expect the final performance to be similar to the expert one.

---

### Official Review · Reviewer_s345 · 2022-07-13

**Rating:** 5
**Confidence:** 2
**Soundness:** 3 good
**Presentation:** 2 fair
**Contribution:** 2 fair

**Summary:**

This paper proposed an imitation learning algorithm to handle both online and offline setting. First, an intractable convex problem is obtained by relaxing the primal problem. After that, the saddle point problem is derived by applying Lagrangian decomposition. Finally, the authors provided an practical algorithm based on the problem, named PPIQ-Learn. For the policy evaluation, PPIQ-Learn uses a convex and smooth objective over both cost and Q-value function. In addition, the theoretical analyses support the performance guarantee. Finally, the provided experiments show that the proposed algorithm outperforms baselines.

**Questions:**

I enjoyed reading this paper, but I seem to have missed the main contribution of this paper (especially, compared to the IQ-Learn). It would be very helpful to understand if the authors clarify the following questions.

- I'm not sure why PPIQ-Learn shows better performance than IQ-learn. Would you explain the main contribution (or intuitive explanation) of PPIQ-Learn compared to IQ-Learn?
- Is there any reason to choose D to measure the distance between $\lambda$ and $\lambda’$? In addition, why the authors use fixed $\eta$ instead of optimizing $\eta$?
- The definition of $H$ is different from that of simple KL-divergence – sample $(s,a)$ from $d$ and measure the log-ratio between $\pi_d$ and $\pi_{d’}$ instead of the log-ratio between $d$ and $d’$. Is there any reason or justification?
- Why authors use only 2 MuJoCo domains (Ant and HalfCheetah) out of 4 domains used in IQ-Learn?
- I am curious about the feature matrix used in the actual experiments. Would you explain about the choice of feature matrix?
- Is it difficult to simply combine PPM and IQ-Learn? If so, what is the problem? If not, what are the advantages of PPIQ-Learn over such simple combining?


**Ethics Review Area:**

["I don’t know"]

**Limitations:**

Yes

**Strengths And Weaknesses:**

Strengths:

Theoretical analyses support the performance of the proposed algorithm.

The proposed algorithm shows convincing empirical results.

It can be applied to both online and offline imitation learning problem

Weakness:

The novelty of this work is not clear – IQ-Learn also uses a single convex objective to evaluate the policy and handles both online and offline setting.

---

> ### Author Response · Authors · 2022-08-02
> **On the contribution and the novelty**
>
> Thank you for the time, effort, and constructive comments on our manuscript. Below please find our response to the points you raised. We hope they will help strengthen your positive opinion about the submission.
>
> **Main Contributions and Novelty of the Paper**
>
> - **Contributions** Our work is mainly theoretical. Provably efficient IL algorithms remain largely unexplored. To tackle this longstanding question, we present PPIQ-Learn, a scalable IL algorithm with theoretical guarantees rooted in classical optimization tools and the LP approach to MDPs. The most significant merits of PPIQ-Learn are the following: **(I) (Joint Optimization over cost and Q-functions)** It optimizes a convex and smooth logistic Bellman evaluation objective over both cost and Q-functions. In particular, it avoids instability due to adversarial training and can recover an explicit cost along with Q-function. **(II) (Linear MDP setting)** For the context of linear MDPs, it comes with efficient resource guarantees and probabilistic error bounds for the suboptimality of the learned policy (Theorem 2, Corollary 1) under mild assumptions, significantly weaker than those found in the literature until now. To our knowledge, we provide such guarantees in this setting for the first time (lines 34-37). By "efficient" (lines 23-27), we mean efficient in both runtime and sample complexity. Notably, the computational and sample complexities do not depend on the number of states. **(III) (Beyond the linear MDP setting)** It can be implemented for online and offline setups with general function approximation without losing its theoretical specifications. This is justified by our error propagation analysis (Theorems 1 and 4), which guarantees that small optimization errors lead to high-quality output policy (lines 287-291) and (lines 695-700); **(IV)  (Empirical Performance)** It enjoys not only strong theoretical guarantees but also favorable empirical performance for both linear and neural network function approximation (Section 6, lines 716-717).
>
> - **Novelty** In order to state our research question and situate it among prior related theoretical works, we have provided a literature review in Section 1 (lines 44-76) as well as an extended literature review in Appendix A (lines 672-747). We summarize the main points here.
>   1. (Lines 170-174 Main Text) (lines 683-688 App.) Despite being linear, our transition law can still have infinite degrees of freedom. This is a substantial difference from the recent theoretical works on IL [34,35,36,37,40] which consider either tabular MDPs [35], or a linear quadratic regulator [36], or a linear transition law that can be completely specified by a finite-dimensional matrix [34,40]. In the last case, the degrees of freedom are bounded, and thus mitigate the challenges in estimating the transition model.
>   2.  (Lines 692-695) The authors in [70] consider general MDPs but only provide local optimality convergence guarantees, i.e., convergence to a stationary point. On the contrary, our algorithm provides global convergence guarantees for the linear MDP setting.
>   3.  (Lines 706-714) Compared to the primal-dual algorithm [39] (i) we do not require a generative model; (ii) we avoid restrictive coherence assumption (iii) we avoid the problematic occupancy measure approximation.
>   4. (Lines 722-76 Main text, Lines 745-747 ) Our techniques can be used to improve upon the best rate for REPS in the tabular setting ($\varepsilon^{-8}$) improving it to ($\varepsilon^{-5}$) and to extend their guarantees to linear MDPs.
>
>
> We hope that our detailed discussion clarifies our novelty. Please also note that given that an extra page is available for the final version, we will be able to include the extended literature review of Appendix A in the main text.

---

> > ### Author Response · Authors · 2022-08-02
> > **Answers to specific questions ( Part 1 )**
> >
> > **“I'm not sure why PPIQ-Learn shows better performance than IQ-learn. Would you explain the main contribution (or intuitive explanation) of PPIQ-Learn compared to IQ-Learn?”**
> >
> > **[Comparison with state-of-the-art IQ-Learn]** IQ-Learn  also avoids adversarial training by learning a single Q-function (Eq. (9) in IQ-Learn). However, although its model-free implementation achieves significant empirical success, it remains hampered by limited theoretical understanding (see also line 61). This is the main difference from our work, which enjoys both favorable practical performance and strong theoretical guarantees. We provide a more detailed comparison between IQ-Learn and our theoretical work.
> >
> > 1. **[Different IL objective]** (see also lines 66-71) IQ-Learn considers (causal) entropy regularization in the objective and can be seen as smoothing using uniform distribution as the center point. On the other hand, in our online IL algorithm, instead of regularizing the objective, the key idea is to penalize the divergence between the current policy and the policy obtained at the previous iteration. We do so by employing a Bregman proximal point update.
> >
> > 2. **[Different policy evaluation objective]** Compared to IQ-Learn, our algorithm optimizes a convex and smooth logistic policy evaluation objective over both cost and Q-functions (Eq. (5), line 203). On the other hand, the policy is parametrized as a function of Q-values in both methods. This fact suggests that the difference in performance comes from our policy evaluation objective. It is also worth noting that our joint optimization over cost and Q-functions avoids instability due to adversarial training (as in IQ-Learn) and can also recover an explicit cost along with the Q-function without requiring knowledge or access to samples from the environment (which is not the case in IQ-Learn). Please see also the detailed answer to Reviewer 7EDZ with title (Extract a policy from the recovered reward) for more insights in this direction.
> >
> > 3. **[Theoretical Guarantees]** As we have already highlighted, the fundamental difference between IQ-Learn and our work is that IQ-Learn’s convergence properties remain largely elusive in the function approximation and model-free regime. In the IQ-Learn paper, the authors prove concavity of their objective over Q-functions (Eq. (9)) and, therefore, can guarantee asymptotic convergence to the global optimum for tabular MDPs with known dynamics. In other words, Table 1 in the IQ-Learn paper refers to a much “weaker” notion of theoretical guarantees: asymptotic convergence for exact gradient, known dynamics, and small-sized tabular MDPs. However, this viewpoint does not address some of the most fundamental theoretical RL questions, including **(a) (Error Propagation Analysis)** How do they cope with policy evaluation errors in their actor-critic scheme?
> > **(b) (Function Approximation)** It is unclear whether their actor-critic algorithm converges to a global optimum or if it converges at all, even for the simple linear function approximation setting. **(c) (Finite sample analysis)** Does the sampling-based variant of their algorithm converge? If yes, what is the sample complexity, i.e., how many sample transitions are needed to guarantee a near-optimal policy? **(d) (Biased estimates)** How do they cope with the biased empirical objective and gradient estimates? Can they control the bias somehow?
> > In our understanding, these questions are unanswered for the IQ-Learn algorithm. On the other hand, we provide answers for all the above-mentioned points and prove a scalable finite time bound by handling biased stochastic gradients. To our knowledge, the IL literature is missing an algorithm enjoying a comparable theoretical bound. Of course, our newly introduced methods bring challenges and open questions. Please see also our answer to Reviewer 7EDZ with title (Conclusion and Discussion).
> >
> > **“Is there any reason to choose D to measure the distance between λ and λ′?”**
> >
> > The choice of KL-divergence as Bregman divergence for the $\lambda$-variable living in the probability simplex comes with the following merits (1) We can get close form updates for the variable $\lambda$, (2) We avoid projection to the simplex, (3) In case of no errors in Theorem 1 we obtain a nearly dimension-free rate (dependence only on $\log (1/\beta)$ and $\log |A|$) (Notice that $\log (1/\beta)$  upper bounds the divergence $D$ and $\log |A|$ upper bounds the divergence  $H$ when $\pi\_0$ is uniform). Indeed, this Bregman divergence choice for variables living in the simplex is common in the optimization literature and is known to mitigate the effect of dimension (lines 214-216). A useful reference for this property is, e.g., Ch.4 in Duchi Introductory Lectures on Stochastic Optimization (2016).

---

> > > ### Author Response · Authors · 2022-08-02
> > > **Answers to specific questions ( Part 2 )**
> > >
> > > **“In addition, why do the authors use fixed η instead of optimizing η ?”**
> > >
> > > We did not optimize over $\eta$ because we noticed that this could not improve the rate. The reason is that the $\eta$ dependence appears in the constant $C$ of Thm. 1, in the denominator of the first term of Thm. 1, and in the errors $\epsilon_k$ (as can be seen from Thm. 1 and 2). Given these complicated dependencies, we could not obtain a better rate tuning $\eta$. We also note that this is consistent with the known property of PPM that there is no restriction on the step sizes for the final error bound. See, for example, Section 9.5 and Thm. 9.8 in the textbook (Wright and Recht, Optimization for Data Analysis, (2022)).
> > >
> > > **“The definition of H is different from that of simple KL-divergence – sample (s,a) from d and measure the log-ratio between π_d and π_d′ instead of the log-ratio between d and d′. Is there any reason or justification?”**
> > >
> > > Thanks a lot for the question. This is indeed one of the crucial design choices in our algorithm (lines 217-225). In contrast to the relative entropy this choice is less common and has been popularized by (Bas-Serrano et al., 2021). The corresponding Bregman divergence is generated by the so-called conditional relative entropy (see e.g., Th. 1 in Neu et al. 2017). Note that when the center point is the uniform distribution then we end up with the causal entropy regularizer used in the IL objective of IQ-Learn and GAIL.
> > > The main merit of such particular divergence is that
> > > - (i) it mitigates the effect of dimension,
> > > - (ii) it gives analytical softmin updates for the policy $\pi_{d}$ rather than the occupancy measure $d$ (Eq.(4))
> > > - (iii) these softmin updates do not involve the unknown transition matrix $\mathbf{P}$, and
> > > - (iv) it allows a model-free implementation without the need of a generative model.
> > >
> > > Consequently, we avoid the restrictive coherence assumption on the choice of features needed in (Bas-Serrano & Neu, 2020, Kamoutsi et al., 2021), as well as the biased policy updates appearing in REPS (Peters et al., 2010, Pacchiano et al., 2021).
> > >
> > > **“Why do authors use only 2 MuJoCo domains (Ant and HalfCheetah) out of 4 domains used in IQ-Learn?”**
> > >
> > > Since the core contribution of our paper is theoretical, we limited ourselves to 2 MuJoCo environments. Following your suggestion, we ran an additional experiment on Walker2d which shows that PPIQ-Learn is still performing better than IQLearn. The results can be seen at the following anonymous link: https://imgur.com/a/bYu43r7. This additional Figure will be included in Section 6.
> > >
> > > **“I am curious about the feature matrix used in the actual experiments. Would you explain about the choice of feature matrix?”**
> > >
> > > In all tabular environments of Figure 1 we used indicator features. For the Cartpole and Acrobot we used a 3-layer ELU Neural Network as features and optimized all layers. Alternatively, we could use features generated by a frozen layer of a multi layer perceptron. This is similar to the choices of Bas-Serrano et al. 2021.
> > >
> > > **“Is it difficult to simply combine PPM and IQ-Learn? If so, what is the problem? If not, what are the advantages of PPIQ-Learn over such simple combination?”**
> > >
> > > In our opinion, a combination is difficult. The problem is that the IQ-Learn objective (Equation 8 in their paper) is not “proximable”, i.e., implementing a proximal point update is intractable. That means, we do not have an efficient way to solve the problem
> > > $$\pi\_{k+1} = \min\_\pi d\_\psi (\rho^\pi, \rho^E) - H(\pi) + \frac{1}{\eta} D(\pi,\pi\_k), $$
> > > because $H(\pi)$ and $d\_\psi (\rho^\pi, \rho^E)$ are not convex in $\pi$.
> > >
> > > To develop PPIQ-Learn, we use the LP formulation as a starting point and apply a constraint splitting approach that slightly differs from previous works. In this case, PPM updates have an explicit form, given in Proposition 3, because the problem is convex in the occupancy measures. Choosing an LP as a starting point is crucial for both our algorithmic derivation and theoretical guarantees (that are missing in IQ-Learn).
> > >
> > > On the other hand,the soft-$Q$-values in the IQ-Learn are the regularized optimal state-action value functions, i.e., it holds that $Q\_{\textup{soft}}^\star(s,a)=c(s,a)+\gamma\sum\_{s’}V^\star\_{\textup{soft}}(s’)P(s’|s,a). $ This relation does not hold for the logistic Bellman state-action value functions $Q\_\theta$ and the logistic value function $V\_\theta^k$  and so, in our opinion, it is not possible to follow similar reasoning to the IQ-Learn construction.

---

> > > > ### Comment · Reviewer_s345 · 2022-08-09
> > > > **Response to the authors**
> > > >
> > > > Thanks to the authors for their responses.
> > > >
> > > > Some questions have been addressed and therefore, I will keep the score (somewhat positive).

---

> > > > > ### Author Response · Authors · 2022-08-09
> > > > > **Thanks for the response**
> > > > >
> > > > > Dear reviewer,
> > > > >
> > > > > Thank you for your answer. We are happy that we addressed some of your points. Could you please tell us which are your remaining concerns?
> > > > >
> > > > > Thanks again.
> > > > >
> > > > > Best,
> > > > > Authors

---

### Author Response · Authors · 2022-08-02
**General answer to the reviewers**

Dear reviewers,


Thank you for your time and effort in reviewing our paper and for your fruitful suggestions. It is encouraging to read your comments: “Theoretical analyses support the performance of the proposed algorithm. The proposed algorithm shows convincing empirical results” [Reviewer s345], “Solid math derivations”, “Good writing and presentation quality.” [Reviewer 7mcb] and “The resulting work is clearly novel, both algorithmically as well as theoretically”, “ I found the paper very interesting both from a theoretical and practical perspective, and believe that also many other researchers in the field of imitation learning could be inspired by the work.”[Reviewer 7EDZ].

On our end, we believe that in stark contrast with previous theoretical IL works, our method enjoys not only strong theoretical guarantees but also favorable empirical performance for both linear and neural network function approximation. Moreover, the fundamental difference between the state-of-the-art IQ-Learn and our work is that IQ-Learn’s theoretical properties remain largely elusive in the function approximation and model-free regime.

Below, we carefully address and clarify reviewers’ individual raised points, which we believe notably improved our paper.

---

### Author Response · Authors · 2022-08-07
**Hopper experiment**

Dear reviewers,

as the discussion period ends soon, we would like to remind that we are available for further discussion and kindly ask to acknowledge our rebuttal.
As a follow-up to reviewer [Reviewer s345] and [Reviewer 7EDZ] question about an improved benchmarking in MuJoCo, we can share now the results on the Hopper environment at this anonymous link: https://imgur.com/a/80T3ZPO

In addition, we would be particularly interested in hearing your feedback on the novelty contribution, on the differences with IQLearn, on the additional empirical evidence in Walker (https://imgur.com/a/bYu43r7) and on the experiments to recover an optimal policy from the recovered reward under same (https://imgur.com/a/13h2BIQ) or different transition dynamics (https://imgur.com/a/7KdRIlK).

Thanks again for your constructive feedback.

Best,
Authors

---

> ### Comment · Reviewer_7EDZ · 2022-08-08
> **Re: Hopper Experiment**
>
> Dear authors,
>
> sorry for not replying earlier; I don't have any open questions or additional remarks and now acknowledged reading your rebuttal.
>
> I appreciate the additional MuJoCo experiment, and the result on evaluating the reward function is quite exciting. I think the paper has been further improved by these revisions. However, I will not change my assessment because I already recommended "strong accept" before the rebuttal, and I also do not see a reason to decrease my score based on the other reviews:  I still argue that both the derivation based on the LP formulation and the actual algorithms are novel and highly relevant.

---

> > ### Author Response · Authors · 2022-08-08
> > **Thanks**
> >
> > Dear reviewer,
> >
> > Thanks a lot for your answer, your very positive evaluation and valuable suggestions.
> >
> > Best,
> > Authors

---

### Author Response · Authors · 2022-08-09
**Revised Manuscript and Availability for Discussion**

Dear Reviewers,

We thank you again for your time and effort in reviewing our work. We have revised the manuscript to reflect your comments and suggestions and believe it has been notably improved. To facilitate the comparison between the old and the revised manuscript, we highlight major changes with blue color in the new submission.

The main changes are the following:

- In the extended literature review in Appendix A, we compared our work with state-of-the-art IL
  algorithms. (Lines 769-787). We would like, however, to highlight that our main contribution is theoretical.
- We clarify further our practical and algorithmic contributions in the main text. (Section 4 Lines 218-221, Section 5 Lines 310-317, Section   6 Lines 322-322 and  331-3336)
- To avoid the possibility of confusion and differentiate our algorithmic and theoretical contributions from the IQ-Learn, we changed the name of our algorithm to Proximal Point Imitation Learning (P^2-IL). As argued in the rebuttal and the paper, our method solves the same
   problem but is significantly different from IQ-Learn
- We made several rearrangements suggested by the reviewer 7EDZ
- We have added a final section  (6. Discussion and Outlook) to reflect on the main insights and discuss the limitations of the current
   work. For a detailed discussion of future directions, we refer the reader to Appendix B.
- In Section 5 (Experiments), we demonstrate the convincing performance of our algorithm in two additional  Mujoco environments (Walker
  and Hopper)
- In Section 5, we have added a small paragraph discussing our new experiments and results regarding the near optimality of the
   recovered costs under the same or different transition dynamics. The details and a preliminary theoretical justification are given in
   Appendix M


We hope that our rearrangements and additional discussions improve the paper's clarity and presentation and clarify our novelty and contributions even more. Please also note that given that an extra page is available for the final version, we will be able to include the main points of the extended literature review (Appendix A) as well as new FIgures of Appendix M in the main text.

We are still available to clarify the potential doubts of Reviewers s345 and 7mcb during the last hours of the discussion period.

Your comments have been instrumental in improving our manuscript.


 Thank you again for your time,
And we look forward to hearing from you.

---

### Meta-Review · Area_Chair_b2cF · 2022-08-24

**Recommendation:** Accept
**Confidence:** Certain

**Metareview:**

It was agreed among the reviewers and AC that the paper should be accepted. Hope the authors will address the remaining comments from the reviews in preparing the final version of the paper.

**Award:**

No

---

### Decision · Program_Chairs · 2022-09-14

Accept